# EvolArena: An Evolving Arena for Multi-Turn Reasoning in LLMs

## Abstract

Recent advances in LLMs have shown promising results in complex reasoning tasks. However, current evaluations predominantly focus on single-turn reasoning scenarios, leaving interactive tasks largely unexplored. We attribute it to the absence of comprehensive datasets and scalable automatic evaluation protocols. To fill these gaps, we present **EvolArena**, an **Evol**ving **Arena** for LLMs' multi-turn reasoning evaluation. Comprising 4 classes, 40 tasks, and 3600 instances, EvolArena covers diverse reasoning capabilities, fine-grained difficulty granularity, and necessitates multi-turn interactions with the environments. Moreover, EvolArena features fully-automated framework spanning both dataset constructions and model evaluations, which enables scalable assessment without human interventions. Experiments reveal that even the cutting-edge reasoning models fall short of multi-turn, interactive reasoning tasks. And the further analysis upon these results brings valuable insights for future research in interactive AI systems.

## 1 Introduction

With the emergence of reasoning-enhanced Large Language Models (LLMs), such as o1 (Jaech et al., 2024) and R1 (DeepSeek-AI et al., 2025), significant progress has been made in complex reasoning tasks (Wei et al., 2022; Luo et al., 2024; Ye et al., 2025; Lightman et al., 2024). However, most current evaluations focus on single-turn reasoning in domains like mathematics (Cobbe et al., 2021; Hendrycks et al., 2021), commonsense (Talmor et al., 2019; Zellers et al., 2019), logic reasoning (Han et al., 2024; Team et al., 2025b), and code generation (Jain et al., 2025; Chen et al., 2021), which do not reflect the interactive and iterative nature of real-world problem-solving. But multi-turn reasoning is essential for practical reasoning performance. It enables long-term planning, allows for feedback acquisition and reuse, and supports gradual problem solving through iterative refinement. A key question thus arises: *Can frontier LLMs maintain effective reasoning capabilities in dynamic, multi-turn environments?*

To answer this question, we require a rigorous evaluation framework that captures the dynamic and iterative nature of reasoning. However, as summarized in Table 1, existing approaches fall short of providing a comprehensive solution. Static benchmarks like CodeElo (Quan et al., 2025) and LiveCodeBench Pro (Zheng et al., 2025) predominantly focus on single-turn generation, neglecting the essential capabilities of dynamic state tracking. While real-world agent benchmarks such as AgentBench (Liu et al., 2024) and AgentBoard (Chang et al., 2024) introduce interactivity, they largely assess application-specific skills (e.g., web browsing) within noisy environments and rely on fixed datasets that are susceptible to contamination and saturation. Similarly, benchmarks from the AI planning community (e.g., ACPBench (Kokel et al., 2025)) often frame reasoning as static question-answering rather than long-horizon exploration. Furthermore, interactive frameworks like MT-Bench (Zheng et al., 2023) and GameArena (Hu et al., 2025) are limited by subjective scoring (e.g., LLM-as-a-Judge) or scalability bottlenecks due to human involvement. These limitations highlight the urgent need for a fully automated, deterministic, and evolvable framework dedicated to evaluating pure multi-turn logical reasoning.

To bridge these gaps, we propose a novel multi-turn automated reasoning evaluation framework designed to more accurately evaluate LLMs' comprehensive capabilities in interactive environments. The development of such a benchmark presents two primary challenges: (1) designing effective and diverse multi-turn tasks that can measure the multi-dimensional reasoning capabilities of models and

| Category | Benchmarks | Dynamic Interaction | Deterministic Eval. | Parametric Gen. | Abstract Logic |
|---|---|:---:|:---:|:---:|:---:|
| Static Evaluation | CodeElo (Quan et al., 2025)
LiveCodeBench Pro (Zheng et al., 2025) | | ✓
✓ | | |
| Real-world Agent | AgentBench (Liu et al., 2024)
AgentBoard (Chang et al., 2024) | ✓
✓ | ✓
✓ | | |
| AI Planning | TRAC (He et al., 2023)
ACPBench (Kokel et al., 2025)
ActionReasoningBench (Handa et al., 2025) | | ✓
✓
✓ | | ✓
✓
✓ |
| Interactive/Game | MT-Bench (Zheng et al., 2023)
GameArena (Hu et al., 2025)
SPIN-Bench (Yao et al., 2025) | ✓
✓
✓ | | | |
| **Ours** | **EvolArena** | ✓ | ✓ | ✓ | ✓ |

Table 1: Comparison of EvolArena with representative benchmarks. EvolArena uniquely combines **dynamic multi-turn interaction** with **infinite parametric generation** in a **deterministic** environment, while focusing on **abstract logical reasoning**.

(2) establishing an evolving and automated interactive evaluation framework to facilitate scaling and avoid saturation after model advancement (Perlitz et al., 2024).

To address the first challenge, we focus on constructing tasks that inherently require multi-turn reasoning, where each interaction step introduces new constraints or information that necessitates iterative refinement of the model's reasoning process. To achieve this, we manually collect and validate a set of highly reasoning-intensive tasks from various sources for systematically evaluating four fine-grained reasoning abilities: **Inductive**, **Abductive**, **Deductive**, and **Planning Reasoning** (Seel, 2011; Huang & Chang, 2023). Then for each task, we design a structured problem template that explicitly defines interactive rules, format requirements, and example interactions demonstrating valid exchanges. Through these templates, models are required to engage in active reasoning, gather environmental feedback, and iteratively refine their reasoning process in order to accomplish the given reasoning objective.

As for the second challenge, to enable scalable automated evaluation, we implement three components - **Generator**, **Monitor**, and **Evaluator**, to construct an automated interactive evaluation framework. The generator transforms each problem template into tasks of distinct difficulty levels while ensuring solution feasibility through carefully controlled complexity parameters. With the generator, we can smoothly control the difficulty of reasoning as models' performance improves. The rule-based monitor processes model queries through a two-stage validation system: it first checks query format compliance, then provides rule-specific feedback for valid queries while monitoring whether the given reasoning objectives are achieved. The evaluator assesses completed dialogues across multiple dimensions to provide a comprehensive evaluation of models' sustained reasoning capabilities.

Building upon these design principles, we present **EvolArena**, an evolving evaluation framework that encompasses 40 distinct reasoning tasks designed to assess four reasoning abilities, with each task calibrated across three difficulty levels. Through extensive empirical evaluation of 20 reasoning and non-reasoning models, our analysis reveals that o3-mini demonstrates superior overall performance. Our key findings indicate: (1) As the reasoning difficulty increases, even current frontier models struggle significantly. (2) As the number of reasoning steps increases, the advantage of o3-mini over other models becomes more pronounced, which indicates a potential optimization direction for the open-source community. (3) Reasoning ability is not directly correlated with reasoning efficiency; o3-mini often requires more reasoning steps compared to QwQ-32B and R1 on questions where all three models arrive at correct answers.

In summary, our main contributions are as follows:

• We introduce a high-quality benchmark specifically designed to assess models' reasoning capabilities in multi-turn interactive scenarios.

• We propose an automated framework for evolving multi-turn evaluation, capable of producing problems with tunable complexity. This enables the benchmark to evolve alongside advances in model capabilities.

• Our empirical findings reveal several critical limitations of current models in multi-turn reasoning settings, offering valuable insights for future research directions.

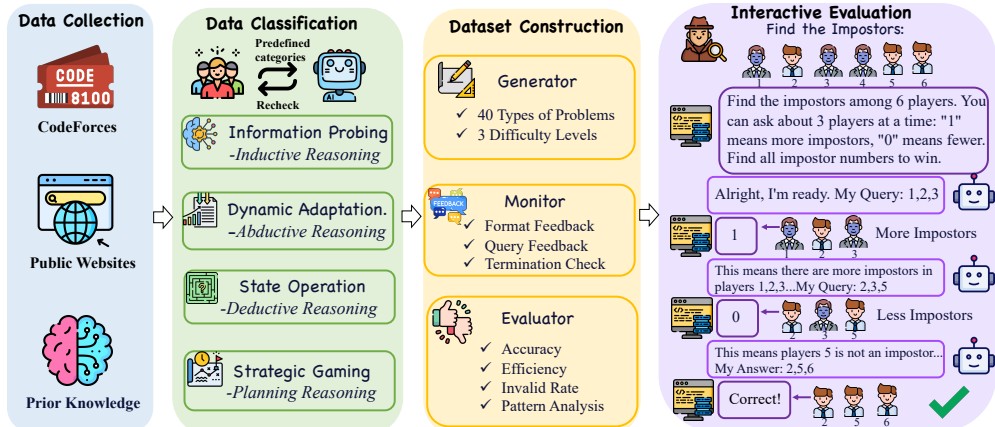

Figure 1: This figure represents the complete framework of our arena, from construction to evaluation. It includes four modules: data collection, data classification, dataset construction, and interactive evaluation. After the dataset is built, the evaluation system can perform automated multi-round interactive evaluations and automatically increase the difficulty of the problems.

## 2 OVERVIEW

In this section, we first propose our automated interactive framework that simulates real-world reasoning scenarios. At its core, the framework enables a model to engage in multiple turns of interaction[1] while maintaining consistent reasoning progress toward solving a given task. Formally, our framework consists of three essential components, including generator, monitor and evaluator, which can work together to create a controlled and automated evaluation environment:

**Generator ($P$)** creates interactive problems with controlled difficulty levels and corresponding reasoning objectives. Formally defined as $p, s = P(t, n, g_n)$, where $p$ represents the generated problem, $s$ defines the reasoning objective, $t$ specifies the problem template, $n$ determines the complexity level, and $g_n$ encodes the corresponding problem parameters. We carefully design $t$ with explicit interaction rules, format requirements and example interactions for each task.

**Monitor ($M$)** generates feedback and determines termination based on the model's query, acting as a deterministic, rule-based environment. The monitoring process can be formalized as: $(m_i, s_i) = M(t, q_i, s_{i-1}, s, I)$, where $s_{i-1}$ and $s_i$ denote the conversation states at turns $i - 1$ and $i$ respectively, and $m_i$ represents the generated feedback for query $q_i$ based on template $t$. The interaction terminates when either the target state $s_i = s$ is achieved or the maximum turn limit $I$ is reached. For each query, $M$ first validates the legality of the query format, then determines whether the current conversation should be terminated, and finally inputs $m_i$ as the response to the model.

**Evaluator ($E$)** assesses multi-turn interactions across multiple dimensions. Formally, $e = E(t, \{(q_1, m_1), ..., (q_T, m_T)\})$, where $T$ denotes the total turns and $e$ encompasses a range of metrics of accuracy, efficiency, invalid rate, and pattern analysis. Specifically,

- **Accuracy (Acc)** measures the proportion of successfully completed tasks. A task is considered successful if and only if its final state $s_T$ matches the task's reasoning objective $s$. Formally, $\text{Acc} = \frac{S_C}{C}$, where $C$ is the total number of tasks and $S_C$ is the number of successful tasks.
- **Efficiency (Eff)** evaluates relative solution efficiency by comparing turn counts on commonly solved tasks between model pairs. For two models $A$ and $B$, let $C_{AB}$ denote their set of commonly solved tasks. The efficiency score of model $A$ over $B$ is computed as: $\text{Eff}_{A,B} = \frac{\sum_{c \in C_{AB}} I(T_A^c < T_B^c)}{|C_{AB}|}$, where $T_A^c$ and $T_B^c$ represent the turn counts for task $c$ by models $A$ and $B$ respectively, and $I(\cdot)$ is an indicator function that equals 1 when the condition is true and 0 otherwise.
- **Invalid Rate (IR)** assesses the proportion of interactions containing invalid operations among all interaction conversations. This metric not only measures the model's ability to follow instructions but also reflects its fundamental reasoning capability to infer valid operations from the current environment. Formally, $\text{IR} = \frac{N_V}{N}$, where $N_V$ is the number of interactions with invalid operations and $N$ is the total number of interactions.

[1]Our tasks involve multi-turn interactions for successful completion. See Appendix D for details.

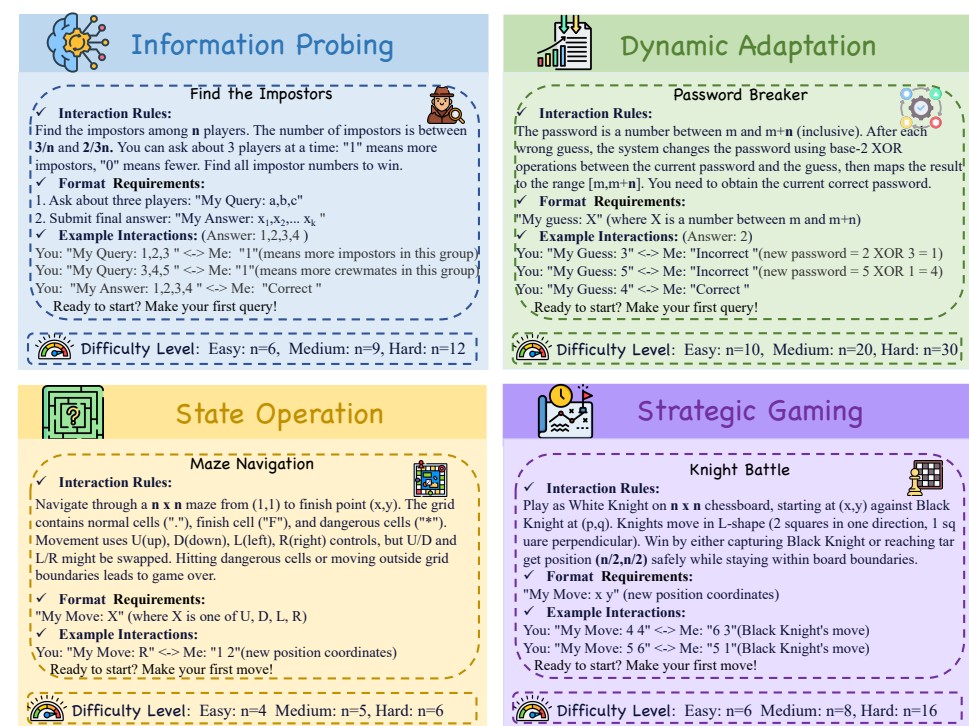

Figure 2: This figure illustrates examples of our four task types. Each task includes interaction rules, query format requirements, and example interactions, with three levels of input difficulty.

- **Pattern Analysis (PA)** examines the model's reasoning patterns across four categories: **Associate** (associating with the original problem), **Verify** (reflecting and verifying the reasoning process), **Plan** (strategically planning subsequent interactions) and **Feedback** (utilizing previous feedback for reasoning). We analyze the occurrence count of each pattern in each interactive turn and calculate $PA_J = \frac{1}{\sum_{c=1}^{C} T_c} \sum_{c=1}^{C} \sum_{i=1}^{T_c} r_{c,i}^J$, where $T_c$ denotes the number of interaction turns for task $c$, and $r_{c,i}^J$ represents the occurrence count of pattern $J$ in the $i$-th turn of task $c$.

Through these components, our framework uses the Generator to create problems, facilitates interactions between the Monitor and models, and ultimately employs the Evaluator to measure performance.

## 3 BENCHMARK CONSTRUCTION

In this section, we first introduce the task classification (§3.1), and then explain how we construct each problem (§3.2), finally we briefly discuss how the interactive evaluation occurs (§3.3) in Figure 1.

### 3.1 DATA CLASSIFICATION

To construct our dataset, we first collect seed tasks from various websites[2][3][4]. To facilitate a systematic analysis of models' reasoning capabilities, we categorize the public seed tasks into four predefined classes as follows using GPT-4o, with subsequent human validation ensuring classification accuracy. While successful task completion generally requires a combination of various reasoning skills, each predefined class is specifically designed to evaluate distinct aspects of reasoning capabilities.

- **Information Probing (IP)**: It involves discovering hidden but fixed information. As shown in Figure 2, in "Find the Impostors", models determine the complete role distribution by querying about different group compositions, with the monitor revealing each group's majority type as clues. In this task, models should progressively eliminate distractors to reach the answer.

---

[2]https://codeforces.com/

[3]https://www.nytimes.com/

[4]Statistics and utilization of raw data are detailed in Appendix E.

- **Dynamic Adaptation (DA)**: Unlike "Information Probing" where answers remain static, this type involves answers that evolve according to deterministic transformation rules. As exemplified in "Password Breaker", each incorrect query triggers specific password modifications based on predefined mechanisms. Success in this type requires models to accurately understand and apply transformation rules to make informed and targeted queries.

- **State Operation (SO)**: This category introduces hidden mechanics, distinguishing it from the previous two categories. For example, in "Maze Navigation", models are required to guide an agent to a target location under an initially unknown control system. Success requires models to rationally analyze the current situation and infer the hidden mechanism through appropriate actions, then proceed with subsequent operations based on this understanding.

- **Strategic Gaming (SG)**: It features adversarial two-player environments where task outcomes depend on the dynamic interaction between model actions and system responses[5]. Taking "Knight Battle" as an instance, models should strategically outpace the system to complete objectives, requiring both competitive awareness and efficient execution.

By leveraging four distinct task categories, we comprehensively assess LLMs' multi-turn reasoning capabilities. Specifically, our framework focuses on the following essential types of reasoning.

- **Inductive Reasoning**: This involves forming general conclusions by identifying patterns from specific observations (Han et al., 2022; Misra et al., 2022; Yang et al., 2024b). For example, in "Find the Imposters" of "Information Probing", models need to gather evidence by querying different group configurations, observe the majority role types within each group, and synthesize these observations to infer the complete role distribution.

- **Abductive Reasoning**: This is the process of inferring the most plausible explanation from limited or incomplete evidence (Seel, 2011; Jung et al., 2022). In "Dynamic Adaptation", where the correct answer evolves according to predefined rules, models require to infer the current state of the target answer based on a limited number of interactions.

- **Deductive Reasoning**: This refers to deriving specific conclusions through the application of known rules or logical implications (Creswell et al., 2023; Saparov & He, 2023). In "State Operation", for instance, models should first infer hidden mechanisms from rule-based environmental feedback and then apply those rules to perform correct reasoning.

- **Planning**: Success in our tasks crucially depends on multi-step planning capabilities (Valmeekam et al., 2023; Huang et al., 2022; Ajay et al., 2023). This is particularly evident in "Strategic Gaming", where models should construct action sequences by anticipating future states and considering both their moves and potential opponent responses.

## 3.2 DATASET CONSTRUCTION

After obtaining the categorized seed task sets, we select 10 representative tasks for each of the four categories, yielding a total of 40 tasks that exhibit diverse interaction patterns and rule structures as detailed in Appendix. Then, we manually convert the seed tasks into structured problem templates. Based on these templates, we develop problem generators with three difficulty levels: "easy", "medium", and "hard". Each level corresponds to different values of $n$, the parameter that determines the task complexity. We further implement monitors tailored to each task's interactive rules, enabling the system to extract model queries, provide real-time feedback, and detect conversation termination. For evaluation purposes, we design task-specific evaluators that assess performance based on the complete conversation history, employing metrics aligned with each task's reasoning objective.

To calibrate difficulty levels, we evaluate task solvability using o3-mini across 10 problems for each $n$, iteratively refining until difficulty gradient exhibits meaningful progression and reasonable feasibility.

Finally, we generate a comprehensive dataset comprising 30 distinct problems per difficulty level for each of 40 tasks, resulting in a total of 3,600 evaluation instances. This structure enables robust and fine-grained assessment of model performance across varying complexity levels.

---

[5]Our experimental results show that models struggle to achieve high accuracy even in simple scenarios with random system actions, leading us to adopt random system responses as our evaluation baseline.

## 3.3 INTERACTIVE EVALUATION

As shown in Figure 2, the interaction process begins with the generator providing the problem to the tested model while passing reasoning objective to the monitor. Upon receiving the problem, the model generates response which is then sent to the monitor. The monitor extracts query from the response, computes appropriate feedback, and returns it to the model. Based on the feedback, the model adjusts its reasoning and continues responding. This iterative cycle repeats until the monitor detects conversation termination conditions. Finally, the evaluator receives the complete conversation history and analyzes it using various metrics.

To illustrate this process, let's consider "Find the Impostors". The generator first creates problems across three difficulty levels by varying the parameter $n$. Along with each problem, it generates reasoning objective in the form of binary sequences of length $n$, where 0 denotes impostors and 1 represents non-impostors (e.g., "000011" for $n = 6$).

During the interaction, the monitor validates model responses against two specific patterns: "My Query: $a, b, c$" and "My Answer: $x_1, x_2, ..., x_k$". Any response not matching these patterns is rejected. For valid queries in the format "My Query: $a, b, c$", the monitor returns "1" if the specified positions contain more impostors according to the answer sequence, and "0" otherwise. When the model submits a final answer, the monitor responds with either "Correct" or "Incorrect" and terminates the conversation if correct. Additionally, the monitor enforces a maximum round limit.

Upon conversation completion, the evaluator processes the entire dialogue history, determining accuracy based on whether the final response received a "Correct" feedback, and calculates other metrics as defined in Section 2.

The difficulty calibration process begins with initial testing using $n = 6, 7, 8$, generating 10 problems with their reasoning objectives per difficulty level. When these values fail to produce sufficient performance gradients, the generator iteratively tests different values until finding suitable ones (e.g., $n = 6, 9, 12$). Once appropriate difficulty parameters are established, we proceed with large-scale evaluation, generating 30 problems per difficulty level and testing them across all models.

## 4 EXPERIMENT

In this section, we conduct extensive experiments to evaluate various LLMs on EvolArena, guided by the following research questions: **- RQ1:** How do current LLMs perform overall on our benchmark? **- RQ2:** How does those LLMs performance vary under increasing reasoning turns? **- RQ3:** Does superior performance equate to greater efficiency in the number of interactions? **- RQ4:** How do the LLMs' instruction following abilities and basic reasoning capabilities under multi-turn scenarios? **- RQ5:** Which reasoning patterns are relatively more important in multi-turn reasoning scenarios?

### 4.1 EXPERIMENT SETUP

**Model Selection** We evaluate both reasoning-enhanced LLMs and non-reasoning LLMs in our experiments. Among the reasoning-enhanced models, we include o3-mini (Jaech et al., 2024), DeepSeek-R1 (DeepSeek-AI et al., 2025), QwQ-32B (Team, 2024), and DeepSeek-R1-Distilled Series (DeepSeek-AI et al., 2025). For non-reasoning models, we select GPT-4o (Hurst et al., 2024), Qwen-Max (Yang et al., 2024a), Gemma-3 (Team et al., 2025a), Qwen2.5 (Yang et al., 2024a), Llama-3.1 (Grattafiori et al., 2024), and Mistral Series (AI, 2025). This diverse selection of both open-source and closed-source models ensures comprehensive coverage of current LLM capabilities in multi-turn reasoning scenarios. [6] [7]

### 4.2 MAIN PERFORMANCE (RQ1)

We first present the overall results of models on four reasoning tasks of our datasets in Table 2. From the results, we can observe the following conclusions:

- **Impact of Task Difficulties**: Across all models, performance decreases progressively from "easy" to "medium" to "hard". This demonstrates the rationality of our dataset's difficulty stratification.

---

[6]For all models, we limit the maximum number of turns to 15 due to the consideration in Appendix F.

[7]See Appendix G for the detailed experimental settings.

| Model | IP | | | DA | | | SO | | | SG | | | AVG | | |
|---|---|---|---|---|---|---|---|---|---|---|---|---|---|---|---|
| | E | M | H | E | M | H | E | M | H | E | M | H | E | M | H |
| *Reasoning Model* | | | | | | | | | | | | | | | |
| o3-mini | 60.33 | 41.56 | 28.22 | 40.33 | 24.18 | 17.13 | 38.61 | 27.00 | 20.22 | 85.00 | 74.44 | 59.17 | 56.07 | 41.80 | 31.19 |
| R1 | **39.22** | 25.00 | 11.11 | 34.58 | **23.11** | **15.22** | 47.67 | 38.56 | 32.78 | 73.00 | 62.67 | 57.67 | 48.62 | **37.33** | **29.19** |
| QwQ-32B | 53.56 | **28.22** | **19.00** | **38.33** | 20.44 | 12.00 | 36.67 | **29.89** | **25.33** | 70.00 | 56.33 | 46.00 | **49.64** | 33.72 | 25.58 |
| R1-Distill-Llama-70B | 33.78 | 13.11 | 6.33 | 25.50 | 11.00 | 5.67 | 15.56 | 10.78 | 7.89 | 61.11 | 44.17 | 28.89 | 33.99 | 19.76 | 12.19 |
| R1-Distill-Qwen-32B | 26.78 | 10.11 | 3.22 | 10.50 | 3.22 | 1.67 | 7.11 | 4.22 | 3.11 | 39.44 | 24.44 | 15.28 | 20.96 | 10.50 | 5.82 |
| R1-Distill-Qwen-7B | 3.89 | 2.33 | 1.11 | 0.44 | 0.00 | 0.00 | 0.67 | 1.11 | 0.22 | 3.67 | 2.67 | 1.00 | 2.17 | 1.53 | 0.58 |
| R1-Distill-Qwen-1.5B | 0.67 | 0.78 | 0.33 | 0.00 | 1.00 | 0.11 | 0.00 | 0.00 | 0.00 | 0.67 | 0.67 | 0.00 | 0.33 | 0.61 | 0.11 |
| *Non-Reasoning Model* | | | | | | | | | | | | | | | |
| GPT-4o | 29.11 | 10.56 | 6.89 | 22.92 | 11.56 | **7.00** | 19.73 | **15.11** | **11.56** | **42.22** | 30.56 | **22.78** | 28.50 | 16.94 | 12.06 |
| Qwen-Max | 33.89 | 11.56 | 7.33 | 27.42 | 17.67 | 8.11 | 20.15 | 13.67 | 10.78 | 49.17 | 33.61 | 22.50 | 32.66 | 19.13 | 12.18 |
| gemma-3-27b-IT | 31.00 | 9.78 | 9.67 | 18.92 | 9.67 | 6.33 | 16.00 | 10.00 | 5.67 | 16.89 | 4.72 | 5.15 | 20.70 | 8.54 | 6.70 |
| gemma-3-12b-IT | 24.78 | 8.33 | 4.56 | 15.03 | 8.44 | 5.89 | 12.22 | 4.56 | 3.56 | 12.61 | 9.17 | 5.17 | 16.16 | 7.63 | 4.79 |
| gemma-3-4b-IT | 11.44 | 4.56 | 2.44 | 8.61 | 6.00 | 4.11 | 9.00 | 4.22 | 2.89 | 10.67 | 2.33 | 0.67 | 9.93 | 4.28 | 2.53 |
| Qwen2.5-72B-IT | 38.22 | **20.00** | 10.89 | 23.22 | 12.44 | 6.33 | 14.78 | 11.00 | 7.89 | 41.50 | **32.78** | **26.67** | 29.43 | **19.06** | 12.94 |
| Qwen2.5-32B-IT | 33.44 | 14.67 | **12.44** | 19.69 | **12.89** | 6.22 | 23.67 | 17.67 | 14.44 | 42.00 | 25.00 | 19.76 | **29.70** | 17.56 | 13.22 |
| Qwen2.5-7B-IT | 27.44 | 11.44 | 3.67 | 18.33 | 9.33 | 6.22 | 9.67 | 6.00 | 4.89 | 22.67 | 10.00 | 8.33 | 19.53 | 9.19 | 5.78 |
| Qwen2.5-1.5B-IT | 2.22 | 0.11 | 0.22 | 6.44 | 4.33 | 0.78 | 9.44 | 0.89 | 1.33 | 17.67 | 14.67 | 12.00 | 8.94 | 5.00 | 3.58 |
| Llama-3.1-70B-IT | 40.11 | 21.22 | 11.89 | **23.81** | 12.00 | 6.78 | 16.78 | 11.44 | 8.78 | 36.50 | 25.33 | 20.72 | 29.30 | 17.50 | 12.04 |
| Llama-3.1-8B-IT | 22.67 | 10.00 | 4.89 | 13.58 | 5.78 | 4.67 | 12.56 | 5.33 | 3.78 | 11.00 | 5.67 | 3.00 | 14.95 | 6.69 | 4.08 |
| Mistral-Small-24B-IT-2501 | 18.67 | 7.78 | 4.56 | 17.92 | 6.22 | 5.00 | 19.56 | 10.00 | 6.78 | 25.56 | 12.83 | 12.28 | 20.42 | 9.21 | 7.15 |
| Ministral-8B-IT-2410 | 8.89 | 4.22 | 2.00 | 13.69 | 5.67 | 5.11 | 16.67 | 11.56 | 4.33 | 21.33 | 5.33 | 8.67 | 15.15 | 6.69 | 5.03 |
| **AVG** | 27.01 | 12.39 | 12.77 | 18.96 | 10.25 | 6.22 | 17.32 | 11.65 | 8.81 | 34.13 | 23.87 | 18.78 | 24.36 | 14.63 | 10.34 |

Table 2: Model Accuracy on EvolArena. **IT**: Instruction-based models. **IP**: Information Probing. **DA**: Dynamic Adaptation. **SO**: State Operation. **SG**: Strategic Gaming. **E / M / H**: Easy / Medium / Hard. The best results (column-wise) for reasoning and non-reasoning models are highlighted in purple and red, respectively. Their second-best results are shown in **bold**. Table 7 shows accuracy with 95% confidence intervals.

- **Comparison Between Reasoning and Non-Reasoning Models**: When comparing state-of-the-art reasoning models (e.g., o1, R1) with non-reasoning models, it is evident that reasoning models significantly outperform their non-reasoning counterparts. Notably, even smaller-parameter reasoning models (e.g., QwQ-32B) surpass the strongest non-reasoning models within the same series (e.g., Qwen-Max). This highlights the necessity of enhancing reasoning capabilities in model design.

- **Comparison Between Non-Reasoning Models and its Distilled Versions**: Comparing the non-reasoning and reasoning-specific version (e.g., R1-Distill) of the same model series shows nearly equivalent performance. While R1-Distill excels in math and code-related tasks, it fails to generalize effectively on our OOD tasks. This indicates that merely applying SFT distillation is insufficient to generalize reasoning, underscoring the necessity of reinforcement learning (Kirk et al., 2024).

- **Task-Specific Observations**: A closer inspection of individual tasks reveals that while o3-mini consistently outperforms other models, particularly in IP and SG, its performance is similarly to QwQ-32B and R1 in DA and SO. The distinction of the two categories lies in the nature of environmental feedback: in DA and SO tasks, the feedback is less straightforward, requiring models to first correctly interpret the feedback before proceeding with their reasoning. This additional interpretation and reasoning may deviate significantly from training distribution.

- **Performance of Small Models**: Models with fewer than 7B parameters achieve almost no meaningful scores, further emphasizing the difficulty of our benchmark. Consequently, in subsequent analyses, we will focus on models with 32B or more parameters.

## 4.3 TURN ANALYSIS (RQ2)

In this section, we analyze how the number of interaction turns affects model performance. Figure 3 illustrates the accuracy of five representative models across various tasks and difficulty levels, with different numbers of interaction turns. Our analysis focuses on four key perspectives:

- **Task-Specific Analysis**: IP benefits the most from increased interaction turns. In contrast, for DA and SO, additional turns do not always lead to significant performance gains. This suggests that even current reasoning models are primarily strong in direct reasoning based on inductive inference, but still weak in deductive and abductive reasoning, which rely on premise assumptions.

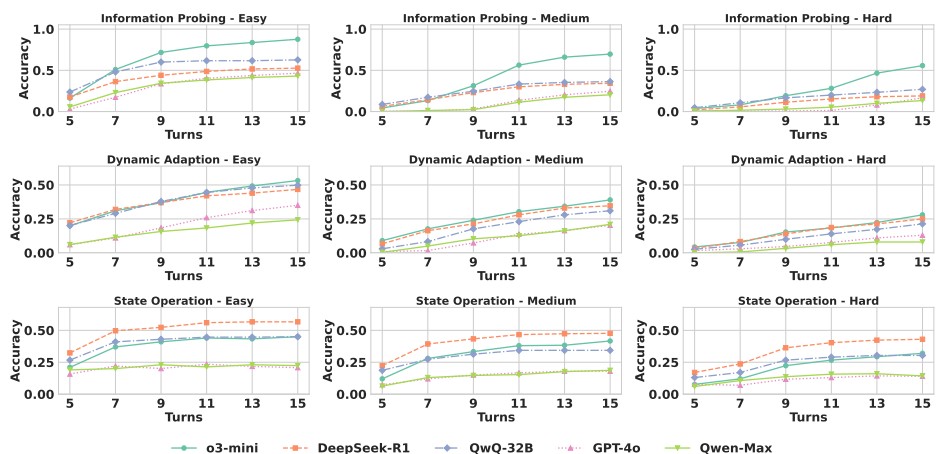

Figure 3: Model accuracy v.s. interaction turns across different tasks and difficulty levels.

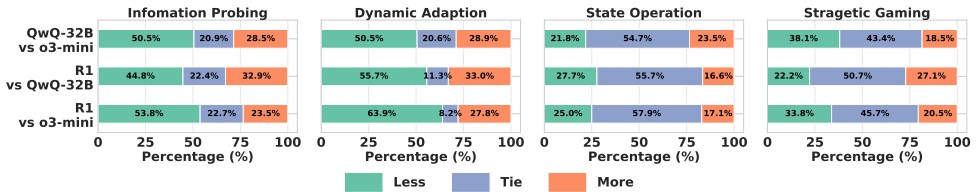

Figure 4: Efficiency comparison of interaction turns between models on correctly-answered problems. For each pair (A vs B), A is labeled as **Less** if it requires fewer turns than B, and **More** otherwise. A higher proportion of **Less** indicates superior efficiency in problem-solving. And Table 5 shows the specific average number of rounds of each model.

- **Reasoning vs. Non-Reasoning Models**: Overall, the accuracy improvement of non-reasoning models with increasing turns is significantly lower than that of reasoning models. This indirectly suggests that non-reasoning models are less effective in utilizing feedback in multi-turn dialogues.
- **Comparison among Reasoning Models**: We find that o3-mini does not have a clear advantage across arbitrary numbers of turns, especially when the number of reasoning turns is small (e.g., 5). However, as the number of turns increases, o3-mini demonstrates the most significant improvement in accuracy, particularly in IP. This further underscores o3-mini's strong abilities in leveraging and integrating historical interaction information over multiple turns.

### 4.4 EFFICIENCY ANALYSIS (RQ3)

To further analyze the relationship between performance and efficiency, we conduct an analysis of three reasoning models.[8] Specifically, we select a random sample of 100 problems that are correctly answered by all three models for each task type. We then compare the number of interaction turns required by each model pair to success, and calculate their efficiency scores defined in Section 2.

As shown in Figure 4.3, surprisingly, among the three models, o3-mini, which demonstrates the best performance, is relatively the least efficient, while R1 achieves the highest efficiency. This suggests that higher performance does not necessarily translate to better efficiency in terms of interaction turns. Combined with the conclusions in Section 4.2, the superior performance of o3-mini does not necessarily lie in its efficient reasoning. Instead, it may be more adept at long-term planning compared to others, making reasonable use of feedback in each turn to tackle more complex tasks.

### 4.5 INVALID OPERATION ANALYSIS (RQ4)

To better understand the poor performance of current LLMs on our benchmark, we conduct a manual review of model responses. Our analysis reveals that beyond limitations in long-term reasoning

---

[8]A more detailed analysis is provided in Appendix J.

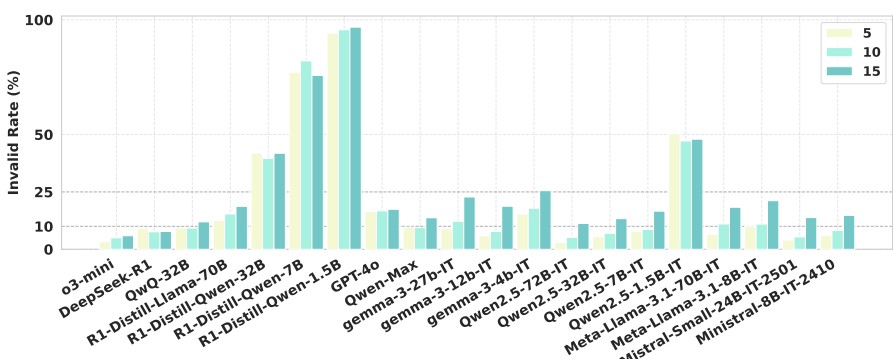

Figure 5: Invalid rate across evaluated models. Larger rate indicates weaker instruction-following and reasoning capabilities.

| Model | IP | | | | DA | | | | SA | | | | SG | | | |
|---|---|---|---|---|---|---|---|---|---|---|---|---|---|---|---|---|
| | Ass. | Ver. | Pla. | Fee. | Ass. | Ver. | Pla. | Fee. | Ass. | Ver. | Pla. | Fee. | Ass. | Ver. | Pla. | Fee. |
| QwQ-32B | 11.1 | 6.9 | 2.3 | 7.2 | 11.6 | 7.7 | 2.7 | 6.2 | 10.0 | 5.2 | 3.9 | 5.5 | 8.7 | 5.4 | 4.1 | 3.1 |
| Deepseek-R1 | 10.6 | 6.6 | 2.2 | 5.2 | 11.1 | 7.0 | 2.3 | 4.1 | 9.9 | 5.3 | 3.8 | 3.7 | 7.0 | 4.1 | 3.1 | 1.9 |
| R1-Distill-Qwen-32B | 7.6 | 2.7 | 2.7 | 3.0 | 8.7 | 3.3 | 3.8 | 3.0 | 8.2 | 2.9 | 3.5 | 4.3 | 8.1 | 2.8 | 4.0 | 2.9 |

Table 3: Pattern analysis on EvolArena. **Ass.**: Associate. **Ver.**: Verify. **Pla.**: Plan. **Fee.**: Feedback.

ability, a significant factor is the presence of "Invalid Operations" even in the best-performing models. These invalid operations fall into two categories: instruction-following failures where models fail to format queries according to format requirements, and operational failures where models cannot perform legitimate operations (e.g., making out-of-bounds moves in "KnightBattle"), which often requires basic reasoning capabilities. As shown in Figure 5, we can lead to the following conclusions:

- Overall, smaller models exhibit higher "Invalid Rate" (IR), particularly 1.5B-sized models which struggle with basic operation validity, reflecting their limited instruction-following capabilities.
- Surprisingly, distilled models show higher IR than their original versions, suggesting that while distillation may enhance reasoning, it potentially compromises stability in multi-turn interactions.
- Comparing state-of-the-art reasoning models with non-reasoning models, the former exhibit lower IR, further confirming the superior capabilities of reasoning models in multi-turn scenarios.

### 4.6 REASONING PATTERN ANALYSIS (RQ5)

To gain deeper insights into the reasoning capabilities of models on our benchmark, we conduct a reasoning pattern analysis on three open-source reasoning models. Specifically, using Qwen2.5-72B as the analyzer, we measure the average per-turn frequency of four reasoning patterns: original problem recall (**Associate**), error checking (**Verify**), strategic planning (**Plan**), and feedback analysis (**Feedback**). The results are summarized in Table 3, from which we draw the following conclusions:

- Stronger reasoning models QwQ-32B and R1 demonstrate superior capabilities in "Associate", "Verify", and "Feedback" compared to R1-Distill-32B, indicating these three abilities are crucial for multi-turn reasoning. Enhancement of these capabilities could potentially yield improvement.
- Although planning is essential for multi-turn tasks, the three models show similar planning frequencies across most tasks. However, SG exhibits notably higher planning frequency, suggesting that competitive scenarios inherently demand stronger strategic planning capabilities.

## 5 RELATED WORK

**Static Evaluation of Reasoning.** Early benchmarks for math (e.g., GSM8K (Cobbe et al., 2021), MATH (Hendrycks et al., 2021)) and code (e.g., HumanEval (Chen et al., 2021), MBPP (Austin et al., 2021)) rely on static, single-turn evaluation. However, these face severe data contamination and saturation risks. Studies like Performative Thinking (Palod et al., 2025) suggest that long-CoT traces in static tasks often reflect pattern matching rather than genuine reasoning. Even recent initiatives like CodeElo (Quan et al., 2025) and LiveCodeBench Pro (Zheng et al., 2025) remain focused on the

final product of single-turn generation, neglecting the dynamic correction and state tracking inherent in the reasoning process.

**Real-world Agent Benchmarks.** Benchmarks like AgentBench (Liu et al., 2024) and Agent-Board (Chang et al., 2024) evaluate task execution in complex, noisy environments (e.g., OS and Web). Unlike these application-focused benchmarks, EvolArena operates within closed, deterministic environments to isolate intrinsic logical capabilities—specifically induction, deduction, and planning—from tool usage or environmental noise. Furthermore, while AgentBoard (Chang et al., 2024) relies on costly human annotation, EvolArena achieves fully automated assessment via procedural generators.

**Benchmarks for Reasoning about Actions and Planning.** The AI planning community proposes PDDL-based benchmarks like TRAC (He et al., 2023), ACPBench (Kokel et al., 2025), and Action-ReasoningBench (Handa et al., 2025) to test formal understanding of actions. However, these are predominantly static question-answering tasks lacking long-horizon exploration. In contrast, EvolArena does not require mastering formal planning semantics. Instead, it compels models to operate in partially observable environments through dynamic multi-turn interaction, progressively uncovering information to construct solutions, which better mirrors general reasoning processes.

**Interactive and Game-based Benchmarks.** Existing interactive benchmarks have limitations: MT-Bench (Zheng et al., 2023) relies on subjective scoring, GameArena (Hu et al., 2025) is limited by scale, and SPIN-Bench (Yao et al., 2025) focuses on social multi-agent settings. Conversely, EvolArena targets single-agent logical reasoning against an environment. Critically, EvolArena features "Evolvability": driven by parametric generators capable of producing infinite instances, it addresses the data contamination and overfitting issues inherent in fixed datasets.

## 6 CONCLUSION

In this paper, we present EvolArena, an evolving arena for evaluating LLMs' multi-turn reasoning capabilities. The benchmark comprises 40 diverse tasks across four reasoning categories with adjustable difficulty levels, supported by an evolving evaluation framework. Our extensive experiments reveal both strengths and limitations of current LLMs in interactive reasoning, providing valuable insights for future research in LLM evaluation.

## REPRODUCIBILITY STATEMENT

To ensure full reproducibility of our results, we submit data and code in the supplementary material. This includes the dataset and the source code for our automated evaluation framework (Generator, Monitor, and Evaluator). The 40 tasks in EvolArena are constructed from publicly available seeds. 32 tasks originate from algorithmic problems on Codeforces (mean difficulty rating: 2453), and 8 are adapted from logic puzzles on the New York Times website. Each seed problem was manually transformed into a novel interactive task by designing specific interaction rules and standardized templates. Our publicly released code will include the generators developed for each task, which can deterministically produce the 3,600 evaluation instances used in this paper, as well as new instances with varying difficulty levels. Our evaluation environment is deterministic; for a given model and input, the interaction process and outcome are fixed. All experiments were conducted using the default inference parameters for each model (e.g., temperature=0.6, top-p=0.95 for R1) to ensure our results reflect the models' standard configurations. The complete experimental settings are provided in Appendix G.

## ETHICS STATEMENT

The research presented in this paper was conducted with a commitment to ethical standards and responsible scientific practice. All tasks are derived from publicly available data sources: algorithmic competition problems from Codeforces and logic puzzles from the New York Times website. No private, sensitive, or personally identifiable information was used in the construction of this benchmark. The adaptation process focused on transforming the logic of these public problems into novel, interactive formats. The primary goal of this work is to advance the scientific understanding of the

reasoning capabilities of LLMs in multi-turn, interactive scenarios. EvolArena is intended to serve as a diagnostic tool for researchers and developers to identify strengths and weaknesses in AI reasoning, thereby fostering progress in the field.

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

## A  STATEMENT ON THE USE OF LLMS

LLMs are the primary subject of the research presented in this paper. Our work is focused on the development of a benchmark (EvolArena) to evaluate the multi-turn reasoning capabilities of various LLMs. While LLMs are the object of our study, we clarify that they were not used as a significant tool for research ideation or for the writing of this manuscript. The authors of this paper take full responsibility for the contents of this work, including all claims made.

## B  MULTI-TURN REASONING FORMULATION

Let $f_\theta$ denote a LLM engaged in interactive reasoning. The model generates a sequence of queries $\{q_i\}_{i=1}^n$ through iterative interaction turns. At each turn $i$, the model's query generation process can be formulated as:

$$q_i = f_\theta(\mathcal{C}_i) = f_\theta(p, \mathcal{H}_{i-1}) \tag{1}$$

where $\mathcal{C}_i$ represents the complete context at turn $i$, $p$ is the initial problem specification, $\mathcal{H}_{i-1} = \{(q_j, m_j)\}_{j=1}^{i-1}$ denotes the interaction history, $q_j$ and $m_j$ are previous queries and their corresponding feedback.

This formulation captures how the model leverages both the original problem and accumulated evidence from previous interactions to inform its next query decision.

## C  EXTENDED DISCUSSION ON EVOLVING NATURE

In this section, we elaborate on the future potential of the "Evolving" mechanism within EvolArena. While our current experiments demonstrate procedural extensibility and parameterized difficulty scaling, we emphasize that the architecture of EvolArena—specifically the parameterized **Generator** and the deterministic **Monitor**—serves as the necessary infrastructure to realize advanced forms of adaptability. We detail this potential across three progressive levels:

### C.1  AUTOMATED CURRICULUM LEARNING (FROM ASSESSMENT TO TRAINING)

This represents the most direct future application of the "Evolving" nature: transforming EvolArena from an *examination venue* into a *gymnasium* for RL training.

- **Problem:** One of the major challenges in training reasoning agents is reward sparsity. If tasks are too difficult, the model rarely succeeds and learns nothing; if tasks are too easy, the learning is inefficient.
- **Solution:** Our framework addresses this by providing the three essential components of an RL environment:
  1. **Environment:** 40 diverse tasks provide a rich training ground.
  2. **Reward:** The Monitor provides immediate, deterministic feedback (Success, Failure, Invalid), serving as a perfect reward signal.
  3. **Curriculum:** The Generator provides a tunable knob for difficulty (parameter $n$) that can be adjusted smoothly.
- **Implementation:** An external "Curriculum Controller" can be constructed to observe the model's win rate at current difficulty $n$. If the win rate $> 90\%$, the controller calls the Generator to increase difficulty to $n + 1$; if $< 10\%$, it decreases to $n - 1$. This ensures the model always trains within its zone of proximal development, maximizing training efficiency and the upper bound of reasoning capabilities.

### C.2  HIGH-RESOLUTION ADAPTIVE EVALUATION

Beyond training, the concept of adapting to the agent's capabilities is equally critical for evaluation.

- **Problem:** Static, one-size-fits-all benchmarks often suffer from low resolution. They struggle to differentiate subtle differences between two strong models (e.g., o3-mini vs. R1) that both solve "Hard" tasks, or distinguish between two weak models that both fail.
- **Solution:** Our evolving architecture supports model-contingent evaluation.
- **Implementation:** Instead of testing on fixed levels (e.g., $n = 6, 9, 12$), the Evaluator can dynamically adjust $n$:
  - For a smaller model (e.g., 7B), the Evaluator starts at $n = 4$ and incrementally increases difficulty until identifying the model's capability inflection point (e.g., failure at $n = 7$).
  - For a strong model (e.g., o3-mini), the Evaluator starts at $n = 12$ and evolves upward to $n = 13, 14, 15...$, probing its true capability ceiling.
- **Value:** This yields a high-resolution capability score (e.g., o3-mini achieves capability level $n = 15$ on Task X), which is crucial for precisely measuring incremental model progress.

### C.3 ADVERSARIAL & STRATEGIC EVOLUTION

This represents the frontier of the "Evolving" concept: the evolution of the *benchmark itself*.

- **Problem:** Models may overfit or game a benchmark by learning specific task heuristics rather than general reasoning abilities.
- **Solution:** Our Generator is controlled not only by the difficulty parameter $n$ but also by problem parameters $g_n$ (e.g., specific configurations).
- **Implementation:** We envision an "Adversarial Generator" that analyzes the failure logs of a specific model (e.g., o3-mini) to identify specific strategic blind spots (e.g., consistent failure in specific opening configurations of *Knight Battle*). The Generator then evolves to specifically produce more of these instances that effectively counter the model's current strategy.
- **Value:** This facilitates a true co-evolutionary paradigm: as models evolve stronger capabilities, the benchmark evolves more challenging problems. This allows EvolArena to continuously expose the frontier defects of SOTA models.

In conclusion, while our current work demonstrates the initial stage of "Evolving" capabilities (i.e., procedural generation and parameterized scaling), the Generator-Monitor architecture constitutes the core innovation. It not only solves the saturation crisis of current static benchmarks but, more importantly, provides the viable technical foundation for true evolution—encompassing automated curriculum learning, adaptive evaluation, and adversarial evolution. We have expanded the definition of "Evolving" in this work from simple parameterized scaling to serving as the infrastructure for adaptive assessment, training, and adversarial evolution.

## D THE INHERENT NECESSITY OF MULTI-TURN INTERACTION OF OUR TASKS

A foundational design principle of EvolArena is that all 40 tasks mechanically enforce multi-turn interaction and cannot be successfully completed in a single turn. We contend that a core component of advanced reasoning involves a LLM's ability to continuously interact with an environment to gather information, verify hypotheses, and dynamically adjust its strategy. Our benchmark is specifically engineered to evaluate this fundamental capability.

The design across all tasks is centered on an essential probe-observe-deduce loop, where a model must first execute an exploratory action, then process the environment's feedback, and only then can it deduce the underlying rules or state required for effective planning. This principle makes multi-turn engagement an inescapable necessity for success. This design philosophy is consistently applied across our four task categories, as detailed below and verifiable in the task prompts in Appendix N.

### D.1 INFORMATION PROBING AND DYNAMIC ADAPTATION

For tasks within these categories, the possibility of a single-turn solution is statistically infinitesimal. The core mechanic is built upon an iterative feedback loop where the model must make a series of

queries to incrementally narrow down the solution space. A prime example is the "Word Guessing" task; in the easy mode, the probability of correctly guessing a four-letter word in one attempt is approximately $(1/26)^4 \approx 0.000002188$. Success is therefore contingent on the model's ability to process feedback over multiple turns—such as "Correct letter in correct position" or "Correct letter but in wrong position"—to logically deduce the answer. The low accuracy scores achieved by most models further validate this design, as only those with strong iterative multi-turn reasoning capabilities, like o3-mini, demonstrate an ability to improve their chances of success.

## D.2 STATE OPERATION

The design philosophy for all tasks in this category is centered on incomplete information, making multi-turn interaction a prerequisite for understanding the environment. In tasks like "Maze Navigation," the system's rules are deliberately obscured; for instance, the model is not informed if directional controls like "up/down" and "left/right" are swapped. The model is thus forced to engage in an exploratory phase over several turns, experimenting with actions and observing outcomes to deduce the full set of hidden mechanics before a successful path can be planned and executed. This requirement for empirical discovery through interaction is a consistent feature across all tasks in this category.

## D.3 STRATEGIC GAMING

In our strategic gaming scenarios, the task Generator is programmatically designed to ensure that a one-move victory is impossible for either side. This guarantees that a strategic, multi-turn engagement unfolds from the start. For example, in the "Knight Battle" task, the initial board positions for the player's White Knight and the system's Black Knight are algorithmically set to prevent a capture or a target-reaching move on the first turn. This forces the model to engage in a sustained exchange, requiring it to plan several steps ahead while anticipating and reacting to the opponent's moves over multiple rounds.

# E RAW DATA STATISTICS AND UTILIZATION

The initial seeds for the 40 tasks in EvolArena were sourced from two public websites.

- **Codeforces:** 32 tasks originate from algorithmic competition problems on Codeforces. These problems have official difficulty ratings ranging from 1700 to 3500, with a mean rating of 2453.13. This range signifies a high degree of difficulty, presenting a significant challenge even for expert human programmers and ensuring the rigorous nature of our benchmark.

- **New York Times:** The remaining 8 tasks are adapted from popular logic puzzles published on the New York Times website.

It is crucial to note that we did not use these seed problems in their original, static form. Instead, each seed was manually and meticulously adapted into a novel, interactive task requiring multi-turn engagement. This comprehensive adaptation process involved three key steps:

1. **Designing Interaction Rules:** We deliberately designed a new set of interaction rules for each original problem to transform it into a dynamic task that necessitates multi-turn interaction for its solution.

2. **Creating Question Templates:** We manually created standardized question templates for every task. These templates include a clear description of the interaction rules, strict input/output format requirements, and illustrative examples of the interaction flow.

3. **Developing Generators:** Based on these structured templates, we developed corresponding generators. These generators are capable of automatically producing numerous instances of each task at varying difficulty levels, all of which can be evaluated by our automated framework.

This structured process clarifies how we utilized existing data sources to construct the novel, interactive challenges within EvolArena.

## F    DISCUSSION ON THE UPPER LIMIT OF ROUNDS

**Evaluating Reasoning Efficiency.**    Setting an upper limit on interaction turns is a core element of our evaluation philosophy, not merely a consideration of cost. We believe that efficient reasoning is a key marker of advanced intelligence. Many real-world scenarios require problem-solving that is not only correct but also completed within a finite number of steps. Therefore, by setting a cap, EvolArena evaluates a model's ability to solve problems efficiently under resource constraints, compelling it to seek more concise and direct reasoning paths rather than engaging in endless trial and error.

**Empirical Justification for the 15-Turn Cap.**    Regarding the sensitivity to the specific 15-turn limit, our experimental results provide strong support. The analysis presented in Figure 3 of our paper shows that for many tasks, performance gains tend to plateau around the 10-turn mark. This suggests that the 15-turn limit provides sufficient exploratory space for models in most cases. Furthermore, we observe a practical engineering constraint: beyond 15 turns, the accumulated conversation history often causes models to exceed their maximum context length, which can lead to truncated outputs that compromise the validity of the evaluation.

### F.1    PRACTICAL CONSIDERATIONS AND TRADE-OFFS.

Finally, we acknowledge that this cap is also influenced by practical computational costs and represents a trade-off between evaluating efficiency and exploring the absolute limits of performance. This limit may pose a challenge for "slow-thinking" models that require longer reasoning chains to arrive at a solution.

## G    DETAILED EXPERIMENTAL SETTINGS

### G.1    DATASET AND SAMPLE SIZE

The performance metrics reported in Table 2 represent the average performance over 300 distinct samples for each of the four task categories: Information Probing, Dynamic Adaptation, State Operation, and Strategic Gaming. This sample set consists of 10 unique tasks within each category, where each task comprises 30 distinct problem instances (10 tasks $\times$ 30 questions/task $=$ 300 samples). This scale provides a statistically robust foundation for our performance analysis.

### G.2    EVALUATION SETTING AND RATIONALE

Our benchmark is intentionally designed for a zero-shot interactive setting, with the crucial clarification that each task prompt includes a built-in, one-shot demonstration. As illustrated in Figure 2, the "Example Interactions" section within each prompt provides an in-context example of a successful dialogue. This example effectively serves as a single "shot" to guide the model on the required interaction format and rules.

We deliberately opted against a traditional few-shot evaluation for two primary reasons stemming from the multi-turn nature of our benchmark:

- **Context Length Limitations:** In multi-turn tasks, the accumulated conversation history occupies a significant portion of the context window. Adding multiple, complete dialogue examples for a few-shot setup would risk exceeding the context length limits of many models, making a fair and practical evaluation challenging.

- **Multi-Turn Evaluation Paradigm:** Unlike static, single-turn tasks, multi-turn interactive benchmarks like MT-Bench typically focus more on a model's performance in a dynamic, continuous dialogue rather than employing traditional few-shot configurations.

Therefore, our "zero-shot with a built-in demonstration" approach is a deliberate design choice tailored to the unique challenges of evaluating multi-turn reasoning.

### G.3 INFERENCE PARAMETERS

For all experiments, we utilize the default inference parameters for each model as recommended upon their public release. This approach ensures a fair and representative evaluation that aligns with best practices. For instance, we evaluate the R1 model using a temperature of 0.6 and a top-p value of 0.95.

### G.4 EVALUATION METRICS AND STABILITY

We report pass@1 for our main results. Given that our evaluation environment is deterministic, the interaction process and outcome are fixed for any given model and input, which makes pass@1 a direct and reliable metric. To investigate potential performance variance while managing computational costs, we conduct supplementary pass@16 experiments on the R1 model across four representative tasks. The results, presented in Table 4, demonstrate minimal performance variance, which reinforces the stability and reliability of our evaluation framework.

| Category | IP (%) | | | DA (%) | | | SO (%) | | | SG (%) | | |
|---|---|---|---|---|---|---|---|---|---|---|---|---|
| Difficulty | E | M | H | E | M | H | E | M | H | E | M | H |
| Std. Dev. | 0.25 | 0.81 | 0.31 | 0.35 | 0.97 | 1.04 | 0.26 | 0.59 | 0.60 | 0.24 | 0.67 | 0.55 |

Table 4: Performance variance (standard deviation in %) for R1 on pass@16 experiments across four task categories (E: Easy, M: Medium, H: Hard).

## H    LIMITATIONS AND FUTURE WORK

Our work provides a robust framework for evaluating multi-turn reasoning; however, it is essential to acknowledge its limitations and outline directions for future research.

**Closed-World Design and External Validity.**    EvolArena operates as a closed-world system with highly structured environments and deterministic rules. This design contrasts sharply with ambiguous, open-world problems that are characterized by incomplete information. We made this trade-off to achieve full automation, objectivity, and reproducibility in our evaluations. Consequently, strong performance on EvolArena indicates proficiency in structured reasoning but is not a direct measure of a model's ability to generalize to unstructured, real-world applications. Performance should be viewed as a necessary, but not sufficient, condition for general reasoning ability.

**Risk of Overfitting and Responsible Interpretation.**    There is a risk that models could achieve high scores by "gaming" the benchmark—learning techniques specific to its tasks rather than developing general-purpose reasoning skills. While the diversity of 40 tasks across four categories mitigates this risk by requiring a broad set of skills, the fundamental possibility remains. Therefore, EvolArena should be used as a diagnostic tool rather than a definitive measure of general intelligence. Optimizing solely for this benchmark may create excellent puzzle solvers instead of true general reasoners.

**Interaction Modality.**    Another limitation is the structured, non-natural language interaction format of EvolArena. This design was a deliberate choice to isolate and measure a model's core logical reasoning capabilities, separate from the complexities of natural language processing. The trade-off is that our benchmark currently cannot assess a model's ability to reason within a natural language dialogue, which is a crucial skill for many real-world applications.

**Future Work.**    To address these limitations, our future work will focus on bridging the gap between our benchmark and real-world complexity. Key directions include: (1) Extending the framework to support and evaluate reasoning within natural language interactions. (2) Introducing more complex adversarial strategies to further challenge the models. (3) Utilizing EvolArena as a reinforcement learning environment to train more powerful and generalizable reasoning agents.

## I  BROADER IMPACT

Our work on evaluating LLMs' strategic reasoning through interactive tasks has implications beyond just testing model capabilities. The evaluation framework provides an engaging and intuitive way to understand how language models approach complex decision-making tasks. This could help bridge the gap between technical AI research and public understanding, as this work offers a familiar context for demonstrating both the capabilities and limitations of current AI systems. Additionally, the insights gained from observing how models handle strategic planning and adaptation in interactive environments could inform the development of more effective AI assistants for everyday problem-solving tasks. We believe our approach of using structured tasks for evaluation could inspire similar frameworks in other domains where step-by-step reasoning and strategic thinking are important.

## J  EFFICIENCY ANALYSIS

We evaluate model efficiency from two distinct perspectives: strategic efficiency, which measures the number of interactions required to find a solution, and computational efficiency, which measures the token cost of those interactions.

### J.1  NUMBER OF INTERACTION TURNS

Our initial analysis used pairwise comparison win rates to intuitively demonstrate direct competition between top models on identical problems. However, a more direct metric for strategic efficiency is the average number of turns a model takes to correctly solve a problem. We present these statistics in Table 5. These results align with the conclusions in body text: although o3-mini demonstrates the strongest overall performance in terms of accuracy, it typically requires more turns to arrive at a solution, making it the least strategically efficient among the top models.

| Model | IP | | | DA | | | SO | | | SG | | | AVG |
|---|---|---|---|---|---|---|---|---|---|---|---|---|---|
| | E | M | H | E | M | H | E | M | H | E | M | H | |
| o3-mini | 8.25 | 10.18 | 8.64 | 10.45 | 9.35 | 9.41 | 7.03 | 9.41 | 9.62 | 3.52 | 5.80 | 8.21 | 8.97 |
| R1 | 5.38 | 6.84 | 6.25 | 5.29 | 5.60 | 6.26 | 5.43 | 5.59 | 7.13 | 4.13 | 6.05 | 8.13 | 6.94 |
| QwQ-32B | 7.57 | 5.77 | 5.79 | 7.50 | 7.22 | 6.55 | 4.25 | 3.29 | 3.49 | 3.29 | 5.70 | 7.64 | 5.87 |

Table 5: Average number of interaction turns on correctly solved problems.

### J.2  TOKEN CONSUMPTION

To provide a more complete picture, we also analyze the computational efficiency by measuring the average token consumption. Table 6 shows a comparison between R1 and QwQ-32B. The data indicates that R1 is not only more strategically efficient (fewer turns) but is also significantly more computationally efficient (lower token consumption) than QwQ-32B in most categories. This dual-dimensional analysis provides a more comprehensive and nuanced view of model efficiency, reinforcing body text's conclusions.

| Category | A >= B (%) | A <= B (%) | A = B (%) |
|---|---|---|---|
| IP | 45.13 | 54.87 | 0.00 |
| DA | 35.05 | 64.95 | 0.00 |
| SO | 62.13 | 37.87 | 0.00 |
| SG | 31.13 | 68.87 | 0.00 |

Table 6: Token Consumption Comparison: R1 vs. QwQ-32B. A represents R1, and B represents QwQ-32B.

## K  HUMAN PERFORMANCE BASELINE

Providing a human baseline is crucial for calibrating the difficulty of our benchmark. To offer this perspective, we clarify that the majority of our seed tasks originate from the competitive programming platform Codeforces. The problems we selected have established human difficulty ratings on this platform, with a mean rating of 2453, a minimum of 1700, and a maximum of 3500. On Codeforces, a rating of approximately 2400 corresponds to the "Master" tier, indicating that these tasks are designed to be challenging even for highly skilled human experts. Therefore, these ratings serve as a strong proxy for expert human performance and confirm that EvolArena is calibrated to assess reasoning on tasks of significant difficulty.

## L  IMPLEMENTATION DETAILS OF EVOLARENA

To ensure full technical transparency and reproducibility, we provide the detailed algorithmic implementation of our automated framework. This section covers the main interaction loop and the specific logic for the Generator, Monitor, and Evaluator across representative tasks from each of the four reasoning categories.

### L.1  MAIN EVALUATION LOOP

The core of EvolArena is an automated pipeline that manages the interaction between the Large Language Model (LLM) and the environment. Algorithm 1 outlines the process implemented in our evaluation script.

---
**Algorithm 1** EvolArena Main Evaluation Loop
---
**Require:** Model $M$, Task Template $T$, Difficulty Parameter $n$, Max Rounds $K$
 1: **Initialization:**
 2: $(p, s) \leftarrow \text{Generator}(T, n)$                  ▷ Generate problem instance $p$ and reasoning objective $s$
 3: $H \leftarrow []$                                              ▷ Initialize conversation history
 4: $State \leftarrow \text{InitialState}(p)$
 5: $Round \leftarrow 1$
 6: **while** $Round \leq K$ **and** $\neg\text{IsTerminated}(State)$ **do**
 7:     $Prompt \leftarrow \text{ConstructPrompt}(p, H)$
 8:     $Response \leftarrow M(Prompt)$                         ▷ Get model output
 9:     $Query \leftarrow \text{Parse}(Response)$
10:     $(Feedback, State) \leftarrow \text{Monitor}(T, Query, State, s)$        ▷ Update state & get feedback
11:     $H.\text{append}(\text{User} : Query, \text{System} : Feedback)$
12:     $Round \leftarrow Round + 1$
13: **end while**
14: $Result \leftarrow \text{Evaluator}(H, s)$                  ▷ Compute Accuracy, Efficiency, etc.
15: **return** $Result$
---

### L.2  TASK-SPECIFIC IMPLEMENTATION DETAILS

We provide the detailed implementation logic for the Generator, Monitor, and Evaluator across representative tasks. These components ensure that the generated problems are solvable, the interactions are deterministic, and the evaluations are rigorous.

#### L.2.1  INFORMATION PROBING: FIND THE IMPOSTORS

**Generator:**

| Model | IP | | | DA | | | SO | | | SG | | | AVG | | |
|---|---|---|---|---|---|---|---|---|---|---|---|---|---|---|---|
| | E | M | H | E | M | H | E | M | H | E | M | H | E | M | H |
| *Reasoning Model* | | | | | | | | | | | | | | | |
| o3-mini | 60.33 | 41.56 | 28.22 | 40.33 | 24.18 | 17.13 | 38.61 | 27.00 | 20.22 | 85.00 | 74.44 | 59.17 | 56.07 | 41.80 | 31.19 |
| | (±3.06) | (±3.22) | (±2.94) | (±3.24) | (±2.80) | (±2.46) | (±3.17) | (±2.90) | (±2.63) | (±3.71) | (±4.41) | (±4.82) | (±3.30) | (±3.33) | (±3.21) |
| R1 | 39.22 | 25.00 | 11.11 | 34.58 | **23.11** | **15.22** | 47.67 | 38.56 | 32.78 | 73.00 | 62.67 | 57.67 | 48.62 | **37.33** | 29.19 |
| | (±3.12) | (±2.83) | (±2.05) | (±3.13) | (±2.76) | (±2.35) | (±3.26) | (±3.18) | (±3.07) | (±5.03) | (±5.48) | (±5.60) | (±3.64) | (±3.56) | (±3.27) |
| QwQ-32B | **53.56** | **28.22** | **19.00** | **38.33** | 20.44 | 12.00 | 36.67 | **29.89** | 25.33 | 70.00 | 56.33 | 46.00 | **49.64** | 33.72 | 25.58 |
| | (±2.21) | (±1.50) | (±1.13) | (±2.05) | (±1.25) | (±1.09) | (±2.16) | (±1.57) | (±1.69) | (±4.09) | (±3.14) | (±2.77) | (±2.63) | (±1.87) | (±1.67) |
| R1-Distill-Llama-70B | 33.78 | 13.11 | 6.33 | 25.50 | 11.00 | 5.67 | 15.56 | 10.78 | 7.89 | 61.11 | 44.17 | 28.89 | 33.99 | 19.76 | 12.19 |
| | (±3.09) | (±2.21) | (±1.59) | (±2.89) | (±2.05) | (±1.51) | (±2.37) | (±2.03) | (±1.76) | (±5.04) | (±5.14) | (±4.69) | (±3.35) | (±2.86) | (±2.39) |
| R1-Distill-Qwen-32B | 26.78 | 10.11 | 3.22 | 10.50 | 3.22 | 1.67 | 7.11 | 4.22 | 3.11 | 39.44 | 24.44 | 15.28 | 20.96 | 10.50 | 5.82 |
| | (±2.89) | (±1.97) | (±1.15) | (±2.04) | (±1.15) | (±0.84) | (±1.68) | (±1.31) | (±1.13) | (±5.06) | (±4.45) | (±3.72) | (±2.92) | (±2.22) | (±1.71) |
| R1-Distill-Qwen-7B | 3.89 | 2.33 | 1.11 | 0.44 | 0.00 | 0.00 | 0.67 | 1.11 | 0.22 | 3.67 | 2.67 | 1.00 | 2.17 | 1.53 | 0.58 |
| | (±1.26) | (±0.99) | (±0.69) | (±0.44) | (±0.00) | (±0.00) | (±0.53) | (±0.69) | (±0.31) | (±2.13) | (±1.83) | (±1.13) | (±1.09) | (±0.88) | (±0.53) |
| R1-Distill-Qwen-1.5B | 0.67 | 0.78 | 0.33 | 0.00 | 1.00 | 0.11 | 0.00 | 0.00 | 0.00 | 0.67 | 0.67 | 0.00 | 0.33 | 0.61 | 0.11 |
| | (±0.50) | (±0.57) | (±0.38) | (±0.00) | (±0.65) | (±0.22) | (±0.00) | (±0.00) | (±0.00) | (±0.92) | (±0.92) | (±0.00) | (±0.36) | (±0.54) | (±0.15) |
| *Non-Reasoning Model* | | | | | | | | | | | | | | | |
| GPT-4o | 29.11 | 10.56 | 6.89 | 22.92 | 11.56 | **7.00** | 19.73 | **15.11** | **11.56** | 42.22 | 30.56 | **22.78** | 28.50 | 16.94 | 12.06 |
| | (±2.90) | (±2.01) | (±1.66) | (±2.78) | (±2.09) | (±1.67) | (±2.60) | (±2.34) | (±2.09) | (±5.11) | (±4.77) | (±4.34) | (±3.35) | (±2.80) | (±2.44) |
| Qwen-Max | 33.89 | 11.56 | 7.33 | 27.42 | 17.67 | 8.11 | **20.15** | 13.67 | 10.78 | 49.17 | 33.61 | 22.50 | 32.66 | 19.13 | 12.18 |
| | (±3.01) | (±2.09) | (±1.70) | (±2.95) | (±2.49) | (±1.78) | (±2.62) | (±2.25) | (±2.03) | (±5.17) | (±4.89) | (±4.32) | (±3.44) | (±2.93) | (±2.46) |
| gemma-3-27b-IT | 31.00 | 9.78 | 9.67 | 18.92 | 9.67 | 6.33 | 16.00 | 10.00 | 5.67 | 16.89 | 4.72 | 5.15 | 20.70 | 8.54 | 6.70 |
| | (±3.02) | (±1.94) | (±1.93) | (±2.59) | (±1.93) | (±1.59) | (±2.40) | (±1.96) | (±1.51) | (±3.77) | (±2.19) | (±2.49) | (±2.95) | (±2.01) | (±1.88) |
| gemma-3-12b-IT | 24.78 | 8.33 | 4.56 | 15.03 | 8.44 | 5.89 | 12.22 | 4.56 | 3.56 | 12.61 | 9.17 | 5.17 | 16.16 | 7.63 | 4.79 |
| | (±2.82) | (±1.81) | (±1.36) | (±2.36) | (±1.82) | (±1.54) | (±2.14) | (±1.36) | (±1.21) | (±3.48) | (±3.04) | (±2.62) | (±2.70) | (±2.01) | (±1.68) |
| gemma-3-4b-IT | 11.44 | 4.56 | 2.44 | 8.61 | 6.00 | 4.11 | 9.00 | 4.22 | 2.89 | 10.67 | 2.33 | 0.67 | 9.93 | 4.28 | 2.53 |
| | (±2.08) | (±1.36) | (±1.01) | (±1.86) | (±1.55) | (±1.30) | (±1.87) | (±1.31) | (±1.09) | (±3.50) | (±1.71) | (±0.92) | (±2.33) | (±1.48) | (±1.08) |
| Qwen2.5-72B-IT | **38.22** | **20.00** | 10.89 | 23.22 | 12.44 | 6.33 | 14.78 | 11.00 | 7.89 | 41.50 | **32.78** | 26.67 | 29.43 | **19.06** | **12.94** |
| | (±3.18) | (±2.61) | (±2.04) | (±2.78) | (±2.16) | (±1.59) | (±2.32) | (±2.05) | (±1.76) | (±4.85) | (±4.65) | (±4.41) | (±3.28) | (±2.87) | (±2.45) |
| Qwen2.5-32B-IT | 33.44 | 14.67 | **12.44** | 19.69 | **12.89** | 6.22 | 23.67 | 17.67 | 14.44 | 42.00 | 25.00 | 19.76 | **29.70** | 17.56 | 13.22 |
| | (±3.08) | (±2.31) | (±2.16) | (±2.63) | (±2.19) | (±1.58) | (±2.78) | (±2.49) | (±2.30) | (±4.85) | (±4.91) | (±4.20) | (±3.34) | (±2.98) | (±2.56) |
| Qwen2.5-7B-IT | 27.44 | 11.44 | 3.67 | 18.33 | 9.33 | 6.22 | 9.67 | 6.00 | 4.89 | 22.67 | 10.00 | 8.33 | 19.53 | 9.19 | 5.78 |
| | (±2.92) | (±2.08) | (±1.23) | (±2.58) | (±1.90) | (±1.58) | (±1.93) | (±1.55) | (±1.41) | (±4.75) | (±3.40) | (±3.13) | (±3.05) | (±2.23) | (±1.84) |
| Qwen2.5-1.5B-IT | 2.22 | 0.11 | 0.22 | 6.44 | 4.33 | 0.78 | 9.44 | 0.89 | 1.33 | 17.67 | 14.67 | 12.00 | 8.94 | 5.00 | 3.58 |
| | (±0.96) | (±0.22) | (±0.31) | (±1.64) | (±1.33) | (±0.57) | (±1.91) | (±0.61) | (±0.75) | (±4.32) | (±4.01) | (±3.68) | (±2.21) | (±1.54) | (±1.33) |
| Llama-3.1-70B-IT | 40.11 | 21.22 | 11.89 | **23.81** | 12.00 | 6.78 | 16.78 | 11.44 | 8.78 | 36.50 | 25.33 | 20.72 | 29.30 | 17.50 | 12.04 |
| | (±3.20) | (±2.67) | (±2.12) | (±2.82) | (±2.12) | (±1.64) | (±2.44) | (±2.08) | (±1.85) | (±4.76) | (±4.36) | (±4.14) | (±3.31) | (±2.81) | (±2.44) |
| Llama-3.1-8B-IT | 22.67 | 10.00 | 4.89 | 13.58 | 5.78 | 4.67 | 12.56 | 5.33 | 3.78 | 11.00 | 5.67 | 3.00 | 14.95 | 6.69 | 4.08 |
| | (±2.74) | (±1.96) | (±1.41) | (±2.28) | (±1.53) | (±1.38) | (±2.17) | (±1.47) | (±1.25) | (±3.55) | (±2.62) | (±1.93) | (±2.69) | (±1.90) | (±1.49) |
| Mistral-Small-24B-IT-2501 | 18.67 | 7.78 | 4.56 | 17.92 | 6.22 | 5.00 | 19.56 | 10.00 | 6.78 | 25.56 | 12.83 | 12.28 | 20.42 | 9.21 | 7.15 |
| | (±2.55) | (±1.75) | (±1.36) | (±2.53) | (±1.58) | (±1.42) | (±2.59) | (±1.96) | (±1.64) | (±4.38) | (±3.53) | (±3.57) | (±3.01) | (±2.21) | (±2.00) |
| Ministral-8B-IT-2410 | 8.89 | 4.22 | 2.00 | 13.69 | 5.67 | 5.11 | 16.67 | 11.56 | 4.33 | 21.33 | 5.33 | 8.67 | 15.15 | 6.69 | 5.03 |
| | (±1.86) | (±1.31) | (±0.92) | (±2.28) | (±1.51) | (±1.44) | (±2.44) | (±2.09) | (±1.33) | (±4.64) | (±2.55) | (±3.19) | (±2.81) | (±1.87) | (±1.72) |
| **AVG** | 27.01 | 12.39 | 12.77 | 18.96 | 10.25 | 6.22 | 17.32 | 11.65 | 8.81 | 34.13 | 23.87 | 18.78 | 24.36 | 14.63 | 10.34 |
| | (±2.63) | (±1.87) | (±1.50) | (±2.37) | (±1.73) | (±1.38) | (±2.23) | (±1.82) | (±1.55) | (±4.21) | (±3.55) | (±3.15) | (±2.86) | (±2.24) | (±1.90) |

Table 7: Model Accuracy with 95% confidence intervals on EvolArena.

---

**Algorithm 2** Generator for Find the Impostors

---

**Require:** Total players $N$, Existing Answers Set $\mathcal{D}$
1: **loop**
2:     $A \leftarrow$ RandomBinaryString($N$)                             ▷ 0: Impostor, 1: Crewmate
3:     $Zeros \leftarrow$ Count($A$, '0')
4:                                              ▷ Constraint: Impostors between $N/3$ and $2N/3$
5:     **if** $N/3 \leq Zeros \leq 2N/3$ **and** $A \notin \mathcal{D}$ **then**
6:         $\mathcal{D}$.add($A$)
7:         **return** $A$
8:     **end if**
9: **end loop**

**Monitor:**

---

**Algorithm 3** Monitor for Find the Impostors

---

**Require:** User Input $I$, Hidden Sequence $A$
1: **Regex (Query):** r"My Query:\s*(\d+),(\d+),(\d+)"
2: **Regex (Answer):** r"My Answer:\s*((?:\d+,)*\d+)"
3: **if** $I$ matches Query format with indices $P = \{p_1, p_2, p_3\}$ **then**
4:     $ImpostorCount \leftarrow \sum_{p \in P}(1 \text{ if } A[p] == \text{'0' else } 0)$
5:     **if** $ImpostorCount > 3 - ImpostorCount$ **then**
6:         **return** "0"                               ▷ Majority are impostors
7:     **else**
8:         **return** "1"                               ▷ Majority are crewmates
9:     **end if**
10: **else if** $I$ matches Answer format with indices $G$ **then**
11:     $PredictedA \leftarrow$ IndicesToBinary$(G)$
12:     **if** $PredictedA == A$ **then**
13:         **return** "1"
14:     **else**
15:         **return** "0"
16:     **end if**
17: **else**
18:     **return** "Invalid", "-1"
19: **end if**

---

**Evaluator:**

---

**Algorithm 4** Evaluator for Find the Impostors

---

**Require:** Interaction History $H$, Ground Truth Sequence $A$
1: **Initialize Metrics:**
2: $Success \leftarrow$ False
3: $TurnCount \leftarrow$ Length$(H)$
4: $InvalidCount \leftarrow 0$
5: $Patterns \leftarrow \{\text{Associate} : 0, \text{Verify} : 0, \text{Plan} : 0, \text{Feedback} : 0\}$
6: **for** each turn $t$ in $H$ **do**
7:     $Feedback \leftarrow H[t].\text{SystemOutput}$
8:     $Thought \leftarrow H[t].\text{ModelThought}$
9:                                 ▷ 1. Metric: Invalid Rate (Instruction Following)
10:     **if** $Feedback == $ "-1" $\vee Feedback == $ "Invalid Format" **then**
11:         $InvalidCount \leftarrow InvalidCount + 1$
12:     **end if**
13:                            ▷ 2. Metric: Pattern Analysis (Cognitive Process)
14:     $Patterns \leftarrow Patterns +$ LLM_Pattern_Analyzer$(Thought)$
15:                                 ▷ 3. Metric: Accuracy (Final Outcome)
16:     **if** $t == TurnCount$ **then**
17:         $Query \leftarrow H[t].\text{UserQuery}$
18:         **if** $Query$ starts with "My Answer:" **then**
19:             $SubmittedIndices \leftarrow$ ParseAnswer$(Query)$
20:             $TrueIndices \leftarrow$ GetIndicesOfZeros$(A)$
21:             **if** $SubmittedIndices == TrueIndices$ **then**
22:                 $Success \leftarrow$ True
23:             **end if**
24:         **end if**
25:     **end if**
26: **end for**
27: $InvalidRate \leftarrow InvalidCount/TurnCount$
28: **return** $\{Success, TurnCount, InvalidRate, Patterns\}$

---

### L.2.2   DYNAMIC ADAPTATION: PASSWORD BREAKER

**Generator:**

**Algorithm 5** Generator for Password Breaker

**Require:** Base $k$, Group Index $i$
1: $Min \leftarrow i \times 10 + 1$
2: $Max \leftarrow Min + 9$
3: $P_{curr} \leftarrow \text{RandomInteger}(Min, Max)$
4: **return** $P_{curr}, Min, Max$

**Monitor:**

**Algorithm 6** Monitor for Password Breaker

**Require:** Input $I$, Password $P$, Base $k$, Range $[Min, Max]$
1: **Regex:** r"My Guess:\s*(\d+)"
2: **if** $I$ matches Regex with guess $G$ **then**
3:     **if** $G < Min \vee G > Max$ **then**
4:         **return** "Invalid"
5:     **end if**
6:     **if** $G == P$ **then**
7:         **return** "Correct"
8:     **else**
9:         $D_P \leftarrow \text{ToBaseK}(P, k)$
10:        $D_G \leftarrow \text{ToBaseK}(G, k)$
11:       $D_{new} \leftarrow []$
12:       **for** $j \leftarrow 0$ **to** $\max(\text{len}(D_P), \text{len}(D_G))$ **do**
13:          $digit \leftarrow (D_P[j] + D_G[j]) \pmod{k}$
14:          $D_{new}.\text{append}(digit)$
15:       **end for**
16:       $Val \leftarrow \text{FromBaseK}(D_{new}, k)$
17:       $P \leftarrow (Val \pmod{(Max - Min + 1)}) + Min$       ▷ Update Hidden State
18:       **return** "Incorrect"
19:     **end if**
20: **else**
21:     **return** "Invalid"
22: **end if**

**Evaluator:**

---

**Algorithm 7** Evaluator for Password Breaker

---

**Require:** Interaction History $H$
1: **Initialize Metrics:**
2: $Success \leftarrow$ False
3: $SolvedAtTurn \leftarrow$ None
4: $InvalidCount \leftarrow 0$
5: $Patterns \leftarrow \{\text{Assoc} : 0, \text{Ver} : 0, \text{Plan} : 0, \text{Feed} : 0\}$
6: **for** $t \leftarrow 1$ **to** Length($H$) **do**
7:     $Feedback \leftarrow H[t].\text{SystemOutput}$
8:                                            ▷ Check for Invalid Rate
9:     **if** $Feedback ==$ "Invalid" **then**
10:         $InvalidCount \leftarrow InvalidCount + 1$
11:     **end if**
12:                                            ▷ Run Pattern Analysis
13:     $Patterns \leftarrow Patterns + \text{LLM\_Pattern\_Analyzer}(H[t].\text{Thought})$
14:                               ▷ Check for Success (Can happen at any turn)
15:     **if** $Feedback ==$ "Correct" **then**
16:         $Success \leftarrow$ True
17:         $SolvedAtTurn \leftarrow t$
18:         **break**                ▷ Stop counting turns after success for Efficiency
19:     **end if**
20: **end for**
21: $Efficiency \leftarrow SolvedAtTurn$ if $Success$ else Length($H$)
22: $InvalidRate \leftarrow InvalidCount/\text{Length}(H)$
23: **return** $\{Success, Efficiency, InvalidRate, Patterns\}$

---

### L.2.3   STATE OPERATION: MAZE NAVIGATION

**Generator:**

---

**Algorithm 8** Generator for Maze Navigation

---

**Require:** Grid Size $N \times M$
1: $Grid \leftarrow \text{Initialize}(N, M, \text{'.'}), Start \leftarrow (0, 0)$
2: **loop**
3:     $F \leftarrow \text{RandomPos}(N, M)$
4:     **if** $F \neq Start$ **then** $Grid[F] \leftarrow$ 'F'; **break**
5:     **end if**
6: **end loop**
7: **for** $k \leftarrow 1$ **to** $N \times M//3$ **do**
8:     $P \leftarrow \text{RandomPos}(N, M)$
9:     **if** $Grid[P] ==$ '.' **then**
10:         $Grid[P] \leftarrow$ '*'; $Valid \leftarrow \text{DFS\_CheckPath}(Start, F, Grid)$
11:         **if** $\neg Valid$ **then** $Grid[P] \leftarrow$ '.'
12:         **end if**
13:     **end if**
14: **end for**
15: $S_{UD}, S_{LR} \leftarrow \text{RandomBool}(), \text{RandomBool}()$
16: **return** $(Grid, S_{UD}, S_{LR})$

---

**Monitor:**

---

**Algorithm 9** Monitor for Maze Navigation

**Require:** User Input $I$, Current Pos $P$, Grid $G$, Swap Flags $S_{LR}, S_{UD}$
1: **Regex:** r"My Move:\s*([UDLR])"
2: **if** $I$ matches Regex with direction $D$ **then**
3:                                                          ▷ Apply Control Swaps
4:     **if** $S_{LR}$ **and** $D \in \{L, R\}$ **then**
5:         $D \leftarrow \text{Flip}(D)$
6:     **end if**
7:     **if** $S_{UD}$ **and** $D \in \{U, D\}$ **then**
8:         $D \leftarrow \text{Flip}(D)$
9:     **end if**
10:    $P_{new} \leftarrow P + \text{Delta}(D)$
11:                                                 ▷ Check Boundaries
12:    **if** $\neg\text{InGrid}(P_{new})$ **then**
13:        $P_{new} \leftarrow P$
14:    **end if**
15:    $Cell \leftarrow G[P_{new}]$
16:    **if** $Cell ==' *'$ **then**
17:        **return** "My Move: $D$", "-1 -1 You lose!"
18:    **else if** $Cell ==' F'$ **then**
19:        **return** "My Move: $D$", "$P_{new}.x$ $P_{new}.y$ You win!"
20:    **else**
21:        $P \leftarrow P_{new}$                                    ▷ Update Agent Position
22:        **return** "My Move: $D$", "$P_{new}.x$ $P_{new}.y$"
23:    **end if**
24: **else**
25:    **return** "Invalid", "Invalid format"
26: **end if**

**Evaluator:**

**Algorithm 10** Evaluator for Maze Navigation

**Require:** Interaction History $H$, Max Turns $K$
1: **Initialize Metrics:**
2: $Success \leftarrow$ False
3: $InvalidCount \leftarrow 0$
4: $Patterns \leftarrow$ InitializeCounts()
5: **for** each turn $t$ in $H$ **do**
6:    $Feedback \leftarrow H[t].\text{SystemOutput}$
7:                                   ▷ 1. Invalid Rate: Capture both Format and Logic Errors
8:    **if** $Feedback$ contains "Invalid" **then**
9:        $InvalidCount \leftarrow InvalidCount + 1$               ▷ Format Error
10:    **else if** $Feedback ==$ "-1 -1 You lose!" **then**
11:                                  ▷ Operational Error (Hit Obstacle)
12:        $InvalidCount \leftarrow InvalidCount + 1$
13:        $Success \leftarrow$ False
14:        **break**
15:    **end if**
16:                                       ▷ 2. Pattern Analysis
17:    $Patterns \leftarrow Patterns + \text{LLM\_Pattern\_Analyzer}(H[t].\text{Thought})$
18:                                         ▷ 3. Check Success
19:    **if** $Feedback$ contains "You win!" **then**
20:        $Success \leftarrow$ True
21:        **break**
22:    **end if**
23: **end for**
24: **return** $\{Success, \text{Turns Used}, InvalidRate, Patterns\}$

### L.2.4   STRATEGIC GAMING: KNIGHT BATTLE

**Generator:**

**Algorithm 11** Generator for Knight Battle

**Require:** Board Size $N$
1: $T_W \leftarrow (N/2, N/2); T_B \leftarrow (N/2 + 1, N/2)$
2: **loop**
3:     $P_W, P_B \leftarrow$ RandomPos$(N)$, RandomPos$(N)$
4:                                       ▷ Constraint: Distinct positions, not on targets
5:     **if** $P_W \neq P_B$ **and** $P_W \notin \{T_W, T_B\}$ **and** $P_B \notin \{T_W, T_B\}$ **then**
6:         **break**
7:     **end if**
8: **end loop**
9: **return** $(P_W, P_B, T_W, T_B)$

**Monitor:**

**Algorithm 12** Monitor for Knight Battle

**Require:** User Input $I$, Board $B$, Positions $Pos_W, Pos_B$, Targets $T_W, T_B$
1: **Regex:** `r"My Move:\s*(\d+)\s+(\d+)"`
2: **if** $I$ matches Regex with new white pos $P'_W$ **then**
3:     **if** $\neg$IsValidKnightMove$(Pos_W, P'_W)$ **then**
4:         **return** "Invalid", "Invalid knight move"
5:     **end if**
6:     $Pos_W \leftarrow P'_W$
7:                                   ▷ Check White Win Conditions
8:     **if** $Pos_W == Pos_B$ **then**
9:         **return** "Move: $P'_W$", "White wins!"                       ▷ Capture
10:     **else if** $Pos_W == T_W$ **and** $\neg$UnderAttack$(Pos_W, Pos_B)$ **then**
11:         **return** "Move: $P'_W$", "White wins!"          ▷ Target Reached
12:     **end if**
13:                                  ▷ System (Black) Turn
14:     $Moves \leftarrow$ GetValidLShapes$(Pos_B)$
15:     **if** $Moves$ is empty **then**
16:         **return** "Move: $P'_W$", "White wins!"
17:     **end if**
18:     $Pos_B \leftarrow$ RandomChoice$(Moves)$
19:                                ▷ Check Black Win Conditions
20:     **if** $Pos_B == Pos_W$ **then**
21:         **return** "Move: $P'_W$", "Black wins!"
22:     **else if** $Pos_B == T_B$ **and** $\neg$UnderAttack$(Pos_B, Pos_W)$ **then**
23:         **return** "Move: $P'_W$", "Black wins!"
24:     **end if**
25:     **return** "Move: $P'_W$", "$Pos_B.x\ Pos_B.y$"
26: **else**
27:     **return** "Invalid", "Invalid format"
28: **end if**

**Evaluator:**

**Algorithm 13** Evaluator for Knight Battle

---

**Require:** Interaction History $H$
 1: **Initialize Metrics:**
 2: $Outcome \leftarrow$ "Loss"
 3: $InvalidCount \leftarrow 0$
 4: $Patterns \leftarrow$ InitializeCounts()
 5: **for** each turn $t$ in $H$ **do**
 6:     $Feedback \leftarrow H[t].$SystemOutput
 7:                                      ▷ 1. Invalid Rate: Logic Constraint Violation
 8:     **if** $Feedback ==$ "Invalid move" **then**
 9:         $InvalidCount \leftarrow InvalidCount + 1$
10:         $Outcome \leftarrow$ "Loss (Invalid)"
11:         **break**
12:     **end if**
13:                                        ▷ 2. Pattern Analysis (Focus on Planning)
14:     $Patterns \leftarrow Patterns +$ LLM_Pattern_Analyzer($H[t].$Thought)
15:                                      ▷ 3. Check Win/Loss Conditions
16:     **if** $Feedback ==$ "White wins!" **then**
17:         $Outcome \leftarrow$ "Win"
18:         **break**
19:     **else if** $Feedback ==$ "Black wins!" **then**
20:         $Outcome \leftarrow$ "Loss (Captured)"
21:         **break**
22:     **end if**
23: **end for**
24: $Success \leftarrow (Outcome ==$ "Win")
25: **return** $\{Success,$ Turns Used, $InvalidRate, Patterns\}$

---

## M   TAXONOMY OF REASONING FAILURE MODES

To provide diagnostic insights beyond the quantitative "Invalid Rate," we conduct a manual inspection of 50 randomly sampled failure instances. Based on this analysis, we identify five distinct categories of core reasoning failures. We formally introduce this taxonomy to better understand the cognitive limitations of current models:

**State Tracking Collapse**   In dynamic tasks, models often fail to maintain and update a coherent environmental state across multiple turns. For instance, in *Dynamic Adaptation (DA)* tasks such as *Password Breaker*, even after the Monitor returns "Incorrect" (signaling that the password has changed via XOR rules), the model frequently continues to reason based on the outdated password state from previous turns. This failure to update the internal belief state causes the entire subsequent reasoning chain to derail.

**Hasty Generalization**   This failure mode is prevalent in tasks that require an "explore-then-exploit" strategy. Models often prematurely lock onto an incorrect global hypothesis before gathering sufficient evidence to support the conclusion. For example, in *State Operation (SO)* tasks like *Maze Navigation*, a model might verify only the U/D control swap and erroneously assume the L/R controls are normal without testing. Subsequent planning based on this unverified assumption leads to inevitable failure.

**Greedy & Myopic Planning**   This is commonly observed in *Strategic Gaming (SG)* tasks. Models tend to select a "local optimum" for the current turn while ignoring that this move leads to a "global worst-case" scenario in the near future. In *Knight Battle*, for instance, a model might choose a move to capture a piece or check the opponent, failing to foresee that this specific position exposes it to an unavoidable counter-attack or checkmate in the subsequent turns.

**Inefficient Exploration**   This represents a strategic failure where models fail to employ optimal search strategies (e.g., binary search) to maximize information gain within the limited horizon (e.g., 15 turns). In *Information Probing (IP)* tasks like *Find the Impostors*, failing models often perform redundant or overlapping queries (e.g., querying $\{1, 2, 3\}$ followed immediately by $\{1, 2, 4\}$) rather than querying disjoint sets (e.g., $\{4, 5, 6\}$) to rapidly narrow down the possibility space.

**Logical Constraint Violation**    This category transcends the simple formatting errors captured by the "Invalid Rate." Here, the model maintains correct syntax but violates the core logical constraints of the task. For example, in *Strategic Gaming (SG)* (e.g., *Knight Battle*), a model might output "My Move: 9 9". While syntactically correct, this move is logically illegal on an $8 \times 8$ chessboard. This indicates a defect in fundamental reasoning capabilities, such as the understanding of spatial boundaries, rather than a failure in instruction following.

# N    TASK INTRODUCTION

We classify problems into four types based on their characteristics and testing capabilities: Information Probing (IP), Dynamic Adaptation (DA), State Operation (SO), and Strategic Gaming (SG). Each type contains 10 tasks that described in detail below.

## N.1    INFORMATION PROBING

**FindTheImpostors**    In this task, models need to identify all impostors among $n$ players through strategic queries about groups of three players. Models can make queries to compare impostors and crewmates in specified groups, ultimately determining the complete set of impostors.

---

**Case N.1: FindTheImpostors Problem Template**

Let's play Find the Impostors! Your task is to identify all impostors among $n$ players.
Rules:
1. There are $n$ players
2. Some players are impostors $(k)$ and others are crewmates $(n - k)$
3. The number of impostors $k$ is between $1/3n$ and $2/3n$
Query Types:
1. Ask about three players:
Format: "My Query: $a, b, c$" (three different player numbers)
Response will be:
- 0: if there are more impostors than crewmates among these three
- 1: if there are more crewmates or equal numbers
- -1: if query is invalid
2. Submit final answer:
Format: "My Answer: $x_1, x_2, ..., x_k$"
($k$ is number of impostors, followed by their indices)
Response will be:
- 0 if incorrect
- 1 if correct
Example interaction:
You: "My Query: 1,2,3"
Me: "0" (means more impostors in this group)
You: "My Query: 3,4,5"
Me: "1" (means more crewmates in this group)
You: "My Answer: 1,2,3,4"
Me: 1 (if correct)
Instructions:
1. You must explain your reasoning before each query
2. Format your responses exactly as shown above
Remember:
- Player numbers must be between 1 and $n$
- All three numbers in a query must be different
Ready to start? Make your first query!

---

**Case N.2: FindTheImpostors Difficulty Levels**

Easy: $n = 6$, Medium: $n = 9$, Hard: $n = 12$

---

**GuessMax**  In this task, models need to discover a hidden password by querying maximum values from specific positions in an array. The password consists of maximum values from complementary position sets defined by given exclusion rules.

---

**Case N.3: GuessMax Problem Template**

Let's play Guess The Maximums!
Rules:
1. Hidden array $A[1...50]$ contains numbers from 1 to 50
2. You need to guess $n$ numbers forming the password
3. For password position $i$, you are given $S_i$ = subset of positions to exclude
4. Password$[i]$ = max value among all positions EXCEPT those in $S_i$
Your subsets are:
{subset desc}
Password Example:
For $x = 4$, $n = 2$, if:
$S_1 = \{1, 3\}$, $S_2 = \{2, 4\}$
And hidden array $A = [3, 1, 2, 4]$
Then:
- Password$[1]$ ignores positions 1, 3 ($S_1$)
So looks at $A[2] = 1$, $A[4] = 4$
Password$[1] = 4$
- Password$[2]$ ignores positions 2, 4 ($S_2$)
So looks at $A[1] = 3$, $A[3] = 2$
Password$[2] = 3$
Therefore, the answer is "4 3".
Query Types:
1. Make a query:
Format: "My Query: $x_1$ $x_2$ ... $x_m$"
where:
- $x_i$ = positions you want to query ($1 \leq m < 50$)
- You'll receive the maximum value at these positions
2. Submit final answer:
Format: "My Answer: $p_1$ $p_2$ ... $p_n$"
where:
- $p_i$ = your guess for each password slot
- You'll receive "Correct" or "Incorrect"
Simple Example Interaction:
Given: $x = 4$, $n = 2$, $S_1 = \{1, 3\}$, $S_2 = \{2, 4\}$, $A = [3, 1, 2, 4]$(hidden), Answer = $[4, 3]$(hidden)
You: "My Query: 2 4"
Me: "4"
You: "My Query: 1 3"
Me: "3"
You: "My Answer: 4 3"
Me: "Correct"
Instructions:
1. Make queries based on previous results
2. Use exactly the formats shown above
3. Explain your reasoning before each query
Remember:
- Each query reveals maximum value at specified positions
- Password digits come from complementary position sets
- Think carefully about which positions to query
Ready to start? Make your first query!

---

**Case N.4: GuessMax Difficulty Levels**

Easy: $n = 7$, Medium: $n = 10$, Hard: $n = 16$

**CircleFinding**   In this task,models need to discover a hidden circle's parameters (center coordinates and radius) through ray-shooting queries from the origin. Models can make queries in the format "My Query: $x_q$ $y_q$" to shoot a ray through any specified point, receiving the minimum distance from the ray to the circle (0.0 if the ray intersects). Through geometric reasoning and strategic ray placement, models should determine the circle's exact position and size, submitting their answer in the format "My Answer: $x_c$ $y_c$ $r_c$".

---

**Case N.5: CircleFinding Problem Template**

Let's play Circle Finding Game! Your task is to discover a hidden circle on a plane through ray-shooting queries.
Rules:
1. There is a hidden circle with center $(x_c, y_c)$ and radius $r_c$
2. All parameters are integers and $|x_c|, |y_c|, |r_c| \leq \{n\}$
3. The radius $r_c$ satisfies: $1 \leq r_c \leq \sqrt{x_c^2 + y_c^2} - 1$
4. You can shoot rays from origin $(0, 0)$ through any point $(x_q, y_q)$ you specify
Query Types:
1. To shoot a ray:
Format: "My Query: $(x_q, y_q)$ "
where:
- $x_q, y_q$ are integers with $|x_q|, |y_q| \leq \{n\}$
- At least one of $x_q$ or $y_q$ must be non-zero
Example: "My Query: 0 -10"
You'll receive the minimum distance from the ray to the circle
(0.0 if the ray intersects the circle)
2. To submit final answer:
Format: "My Answer: $x_c$ $y_c$ $r_c$"
where $x_c, y_c, r_c$ are the circle's parameters
Example: "My Answer: 20 10 10"
You'll receive the correctness of your answer.
Instructions:
1. Make queries based on previous results
2. Use exactly the formats shown above
3. Explain your reasoning before each query
4. All distances are precise to $10^{-10}$
Remember:
- Circle parameters are integers
- Rays start from origin $(0, 0)$
- Think carefully about ray directions
- Use geometric properties to deduce circle location
- Distance is 0 when ray intersects circle
Ready to start? Make your first query!

---

**Case N.6: CircleFinding Difficulty Levels**

Easy: $n = 200$, Medium: $n = 1000$, Hard: $n = 1500$

---

**BitCompare**   In this task, models need to find two positions in a hidden permutation of numbers that will yield the maximum XOR value when combined. Models can make queries in the format "My Query: $a$ $b$ $c$ $d$" to compare the bitwise OR results of different positions, receiving "<", "=" or ">" as responses, and ultimately submit their answer in the format "My Answer: $i$ $j$". Through strategic querying, models should determine which two positions will produce the largest XOR value.

---

**Case N.7: BitCompare Problem Template**

Let's play Bitwise Comparison Game! Your task is to find two positions in a hidden permutation that maximize their XOR value.
Rules:
1. There is a hidden permutation of $\{n\}$ numbers (0 to $\{n-1\}$)

2. Each position contains a unique number from 0 to $\{n-1\}$
3. You can make comparison queries between OR operations:
- Each query compares $(a \mid b)$ with $(c \mid d)$
- $\mid$ denotes bitwise OR operation
- You'll receive "<", "=", or ">" as response
Query Types:
1. To make a comparison query:
Format: "My Query: $a\ b\ c\ d$"
where:
- $a, b, c, d$ are positions in array (0-based indexing)
Example: "My Query: 0 2 3 1"
Response will be one of: "<", "=", ">"
2. To submit final answer:
Format: "My Answer: $i\ j$"
where $i$ and $j$ are the positions with maximum XOR value
Example: "My Answer: 3 2"
Instructions:
1. Make queries based on previous comparisons
2. Use exactly the formats shown above
3. Explain your reasoning before each query
Remember:
- All positions contain unique numbers from 0 to $\{n-1\}$
- Position indices start from 0
- Think carefully about which positions to compare
- Use your queries wisely to find maximum XOR pair
Ready to start? Make your first query!

---

**Case N.8: BitCompare Difficulty Levels**

Easy: $n = 5$, Medium: $n = 7$, Hard: $n = 9$

---

**TreeDiscovery**   In this task, models need to discover the structure of a hidden tree through strategic path queries. For each query, models specify two disjoint vertex sets and a target vertex, receiving the number of paths between vertices from these sets that pass through the target vertex.

---

**Case N.9: TreeDiscovery Problem Template**

Let's play Legendary Tree! Your task is to discover the structure of a hidden tree through strategic queries.
Rules:
1. There is a hidden tree with $n$ vertices (numbered 1 to $n$)
2. You can ask questions to discover the tree's structure
3. For each question, you need to specify:
- Set $S$: A group of vertices (at least one vertex)
- Set $T$: Another group of vertices (at least one vertex)
- Vertex $v$: Any vertex you choose
Note: $S$ and $T$ must not have any common vertices
Query Types:
1. To make a query:
Format: "My Query: $S \mid T \mid v$" where:
- $S$ is your first set of vertices (space-separated numbers)
- $T$ is your second set of vertices (space-separated numbers)
- $v$ is the vertex you want to check
Example: "My Query: 1 2 | 3 | 2"
Response:
You will receive the number of vertex pairs $(s, t)$ where:
- $s$ is from set $S$
- $t$ is from set $T$

- The path from $s$ to $t$ passes through vertex $v$
2. To submit final answer:
Format: "My Answer: $edge_1$ $edge_2$ ..." where each edge is "u-v"
Example: "My Answer: 1-2 2-3"
Example Interaction:
You: "My Query: 1 2 | 3 | 2"
Me: "2" (meaning 2 paths through vertex 2)
Instructions:
1. Use queries to gather information about the tree
2. Format your queries exactly as shown above
3. Think carefully about which vertices to select
Remember:
- Sets $S$ and $T$ must be non-empty and disjoint
- Use your queries wisely to gather maximum information
- Each edge in final answer should appear exactly once
Ready to start? Make your first query!

---

### Case N.10: TreeDiscovery Difficulty Levels

Easy: $n = 5$, Medium: $n = 6$, Hard: $n = 7$

---

**LinkedListQuery**    In this task, models need to find a specific value in a sorted linked list through strategic queries. Models can query values and next pointers at specific positions to explore the list structure and determine the target value.

---

### Case N.11: LinkedListQuery Problem Template

Let's play Linked List Query Game! Your task is to find a specific value in a sorted linked list through queries.
Rules:
1. There is a hidden sorted linked list with $n$ elements
2. Each element has a value and a next pointer (pointing to the next element's index)
3. You can make two types of queries:
- VALUE query: you will get both value and next pointer at position $i$
- ANSWER submission: you will get a feedback of "Correct" or "Incorrect"
Query Types:
1. To make a value query:
Format: "My Query: $i$"
where:
- $i$ is the position in list (1-based indexing)
Example: "My Query: 1"
2. To submit final answer:
Format: "My Answer: $ans$"
where $ans$ is either:
- The minimum value in the list
Example: "My Answer: 80"
Example Interaction:
List length = $n$, start = 3, $x = 80$
You: "My Query: 1"
Me: "value=97, next=-1"
You: "My Query: 3"
Me: "value=16, next=2"
You: "My Answer: 80"
Me: "Correct"
Instructions:
1. Make queries to explore the linked list
2. Use exactly the formats shown above
3. Explain your reasoning before each query/answer

Remember:
- Following next pointers, values are in increasing order
- You need to find minimum value of the list
- Position indices start from 1
- Think carefully about which positions to query
Ready to start? Make your first query!

## Case N.12: LinkedListQuery Difficulty Levels

Easy: $n = 5$, Medium: $n = 9$, Hard: $n = 11$

**MedianQuery**  In this task, models need to find specific positions in a hidden permutation through queries about subsequence medians. For each query, models specify positions to examine and receive the two middle values, ultimately locating target values in the permutation.

## Case N.13: MedianQuery Problem Template

Let's play Median Query Game! Your task is to find specific positions in a hidden permutation through median queries.
Rules:
1. There is a hidden permutation $p$ of length $n$ (numbers 1 to $n$)
2. You can make queries about subsequences of even length
3. Each query returns the two middle values (medians) of your chosen subsequence
4. Your goal is to find positions of values $\{n//2\}$ and $\{n//2 + 1\}$
Query Types:
1. To make a query:
Format: "My Query: $k$ $x_1$ $x_2$ ... $x_k$"
where:
- $k$ is the length of subsequence (even number, $4 \leq k \leq n$)
- $x_1$ to $x_k$ are distinct positions (1-based indexing)
Example: "My Query: $n$ 1 2 3 4 5 6"
Response will be two numbers: the $k/2$-th and $(k/2 + 1)$-th smallest values in the subsequence
2. To submit final answer:
Format: "My Answer: $i$ $j$"
where $i$ and $j$ are positions of values $\{n//2\}$ and $\{n//2 + 1\}$
Example: "My Answer: 3 6"
Instructions:
1. Make queries based on previous results
2. Use exactly the formats shown above
3. Explain your reasoning before each query
Remember:
- The permutation contains numbers 1 to $n$ exactly once
- Position indices start from 1
- Think carefully about which subsequences to query
- Use your queries wisely to locate the target positions
- Order of positions in final answer doesn't matter
Ready to start? Make your first query!

## Case N.14: MedianQuery Difficulty Levels

Easy: $n = 6$, Medium: $n = 8$, Hard: $n = 15$

**MinMax**  In this task, models need to find positions of minimum and maximum elements in a hidden array through pairwise comparison queries. Each query reveals the relative ordering of two elements, helping deduce the extreme values' locations.

**Case N.15: MinMax Problem Template**

Let's play Find Min Max! Your task is to find the minimum and maximum elements in a hidden array.
Rules:
1. You are given an array of length $n$, but you cannot see its elements
2. You can only compare two elements by their positions ($i$ and $j$)
3. After each comparison, you'll receive one of these responses:
- "$<$": element at position $i$ is less than element at position $j$
- "$=$": element at position $i$ equals element at position $j$
- "$>$": element at position $i$ is greater than element at position $j$
Example:
If we have an array of length 3:
- Query "1 2" would get:
"$>$" (means element at position 1 is greater than element at position 2)
- Query "2 3" would get:
"$<$" (means element at position 2 is less than element at position 3)
Query Types:
1. Ask about comparison:
Format: "My Query: $i$ $j$" ($i$ and $j$ are positions to compare)
Response will be "$<$", "$=$" or "$>$"
2. Submit final answer:
Format: "My Answer: ! $i$ $j$" (where $i$ is minimum position, $j$ is maximum position)
Response will be:
- 1 if correct
- 0 if incorrect
Instructions:
1. You must explain your reasoning before each query
2. Format your responses exactly as shown above
3. You can only compare two different positions at a time
Remember:
- Positions must be between 1 and 6
- Choose comparisons wisely to minimize queries
Ready to start? Make your first query!

**Case N.16: MinMax Difficulty Levels**

Easy: $n = 5$, Medium: $n = 6$, Hard: $n = 7$

**WordGuessing**  In this task, models need to discover a hidden $n$-letter word through strategic guesses. Each guess receives feedback indicating correct letters, misplaced letters, and wrong letters, helping narrow down the target word.

**Case N.17: WordGuessing Problem Template**

Let's play Letters Finding! Your task is to guess a $n$-letter English word.
Rules:
1. You must provide exactly ONE $n$-letter English word as your guess
2. After each guess, you'll receive feedback using these symbols:
- R: Correct letter in the correct position
- G: Correct letter but in the wrong position
- W: Wrong letter, not in the word
Example:
If the target word is ABCDUVWZGHIJ
- Guess ACEFOPQMKLLM would get: RGWWWWWWWWWW
(A is correct position, C is correct but wrong position, rest are wrong)
Query Type:
1. Make a guess:
Format: "My Guess: [YOUR $n$-LETTER WORD]"

Response will be:
- A $n$-character string using R, G, and W
- R: right letter, right position
- G: right letter, wrong position
- W: wrong letter
Instructions:
1. Make your guess based on previous feedback (if any)
2. Guess only one word at a time
3. Give your reasoning process before each guess
Remember:
- Each guess must be exactly $n$ letters long
- The same letter can appear multiple times
- Guesses need not be real English words
- Use feedback wisely to deduce the target word
Ready to start? Make your first query!

---

**Case N.18: WordGuessing Difficulty Levels**

Easy: $n = 4$, Medium: $n = 8$, Hard: $n = 12$

---

**BitQuery**   In this task, models need to discover a hidden array by making queries about pairs of positions using bitwise operations (AND, OR, XOR). Models can make queries in the format "My Query: OPERATION $i$ $j$" to get the result of applying the specified bitwise operation on elements at positions $i$ and $j$. After gathering enough information through strategic queries, models should submit their final answer in the format "My Answer: $a_1$ $a_2$ ... $a_n$" representing their guess of the entire hidden array.

---

**Case N.19: BitQuery Problem Template**

Let's play Bitwise Query Game! Your task is to discover the hidden array through bitwise operations.
Rules:
1. There is a hidden array of $\{n\}$ integers
2. Each element in the array is between 0 and $\{n - 1\}$ inclusive
3. You can ask three types of queries about any two positions $i$ and $j$:
- AND query: returns the bitwise AND of elements at positions $i$ and $j$
- OR query: returns the bitwise OR of elements at positions $i$ and $j$
- XOR query: returns the bitwise XOR of elements at positions $i$ and $j$
Query Types:
1. To make a query:
Format: "My Query: OPERATION $i$ $j$"
where:
- OPERATION is one of: AND, OR, XOR
- $i$ and $j$ are positions in array (1-based indexing)
Example: "My Query: OR 1 2"
2. To submit final answer:
Format: "My Answer: $a_1$ $a_2$ ... $a_{\{n\}}$"
where $a_1$ to $a_{\{n\}}$ are your guessed array elements
Example: "My Answer: 0 0 2 3"
Example Interaction:
Array length = $\{n\}$
You: "My Query: OR 1 2"
Me: "0" (result of OR operation)
You: "My Query: OR 2 3"
Me: "2" (result of OR operation)
You: "My Query: XOR 2 4"
Me: "3" (result of XOR operation)
You: "My Answer: 0 0 2 3"

Instructions:
1. Make queries based on previous results
2. Use exactly the formats shown above
3. Explain your reasoning before each query
Remember:
- All array elements are between $0$ and $\{n-1\}$
- Position indices start from 1
- Think carefully about which operations to use
- Use your queries wisely to gather maximum information
Ready to start? Make your first query!

### Case N.20: BitQuery Difficulty Levels

Easy: $n = 4$, Medium: $n = 8$, Hard: $n = 12$

## N.2 DYNAMIC ADAPTATION

**PasswordBreaking** In this task, models need to discover a hidden password through strategic guesses. After each incorrect guess, the password changes according to a base-k XOR operation, requiring careful analysis of the transformation mechanics.

### Case N.21: PasswordBreaking Problem Template

Let's play Password Breaker! Your task is to hack into the RPD database by guessing the correct password.
Rules:
1. The password is always between MIN_VALUE = $m$ and MAX_VALUE = $m + n$ (inclusive)
2. After each guess, you'll receive one of these responses:
- Correct: Correct password, you've successfully broken in!
- Incorrect: Wrong password, and the system has changed the password
- Invalid: Invalid guess
Important Mechanics:
- The system uses base-$\{k\}$ operations ($k = \{k\}$)
- When you guess wrong $(y)$, if the current password was $x$:
* First convert both $x$ and $y$ to base-$\{k\}$ numbers
* Perform digit-by-digit base-$\{k\}$ XOR:
For each digit position $i$: result$[i] = (x[i] + y[i])$ mod $\{k\}$
* Convert result back to decimal to get $z$
* Map $z$ to range $[0, n]$ by taking mod $(n + 1)$
* Add $m$ to get the new password between $[m, m + n]$
Example:
With $k = 2$, if $x = 6$ (base-2: $[1, 1, 0]$) and $y = 5$ (base-2: $[1, 0, 1]$):
1. XOR digits: $[1, 1, 0]$ XOR $[1, 0, 1] = [(1 + 1)\mathrm{mod}2, (1 + 0)\mathrm{mod}2, (0 + 1)\mathrm{mod}2] = [0, 1, 1]$
2. Convert $[0, 1, 1]$ to decimal: $z = 3$
3. Map to range: $z = (3 \bmod (n + 1)) + m$
Example Interaction:
- Original password = 5
- You: "My Guess: 3"
- Me: "Incorrect" (wrong, password changes by XOR mechanism)
- You: "My Guess: 5"
- Me: "Incorrect" (wrong, password changes by XOR mechanism)
- You: "My Guess: 8"
- Me: "Correct" (correct!)
Query Type:
1. Make a guess:
Format: "My Guess: $X$"
where $X$ is a number between $\{min\_value\}$ and $\{max\_value\}$
Instructions:

1. Make your guess based on previous responses
2. Format your response exactly as shown above
3. Give your reasoning before making each guess
Remember:
- Always guess within valid range $[m, \{max\_value\}]$
- Password changes after each incorrect guess
- Think carefully about the base-$\{k\}$ XOR mechanism
Ready to start? Make your first query!

---

### Case N.22: PasswordBreaking Difficulty Levels

Easy: $n = 10$, Medium: $n = 20$, Hard: $n = 30$

---

**RotaryLaserLock**   In this task, models need to discover the relative positions of metal arcs on concentric rings through strategic rotations. Each query rotates a ring and reveals the count of unblocked laser paths passing through all rings.

---

### Case N.23: RotaryLaserLock Problem Template

Let's play the Rotary Laser Lock Game! Your task is to discover the final relative positions of metal arcs after your rotations.
Rules:
1. Lock Structure:
- $\{n\}$ concentric rings numbered 0 to $\{n-1\}$
- Each ring has $\{n*m\}$ sections (0 to $\{n*m-1\}$)
- Each section can be empty or contain metal
- Rings can rotate independently
2. Metal Arcs:
- Each ring has one metal arc
- Each arc covers exactly 6 consecutive sections
- Arcs are solid and cannot be broken
3. Rotation Mechanics:
- You can rotate any ring
- Clockwise rotation: +1 section
- Anticlockwise rotation: -1 section
- Ring 0 is your reference ring
4. Laser Detection:
- $\{n*m\}$ lasers emit from center
- One laser per section
- Metal arcs block lasers
- Display shows count of unblocked lasers
Query Types:
1. Make a rotation:
Format: "My Query: $x\ d$"
where:
- $x$: ring number (0 to $\{n-1\}$)
- $d$: direction (-1 or +1)
Example: "My Query: 2 1" rotates ring 2 clockwise
2. Submit final answer:
Format: "My Answer: $p_1\ p_2\ ...\ p_n$"
where:
- Each $p_i$ is final position of ring $i$ relative to ring 0
- Positions range from 0 to $\{n*m-1\}$
Example Round:
Initial state unknown, $\{n*m\}$ sections per ring
You: "My Query: 1 1"
- Rotating ring 1 clockwise
Me: "10"

- 10 lasers pass through
You: "My Query: 2 -1"
- Rotating ring 2 anticlockwise
Me: "12"
- 12 lasers pass through
You: "My Answer: 3 1 12 11"
- Final positions relative to ring 0
Me: "Correct"
Instructions:
1. Make rotations based on previous results
2. Use exactly the formats shown above
3. Explain your reasoning before each query
Remember:
- Each arc is exactly 6 sections long
- Track your rotations carefully
- All positions are mod $\{n * m\}$
- Invalid query/answer = immediate loss
Ready to start? Make your first query!

---

**Case N.24: RotaryLaserLock Difficulty Levels**

Easy: $n = 3$, Medium: $n = 4$, Hard: $n = 5$

---

**AttendanceCheck**  In this task, models need to identify an absent student through strategic range queries. Students respond by raising hands, but may be dishonest in their responses following specific patterns of truthfulness and deception.

---

**Case N.25: AttendanceCheck Problem Template**

Let's play the Attendance Check Game! Your task is to find the absent student through a series of range queries.
Rules:
1. There are $\{n\}$ students (numbered 1 to $\{n\}$)
2. Exactly one student is absent
3. You can make queries about ranges of students
4. Students might be dishonest in their responses
Query Types:
1. To make a range query:
Format: "My Query: $l$ $r$"
where:
- $l$ and $r$ are the range boundaries ($1 \leq l \leq r \leq \{n\}$)
Example: "My Query: 1 4"
Response will be number of students who raised hands
2. To mark absent student:
Format: "My Answer: $a$"
where:
- $a$ is the student number you think is absent
Example: "My Answer: 3"
Response Types for Range Queries:
For a query $(l, r)$, you'll get either $r - l$ or $r - l + 1$ students raising hands:
1. True Positive: $r - l + 1$ present, $r - l + 1$ raised
2. True Negative: $r - l$ present, $r - l$ raised
3. False Positive: $r - l$ present, $r - l + 1$ raised
4. False Negative: $r - l + 1$ present, $r - l$ raised
Important Rules:
1. Students will never answer honestly 3 times in a row
2. Students will never answer dishonestly 3 times in a row
Example Interaction:

---

You: "My Query: 1 4"
Me: "3" (3 students raised hands)
You: "My Query: 3 5"
Me: "2" (2 students raised hands)
You: "My Answer: 2"
Me: "Correct"
Instructions:
1. Make queries based on previous responses
2. Use exactly the formats shown above
3. Explain your reasoning before each query
Remember:
- Plan your queries carefully
- Students are strategically dishonest
- Pattern of honesty/dishonesty is key
- Think about overlapping ranges
Ready to start? Make your first query!

---

### Case N.26: AttendanceCheck Difficulty Levels

Easy: $n = 5 - 9$, Medium: $n = 10 - 14$, Hard: $n = 15 - 20$

---

**BinaryNumberGuessing**   In this task, models need to discover a hidden number through strategic subtraction operations. Each operation reveals the count of 1s in the binary representation of the resulting number, helping deduce the current value.

---

### Case N.27: BinaryNumberGuessing Problem Template

Let's play Binary Number Guessing! Your task is to guess the original hidden number by performing subtraction operations.
Rules:
1. There is a hidden positive integer $k$ ($1 \leq k \leq n$)
2. You will be told the number of 1s in its binary representation
3. For each operation, you can:
- Subtract any positive integer $x$ from the current number
- After subtraction, you'll be told the new count of 1s in binary
- If you try to subtract a number larger than current $k$, you will get a response of "Invalid"
4. Your goal is to guess the current number after all of your operations
Query Types:
1. Make a subtraction:
Format: "My Operation: $X$"
where $X$ is the number you want to subtract
Response will be:
- Count of 1s in new binary number (if valid)
- "Invalid" (if $X$ larger than current $k$)
2. Submit final answer:
Format: "My Answer: $k$"
where $k$ is your guess for current number
Response will be:
- "Correct" (if right)
- "Incorrect" (if wrong)
- "Invalid" (if invalid format)
Example Interaction:
- Original number = 3 (binary: 11, count of 1s: 2)
You: "My Operation: 1"
Me: "1" (current number is 2, binary: 10)
You: "My Operation: 1"
Me: "1" (current number is 1, binary: 1)
You: "My Answer: 1"

Me: "Correct" (current number is 1, correct!)
Instructions:
1. Make operations based on previous results
2. Use exactly the formats shown above
3. Explain your reasoning before each operation
Remember:
- Don't subtract more than current number
- Track binary representation changes
- Consider patterns in 1s count
- Invalid operations waste moves
Ready to start? Make your first query!

**Case N.28: BinaryNumberGuessing Difficulty Levels**

Easy: $n = 50$, Medium: $n = 150$, Hard: $n = 500$

**HiddenNumberFinding** In this task, models need to discover a hidden number through strategic set queries. Responses might be deceptive, but follow a pattern where at least one of any two consecutive queries is truthful, while direct guesses are always answered honestly.

**Case N.29: HiddenNumberFinding Problem Template**

Let's play Find the Hidden Number Game! Your task is to discover a hidden number through a series of queries and guesses.
Rules:
1. There is a hidden number $x$ between 1 and $\{n\}$
2. For each query, you can ask about a set of numbers:
- You choose any non-empty set of numbers
- System will tell you if $x$ is in that set ("YES") or not ("NO")
- WARNING: Responses might be lies!
- BUT: At least one answer out of any two consecutive queries is truthful
3. For guesses:
- You can directly guess what $x$ is
- Guesses are always answered truthfully
- A correct guess ends the game
Query Types:
1. To make a set query:
Format: "My Query: $k$ $n_1$ $n_2$ ... $n_k$"
where:
- $k$ is the size of your set
- $n_1$ to $n_k$ are the numbers in your set
Example: "My Query: 3 1 2 3"
2. To make a guess:
Format: "My Answer: $x$"
Example: "My Answer: 2"
Example Interaction:
You: "My Query: 3 1 2 3"
Me: "YES"
You: "My Query: 2 4 5"
Me: "YES"
You: "My Answer: 4"
Me: "Correct"
Instructions:
1. Make queries based on previous responses
2. Use exactly the formats shown above
3. Explain your reasoning before each query
Important Notes:
- At least one of any two consecutive queries is truthful

- Guesses are always answered truthfully
- Plan your strategy carefully!
 Remember:
- Track truthful/deceptive patterns
- Use overlapping sets strategically
- Consider binary search approaches
 Ready to start? Make your first query!

---

### Case N.30: HiddenNumberFinding Difficulty Levels

Easy: $n = 19/20$, Medium: $n = 30$, Hard: $n = 40$

---

**MahjongDetective**    In this task, models need to discover a hidden set of Mahjong tiles through strategic tile additions. Each addition reveals changes in the number of valid combinations (triplets and straights), helping deduce the original set composition.

---

### Case N.31: MahjongDetective Problem Template

Let's play Mahjong Detective Game! Your task is to discover Yui's mysterious tile set through careful queries.
Rules:
1. There is a hidden set of Mahjong tiles
2. Each tile has a value from 1 to $\{n\}$
3. Each value appears at most $\{n\}$ times
4. You need to find how many tiles of each value exist
5. You can add tiles to help your investigation
Special Combinations:
- Triplet: Three tiles with same value (e.g., $\{2, 2, 2\}$)
- Straight: Three consecutive values (e.g., $\{2, 3, 4\}$)
Note: Same-value tiles are treated as different piece!
Query Types:
1. To add a tile:
Format: "My Query: $+ x$"
where:
- $x$ is the value of tile to add (1 to $\{n\}$)
Example: "My Query: + 3"
Response will be:
- Number of triplets in new set
- Number of straights in new set
2. To submit final answer:
Format: "My Answer: $a_1 \, a_2 \, ... \, a_{\{n\}}$"
where $a_i$ is number of tiles with value $i$ AFTER ALL YOUR ADDITIONS
Example: "My Answer: 2 1 3 0 2 ..."
Example Interaction:
Initial set has:
- 1 triplet
- 6 straights
You: "My Query: + 1"
Me: "2 9" (new set has 2 triplets, 9 straights)
You: "My Query: + 1"
Me: "5 12" (new set has 5 triplets, 12 straights)
You: "My Query: + 2"
Me: "5 24" (new set has 5 triplets, 24 straights)
You: "My Query: + 5"
Me: "6 24" (new set has 6 triplets, 24 straights)
You: "My Answer: 2 1 3 0 2 ..."
(This answer includes ALL tiles, including the ones you added!)
Instructions:

1. Make queries to add tiles strategically
2. Use exactly the formats shown above
3. Explain your reasoning before each addition
4. Watch how combinations change
Remember:
- Each value appears 0 to $\{n\}$ times
- Same-value tiles count as different pieces
- Watch how triplets and straights change
- Your final answer must include your added tiles
Ready to start? Make your first query!

---

**Case N.32: MahjongDetective Difficulty Levels**

Easy: $n = 3$, Medium: $n = 6$, Hard: $n = 9$

---

**MimicHunting**   In this task, models need to identify a shape-shifting mimic among objects through strategic removals. After each removal, objects are mixed and the mimic may change its type, following specific transformation rules.

---

**Case N.33: MimicHunting Problem Template**

Let's play Mimic Hunt Game! Your task is to find a shape-shifting creature among objects through careful observation and removal.
Rules:
1. There are $\{n\}$ objects in a room, each with a type number (1-9)
2. One object is a mimic that can transform into any type
3. The mimic cannot stay the same type for more than 2 stages
Query Types:
1. To remove objects:
Format: "My Query: - $k$ $x_1$ $x_2$ ... $x_k$"
where: - $k$ is number of objects to remove
- $x_1$ to $x_k$ are positions (1-based indexing)
Example: "My Query: - 2 1 5"
Response will be:
- Remaining objects' types after mixing
2. To identify mimic:
Format: "My Answer: $i$"
where $i$ is the position of suspected mimic
Example: "My Answer: 3"
Example Interaction:
Objects: [1,1,2,2,3]
You: "My Query: - 2 1 5"
Me: "[2,1,2]" (remaining objects after mixing)
You: "My Query: - 4 1 2 3 4"
Me: "[2]" (remaining objects after mixing)
You: "My Answer: 5"
Me: "Correct"
Instructions:
1. Each stage:
- Observe current objects
- Either remove some objects or guess mimic
- After removal, objects are mixed and mimic may change
2. Use exactly the formats shown above
3. Explain your reasoning before each action
4. Remember mimic's transformation rules
Remember:
- Object types are numbers 1-9
- Position indices start from 1

- Mimic can't stay same type > 2 stages
- Track type patterns carefully
Ready to start? Make your first query!

---

**Case N.34: MimicHunting Difficulty Levels**

Easy: $n = 5$, Medium: $n = 10$, Hard: $n = 20$

---

**PermutationDiscovery**    In this task, models need to discover a hidden permutation through dynamic queries. A visible permutation changes after each query according to the hidden permutation's rules, requiring careful analysis of transformation patterns.

---

**Case N.35: PermutationDiscovery Problem Template**

Let's play Permutation Discovery Game! Your task is to find a hidden permutation through dynamic queries.
Rules:
1. There are two permutations of length $\{n\}$:
- $p$: hidden permutation you need to discover
- $q$: visible permutation that changes after each query
2. Initially, $q$ is $[1, 2, ..., \{n\}]$
3. After each query, $q$ changes following this rule:
- For each position $i$: $q'[i] = q[p[i]]$
4. Your goal is to discover permutation $p$
Query Types:
1. To ask about $q$'s value:
Format: "My Query: $i$"
where:
- $i$ is a position (1-based indexing)
Example: "My Query: 3"
Response will be the value at position $i$ in current $q$
2. To submit final answer:
Format: "My Answer: $p_1$ $p_2$ ... $p_{\{n\}}$"
where $p_1$ to $p_{\{n\}}$ form your guessed permutation
Example: "My Answer: 4 2 1 3"
Example Interaction:
Initial $q = [1, 2, ..., \{n\}]$
You: "My Query: 3"
Me: "3"
[$q$ updates based on $p$]
You: "My Query: 2"
Me: "2"
[$q$ updates again]
You: "My Answer: 4 2 1 3"
Instructions:
1. Make queries based on previous results
2. Use exactly the formats shown above
3. Explain your reasoning before each query
4. Watch how $q$ changes after each query
Remember:
- $q$ starts as $[1, 2, ..., \{n\}]$
- Position indices start from 1
- $q$ changes after every query
- Think carefully about which positions to query
Ready to start? Make your first query!

**Case N.36: PermutationDiscovery Difficulty Levels**

Easy: $n = 4$, Medium: $n = 5$, Hard: $n = 6$

**TrainPursuit**   In this task, models need to locate a moving train on a circular railway through range queries. The train moves up to a certain number of stations after each query, following a circular pattern that wraps around from the last station to the first.

**Case N.37: TrainPursuit Problem Template**

Let's play Train Pursuit Game! Your task is to find a moving train on a circular railway through range queries.
Rules:
1. There is a train hidden at one of $\{n\}$ stations (numbered 1 to $\{n\}$)
2. The train moves circularly:
- Can move up to $\{k\}$ stations after each query
- After station $\{n\}$, continues from station 1
- Example: at station $\{n\}$, moving 2 stations means going to station 2
3. You can make range queries to find the train
4. Each query must be in valid format or you'll get "Invalid" response
Query Types:
1. To make a range query:
Format: "My Query: $l\ r$"
where:
- $l$ and $r$ are station numbers (1-based indexing)
- $l \leq r \leq \{n\}$
Example: "My Query: 3 5"
Response will be:
- "Yes" if train is in this range
- "No" if train is not in this range
- "Invalid" if query format is incorrect
2. To catch the train:
Format: "My Answer: $x$"
where $x$ is the station you think the train is now at
Example: "My Answer: 5"
Example Movement:
If train is at station 1 and moves 2 stations:
- First move: station 1 $\rightarrow$ station 3
- Second move: station 3 $\rightarrow$ station 5
Instructions:
1. Make queries based on previous results
2. Use exactly the formats shown above
3. Explain your reasoning before each query
4. Remember circular movement pattern
Remember:
- Train is at a station numbered 1 to $\{n\}$
- Train moves up to $\{k\}$ stations circularly
- Query format must be exact
- Need to find exact location to win
- Invalid queries will receive "Invalid" response
Ready to start? Make your first query!

**Case N.38: TrainPursuit Difficulty Levels**

Easy: $n <= 5$, Medium: $5 < n <= 7$, Hard: $7 < n <= 9$

**ZeroFinding**   In this task, models need to locate the $k$-th zero in a hidden binary array through range sum queries. Non-target zeros transform into ones when discovered, requiring strategic query placement and careful tracking of zero positions.

**Case N.39: ZeroFinding Problem Template**

Let's play Zero Finding Game! Your task is to find the $\{k\}$-th zero in a hidden binary array through range sum queries.
Rules:
1. There is a hidden array of $\{n\}$ elements (all 0s and 1s)
2. You need to find the $\{k\}$-th zero
3. Each time you find a non-target zero (not $\{k\}$-th), it turns into 1
4. The game continues until you find the $\{k\}$-th zero
Query Types:
1. To make a range sum query:
Format: "My Query: $l$ $r$"
where:
- $l$ and $r$ are positions (1-based indexing)
- $l \leq r \leq \{n\}$
Example: "My Query: 4 6"
Response will be the sum of elements in positions $l$ to $r$
2. To submit temporary answer:
Format: "My Answer: $x$"
where $x$ is position of a non-$\{k\}$-th zero
Example: "My Answer: 5"
3. To submit final answer:
Format: "My Final Answer: $x$"
where $x$ is position of the $\{k\}$-th zero
Example: "My Final Answer: 3"
Example Interaction:
Finding 2nd zero:
You: "My Query: 4 6"
Me: "1" (sum in range [4,6])
You: "My Answer: 5"
Me: "Correct! Non-target zero found and turned to 1"
You: "My Final Answer: 3"
Me: "Correct! You found the 2nd zero!"
Instructions:
1. Game Process:
- Make queries to locate zeros
- Use "My Answer" for non-$\{k\}$-th zeros
- Use "My Final Answer" for the $\{k\}$-th zero
- Array updates when non-target zeros are found
2. Use exactly the formats shown above
3. Explain your reasoning before each action
Remember:
- Array only contains 0s and 1s
- Position indices start from 1
- Non-target zeros turn into 1 when found
- Each query shows sum in range
- Use different formats for target and non-target zeros
Ready to start? Make your first query!

**Case N.40: ZeroFinding Difficulty Levels**

Easy: $n = 10$, Medium: $n = 50$, Hard: $n = 100$

N.3 STATE OPERATION

**MazeNavigation** In this task, models need to navigate through a maze with potentially swapped directional controls to reach a finish point. Models must deduce any control swaps while avoiding dangerous cells and staying within grid boundaries.

**Case N.41: MazeNavigation Problem Template**

Let's play Maze Navigation Game! Your task is to navigate through a maze with potentially swapped controls to reach the finish point.
Rules:
1. Game Field:
- A $\{n\} * \{m\}$ grid with three types of cells:
* "." - normal cell you can visit
* "F" - finish cell (exactly one)
* "*" - dangerous cell (avoid these)
- Coordinates are 1-based indexing: (row, column)
- Current cell positions:
* Start: $\{start\_pos\}$ (top-left corner)
* Finish: $\{finish\_pos\}$
* Dangerous cells:
$\{dangerous\_str\}$
2. Movement Controls:
- Four direction buttons: U(up), D(down), L(left), R(right)
- Button Functions may be swapped:
* L and R might be swapped with each other
* U and D might be swapped with each other
- Swaps (if any) are set at game start and remain fixed
- Effects of each button when NOT swapped:
* U: moves to $(current\_row - 1, current\_col)$
* D: moves to $(current\_row + 1, current\_col)$
* L: moves to $(current\_row, current\_col - 1)$
* R: moves to $(current\_row, current\_col + 1)$
3. Movement Rules:
- Each move returns your new position $(x, y)$
- If move is invalid (out of grid), position stays same
- Grid boundaries: $1 \leq row \leq \{n\}, 1 \leq column \leq \{m\}$
- If you hit dangerous cell, returns $(-1, -1)$ and game ends
- When you reach finish cell ($\{finish\_pos\}$), game ends successfully
Move Types:
1. To make a move:
Format: "My Move: $X$"
where $X$ is one of: U, D, L, R
Example: "My Move: R"
2. System Response:
Format: "$x\ y$"
where:
- $(x, y)$ is your new position
- $(-1, -1)$ if you hit dangerous cell
Example: After "My Move: R" at (1, 1), response might be "1 2"
Instructions:
1. Make moves based on previous responses
2. Use exactly the format shown above
3. Explain your reasoning before each move
Remember:
- Start position is $\{start\_pos\}$
- Controls might be swapped
- Avoid dangerous cells at: $\{dangerous\_str\}$
- Target is to reach $\{finish\_pos\}$
- Watch for grid boundaries: $1 \leq row \leq \{n\}, 1 \leq column \leq \{m\}$
Current Grid Layout: $\{grid\_str\}$
Ready to start? Make your first query!

**Case N.42: MazeNavigation Difficulty Levels**

Easy: $n = 4$, Medium: $n = 5$, Hard: $n = 6$

**TreasureHunt**   In this task, models need to explore a forest where junction numbers are hidden and scrambled. Navigation requires strategic use of path counts and flags, as connected junctions appear in random order at each visit.

**Case N.43: TreasureHunt Problem Template**

Let's play the Treasure Hunt Game! Your task is to explore an enchanted forest where a mischievous wizard keeps scrambling the junction numbers to confuse you.
Rules:
1. Game Setup: - Enchanted forest with $\{n\}$ junctions
- Each junction contains a treasure
- You start at junction 1
- Initial flag placed at starting junction
- Junctions are connected by fixed paths
2. Game Mechanics:
What You Can See:
- At each junction, you can only see:
* Number of paths at each connected junction
* Whether you've placed a flag there
The Wizard's Trick:
- The wizard hides real junction numbers
- Each time you visit a junction, connected junctions are shown in random order
- Though connections stay the same, you can't identify specific junctions
- Must use path counts and flags to navigate
3. Information Format:
I provide: "R $d$ $deg_1$ $flag_1$ $deg_2$ $flag_2$ ... $deg_d$ $flag_d$"
- R: you're at current junction
- $d$: number of connected junctions
- $deg_i$: number of paths at connected junction $i$
- $flag_i$: flag status at connected junction $i$ (0=no, 1=yes)
Example: "R 3 2 1 4 0 3 0" means:
- 3 connected junctions
- First has 2 paths and is flagged
- Second has 4 paths and no flag
- Third has 3 paths and no flag
Query Type:
Format your move as: "My Choice: $X$"
where $X$ is from 1 to $d$ (position in current list)
Example Round:
Starting at junction 1:
Me: "R 2 2 0 2 0"
- Two connected junctions
- Both have 2 paths
- Neither has your flag
You: "My Choice: 1"
- Moving to first listed junction
Me: "R 2 2 0 2 1"
- Two connected junctions shown
- One leads back (has your flag)
- One is unexplored (no flag)
You: "My Choice: 1"
- Moving to unflagged junction
Instructions:
1. Give your reasoning before each choice

2. Wait for response before next move
3. Use exactly the format shown above
Remember:
- Real junction numbers are hidden
- Connected junctions appear in random order each visit
- Use path counts and flags to track progress
- Must visit all junctions
- Invalid move = automatic loss
Ready to start? Make your first query!

### Case N.44: TreasureHunt Difficulty Levels

Easy: $n = 6$, Medium: $n = 7$, Hard: $n = 8$

**SafepathFinding**  In this task, models need to navigate from start to goal on a grid while avoiding hidden traps. Each position reveals the number of traps in adjacent cells, requiring careful analysis of danger levels to choose safe moves.

### Case N.45: SafepathFinding Problem Template

Let's play SafepathFinder! Your task is to find a safe path from start to the goal while avoiding hidden traps.
Rules:
1. You are an explorer on a $n*n$ grid
2. Start: $(1, 1)$, Goal: $(n, n)$
3. Each cell can be either:
- SAFE: can move through
- TRAP: ends game if stepped on (hidden)
4. At each cell, you can:
- See the number of traps in adjacent cells (DANGER_LEVEL)
- Cannot see traps until stepped on them
5. Movement rules: - From position $(x, y)$, you can move to any adjacent cell:
- $(x - 1, y - 1)$, $(x - 1, y)$, $(x - 1, y + 1)$
- $(x, y - 1)$,                ,$(x, y + 1)$
- $(x + 1, y - 1)$, $(x + 1, y)$, $(x + 1, y + 1)$
- Cannot move outside grid
- Example: from $(2, 2)$ you can move to any surrounding cell
Query Type:
Format: "My Choice: $X$ $Y$"
where $X, Y$ are coordinates (1-based)
Example: "My Choice: 2 3"
Response Format:
DANGER_LEVEL $v$
- $v$ is the number of traps in the 8 adjacent cells
- Higher number means more danger nearby
- 0 means no traps in adjacent cells
Example interaction:
You: "My Choice: 2 1"
Me: "DANGER_LEVEL 1"
You: "My Choice: 3 2"
Me: "DANGER_LEVEL 2"
Game Ends When:
- SUCCESS: Reach $(n, n)$
- FAILURE: Step on a trap
- INVALID: Try to move outside grid or not to adjacent cell
Instructions:
1. Make moves based on danger levels
2. Use exactly the format shown above

3. Explain your reasoning before each move
Strategy Tips:
- Higher DANGER_LEVEL means more risk
- Watch how DANGER_LEVEL changes as you move
- Use these changes to deduce trap locations
- Sometimes longer path might be safer
- Pay attention to diagonal movements too
Ready to start? Make your first move!

**Case N.46: SafepathFinding Difficulty Levels**

Easy: $n = 5$, Medium: $n = 6$, Hard: $n = 7$

**RainbowCandyFactory** In this task, models need to guide a candy through a factory grid with hidden color-changing devices. The goal is to reach the destination with a specific target color by strategically using dye machines and bleach machines.

**Case N.47: RainbowCandyFactory Problem Template**

Let's play Rainbow Candy Factory! Your task is to guide a candy through hidden devices to reach the destination with target color.
Rules:
1. Control a candy through a $n * n$ factory grid
2. Start at $(1, 1)$ with white color (W), reach $(n, n)$
3. Hidden devices in cells marked by X:
- Dye Machines: R(red), G(green), B(blue)
- Empty cells (-)
4. Bleach Machine is shown as W(white) in the map and it can reset any color to white
5. Each level gives a target color to achieve
Move Types:
1. To make a move:
Format: "My Move: $Y$"
where: - $Y$ is one of: N, E, S, W (directions)
Example: "My Move: E"
Color Rules:
- Initial color: White (W)
- Basic colors: Red (R), Green (G), Blue (B)
- Mixed colors: Yellow (Y), Cyan (C), Purple (P)
- Color mixing: R+G=Y, G+B=C, R+B=P
- Bleach Machine (W) resets ANY color back to White
- For Mixed colors, bleaching machine can make it White, but dyeing machine cannot change its color
Example Interaction:
You: "My Move: E"
Me: "R"
You: "My Move: S"
Me: "W"
You: "My Move: E"
Me: "G"
Instructions:
1. Make moves based on color feedback
2. Use exactly the format shown above
3. Explain your reasoning before each move
4. Watch out for bleach machines that reset progress
Initial Map: $\{initial\_map\}$
Target Color: $\{target\}$
Remember:
- Start at $(1, 1)$ with White color

- Cannot see machine types until encountered
- Bleach machines reset ALL colors to White
- You can go to the cell you've been to
- Moving out of bounds will result in failure
- Must reach $(n, n)$ with target color
Ready to start? Make your first move!

**Case N.48: RainbowCandyFactory Difficulty Levels**

Easy: $n = 3$, Medium: $n = 4$, Hard: $n = 5$

**MagneticFieldExploration**    In this task, models need to navigate through a grid containing magnetic fields that force movement in specific directions. Success requires understanding the behavior of different magnetic fields while avoiding danger zones to reach the goal.

**Case N.49: MagneticFieldExploration Problem Template**

Let's play Magnetic Field Explorer! Your task is to navigate through a grid with mysterious magnetic forces.
Rules:
1. Game Field:
- A $n * n$ grid with:
* Numbers (1-4) - Different types of magnetic fields
* "." - Neutral space
* "X" - Danger zone (avoid these)
* "G" - Goal (reach here to win)
- Start: $(1, 1)$ (top-left corner)
- Goal: $(n, n)$ (bottom-right corner)
2. Magnetic Fields:
- Four types of magnetic fields (1-4)
- Each number represents a unique direction (North, South, East, or West)
- You'll discover the direction of each number through movement
- Same number always means same direction
- When you enter a magnetic field:
* You will be forced to move one step in its direction
* If that step would hit a boundary, you stay on the magnetic field
* If that step would hit a danger zone, you lose
* If that step would hit another magnetic field, you move there and it activates
3. Movement Rules: - Basic moves: U(up), D(down), L(left), R(right)
- Movement sequence for each turn: 1. You move one step in your chosen direction
2. If you land on:
- Magnetic field: Move one step in its direction unless that step would hit a boundary
- Danger zone: You lose
- Neutral space: Stay there
3. If magnetic field pushed you to another magnetic field, repeat step 2
Current Grid Layout (with coordinates):
$\{grid\_str\}$
$\{position\_str\}$
Query Types:
1. To make a move:
Format: "My Move: $X$"
where $X$ is one of: U, D, L, R
Example: "My Move: R"
2. System Response:
Format: "$x \ y$"
- Shows your final position coordinates
- $(-1, -1)$ if you hit danger zone
Instructions:

1. Make moves based on previous results
2. Use exactly the format shown above
3. Explain your reasoning before each move
Remember:
- Each number (1-4) represents a fixed direction
- Figure out what direction each number represents
- Magnetic fields activate when you land on them
- Avoid danger zones (X)
- Reach goal (G) to win
- You don't necessarily need to figure out or pass through the magnetic fields; your goal is only to reach the target zone $(n, n)$ safely
Ready to start? Make your first move!

### Case N.50: MagneticFieldExploration Difficulty Levels

Easy: $n = 3$, Medium: $n = 4$, Hard: $n = 5$

**FindingBiggest** In this task, models need to locate and collect the highest value treasure on a grid through strategic movement. Each position reveals directional hints to nearby treasures, but these hints may be deceptive following specific patterns.

### Case N.51: FindingBiggest Problem Template

Let's play Finding the Biggest! Your task is to find and collect the highest value treasure through strategic movement on the grid.
Rules:
1. You are an explorer on a $n*n$ grid
2. There are exactly 2 treasures hidden on the grid
3. Each treasure has a value between 1 and 100
4. You start at position $(1, 1)$
5. Movement rules:
- From position $(x, y)$, you can move to any of its 8 adjacent cells:
- $(x - 1, y - 1)$, $(x - 1, y)$, $(x - 1, y + 1)$
- $(x, y - 1)$,                    $,(x, y + 1)$
- $(x + 1, y - 1)$, $(x + 1, y)$, $(x + 1, y + 1)$
- Cannot move outside the grid boundaries
6. Direction System:
- N: treasure is somewhere in the region above your current position
- NE: treasure is somewhere in the upper-right region
- E: treasure is somewhere in the region to your right
- SE: treasure is somewhere in the lower-right region
- S: treasure is somewhere in the region below your current position
- SW: treasure is somewhere in the lower-left region
- W: treasure is somewhere in the region to your left
- NW: treasure is somewhere in the upper-left region
The direction indicates a general area, not a specific cell
7. MAGNETIC INTERFERENCE:
- When you get a direction, there's 50% chance it's completely wrong
- However, wrong directions never appear in consecutive moves
- If you get a wrong direction, the next move's direction is guaranteed correct
Query Types:
1. To move to a position:
Format: "My Choice: $X\ Y$"
where $X, Y$ are grid coordinates (1-based)
Example: "My Choice: 2 3" moves to row 2, column 3
2. To collect treasure:
Format: "My Choice: COLLECT"
- Only use when you're sure you're on the highest value treasure

- You only get one collection attempt
Response Types:
- If you find a treasure: "TREASURE $v$" ($v$ is the treasure's value)
- If empty cell: "EMPTY $dir$" ($dir$ indicates which region contains nearest treasure)
- If invalid move: "INVALID_MOVE"
Example interaction:
You: "My Choice: 2 2"
Me: "EMPTY SW" (indicates treasure might be in lower-left region, but could be wrong)
You: "My Choice: 1 2"
Me: "EMPTY NE" (guaranteed correct: treasure is in upper-right region)
You: "My Choice: 2 3"
Me: "TREASURE 80"
You: "My Choice: COLLECT"
Me: "Win"
Instructions:
1. Make moves based on directional hints
2. Use exactly the formats shown above
3. Explain your reasoning before each move
Key Points:
- Directions point to regions, not specific cells
- If a direction seems wrong, the next one will be correct
- Must find and be at highest value treasure to win
- Wrong COLLECT attempt = game over
Ready to start? Make your first move!

### Case N.52: FindingBiggest Difficulty Levels

Easy: $n = 3$, Medium: $n = 4$, Hard: $n = 5$

**DarkMazeExploration** In this task, models need to navigate through a dark maze where walls are only revealed upon encounter. Success requires careful mapping of discovered walls and strategic path planning to reach the exit.

### Case N.53: DarkMazeExploration Problem Template

Let's play DarkMazeExplorer! Your task is to find your way through a dark maze using only directional movements.
Rules:
1. You are exploring a $n*n$ maze
2. Each cell may have walls in any direction (North, East, South, West)
3. You start at position $(1, 1)$ and must reach $(n, n)$
4. You can only make one directional move at a time
5. You cannot move through walls or outside the maze boundaries
Query Type:
Format: "My Choice: $X$"
where:
- $X$ is one of: N, E, S, W (representing directions)
- N = North, E = East, S = South, W = West
Example: "My Choice: E"
Response Types:
- MOVED: successfully moved into the next cell in your chosen direction
- BLOCKED: wall exists in that direction
- INVALID: tried to move outside maze boundaries
- WIN: reached the exit at $(n, n)$
Example Interaction:
Starting at $(1, 1)$ with North and West walls
You: "My Choice: E"
Me: "MOVED"

You: "My Choice: N"
Me: "BLOCKED"
You: "My Choice: S"
Me: "WIN"
Instructions:
1. Make moves based on feedback
2. Use exactly the format shown above
3. Explain your reasoning before each move
4. Plan your path carefully
Remember:
- Starting room $(1, 1)$ has North and West walls
- You can only see walls when you encounter them
- Need to mentally map the maze
- Cannot move through walls or outside boundaries
- Must reach $(n, n)$ to win
Ready to start? Make your first move!

---

**Case N.54: DarkMazeExploration Difficulty Levels**

Easy: $n = 2$, Medium: $n = 3$, Hard: $n = 4$

---

**ColorMagic**  In this task, models need to transform a grid of colored cells to a uniform color through magical operations. Success requires discovering the mapping between operation numbers and their effects while planning strategic color transformations.

---

**Case N.55: ColorMagic Problem Template**

Let's play Color Magic! Your task is to make all cells the same color through magical color transformations.
Rules:
1. You have a $n * n$ grid where each cell contains one of three colors: Red(R), Blue(B), Yellow(Y)
2. There are three magic operations with unknown number assignments (1, 2, or 3):
- Magic Alpha: Selected cell rotates R->B->Y->R, adjacent cells rotate R->Y->B->R
- Magic Beta: Selected cell rotates B->Y->R->B, adjacent cells rotate B->R->Y->B
- Magic Gamma: Selected cell stays same, adjacent cells swap colors (R<->B, B<->Y, Y<->R)
3. Your goal is to make all cells the same color
Move Types:
Format: "My Move: OPERATION POSITION"
where:
- OPERATION is one of: 1, 2, 3 (each corresponds to a magic type)
- POSITION is cell number (1-$n * n$, numbered left to right, top to bottom)
Example: "My Move: 2 5"
Instructions:
1. Make moves based on observed color changes
2. Use exactly the format shown above
3. Explain your reasoning before each move
4. Try to discover which number corresponds to which magic
Example Interaction:
Current Grid:
R B Y
B R B
Y R Y
You: "My Move: 1 5"
Me:
R R Y
R R R
Y B Y
- Note: This is just an example; in reality, 1 may not correspond to this operation.

Initial Grid: $initial\_grid$
Remember:
- Each number (1,2,3) maps to one magic type (Alpha/Beta/Gamma)
- You must figure out the mapping through experimentation
- Grid positions are numbered from 1 to $n*n$ from left to right, top to bottom
- Adjacent means sharing an edge (not diagonal)
- Need to make all cells the same color to win
Ready to start? Make your first move!

### Case N.56: ColorMagic Difficulty Levels

Easy: $n = 3$, Medium: $n = 4$, Hard: $n = 5$

**ChemicalSynthesis**   In this task, models need to create a target compound through strategic chemical operations. Each operation has consistent but unknown number assignments and may produce unexpected results due to chemical instability.

### Case N.57: ChemicalSynthesis Problem Template

Let's play Chemical Synthesis! Your task is to create compound $\{target\}$ containing $n$ elements through different operations in an unstable environment.
Rules:
1. Basic Setup:
- Initial compounds: $\{',' .join(init\_compounds)\}$
- Goal: Create $\{target\}$
- Four types of operations (1,2,3,4)
- Element order matters (ABC $\neq$ CBA)
- After each operation, resulting compounds and original compounds can be used
2. Operation Types (numbers 1-4 each correspond to one of these):
SPLIT:
- Usually breaks a compound into two parts of its first element and the other elements
- Sometimes splits at a random position due to instability
- Example: ABC $\rightarrow$ A + BC (normal) or AB + C (unstable)
- Format: "My Move: $X$ $N$" ($X$ is a compound, and $N = 1/2/3/4$)
MERGE:
- Combines two compounds into one
- May cause a catalytic reaction that changes element order
- Result usually keeps elements in order, but might rearrange
- Example: AB + CD $\rightarrow$ ABCD (normal) or ACBD (catalytic)
- Format: "My Move: $X$ $Y$ $N$" ($X, Y$ are two compounds, and $N = 1/2/3/4$)
SWAP:
- Exchanges elements within a compound
- High energy might cause multiple swaps
- Example: ABC $\rightarrow$ CBA (normal) or BAC (partial)
- Format: "My Move: $X$ $N$" ($X$ is a compound, and $N = 1/2/3/4$)
EXTRACT:
- Takes out one element from a compound
- Usually the last element, but might extract a random element
- Example: ABC $\rightarrow$ C (normal) or B (unstable)
- Format: "My Move: $X$ $N$" ($X$ is a compound, and $N = 1/2/3/4$)
3. Operation Format and Responses:
Single Compound Operations (SPLIT, SWAP, EXTRACT):
- Format: "My Move: $X$ $N$"
Example: "My Move: BC 1"
MERGE Operation:
- Format: "My Move: $X$ $Y$ $N$"
Example: "My Move: AB CD 2"
System Responses:

- Valid query: "Available: [list of unrepeated available compounds]"
- Invalid query: "Wrong type"/"Invalid format"/"Invalid compound"
- Success: "WIN"
4. Current State:
Available Compounds: {$init\_compounds$}
Important Notes:
- Element order matters (ABC ≠ CBA)
- Operations are consistent but their numbers (1-4) are unknown
- Chemical instability may cause unexpected results
- Goal compound must match exactly (including element order)
- Can only operate on currently available compounds
- System will return "Wrong type" if:
* Using single-element compounds for SPLIT/SWAP/EXTRACT
* Using wrong number of compounds for operation
Example Interactions:
Initial: "ABC AB D"
You: "My Move: ABC 1"
Me: "Available: ABC A BC AB D" (normal split)
You: "My Move: AB D 2"
Me: "Available: ABC A BC AB D DAB" (unstable merge)
Example Invalid Interactions:
You: "My Move: A B 1" (invalid: single element for SPLIT)
Me: "Wrong type"
You: "My Move: AB 2" (invalid: MERGE needs two compounds)
Me: "Wrong type"
Goal: Create {$target$} (exact order matters)
Ready to start! Make your move using the correct format!

## Case N.58: ChemicalSynthesis Difficulty Levels

Easy: $n = 4$, Medium: $n = 6$, Hard: $n = 7$

**CactusSearch**  In this task, models need to find a secret vertex in a cactus graph through strategic guessing. Each incorrect guess reveals which adjacent vertex leads closer to the target, requiring careful navigation of the graph structure.

## Case N.59: CactusSearch Problem Template

Let's play Cactus Search Game! Your task is to find a secret vertex in a cactus graph through strategic guessing.
Rules:
1. The game is played on a cactus graph with {$n$} vertices (numbered from 1 to {$n$})
2. A secret vertex $v$ has been chosen
3. After each incorrect guess, you'll be told which adjacent vertex leads closer to $v$
Game Setup:
This cactus graph consists of {$n$} vertices and {$m$} distinct paths: {$paths\_text$}
Each path represents a sequence of connected vertices, where consecutive vertices are connected by edges.
The graph is structured as a cactus, meaning each edge belongs to at most one cycle.
Query Type:
1. To make a guess:
Format: "My Guess: $x$"
where $x$ is the vertex number ($1 \leq x \leq$ {$n$})
Example: "My Guess: 3"
2. System Response:
- If correct: "FOUND"
- If incorrect: "GO $w$" ($w$ is adjacent vertex closer to target)
Example Interaction:

You: "My Guess: 3"
System: "GO 4"
You: "My Guess: 4"
System: "FOUND"
Instructions:
1. Make guesses based on previous responses
2. Use exactly the format shown above
3. Explain your reasoning before each guess
Remember:
- Each vertex is numbered from 1 to $\{n\}$
- The graph structure is fixed as described above
- Adjacent vertices in paths are directly connected
- Use responses wisely to navigate towards target
Ready to start? Make your first query!

## Case N.60: CactusSearch Difficulty Levels

Easy: $n = 10$, Medium: $n = 12$, Hard: $n = 15$

### N.4    STRATEGIC GAMING

**KnightBattle**    In this task, models need to win a strategic battle between knights through either capture or reaching a target position. Success requires careful planning of L-shaped movements while considering opponent's potential threats.

## Case N.61: KnightBattle Problem Template

Let's play the Knight Battle Game! You are the White Knight and will move first. Your task is to win by either capturing the Black Knight or reaching your target position safely.
Rules:
1. Game Setup:
- Chessboard size: $\{n\}$*$\{m\}$
- You (White Knight) start at: ($\{x1\}, \{y1\}$)
- Opponent (Black Knight) starts at: ($\{x2\}, \{y2\}$)
- Your target: ($\{tw\_x\}, \{tw\_y\}$)
- Opponent's target: ($\{tb\_x\}, \{tb\_y\}$)
2. Knight's Movement Rules:
From your current position $(x, y)$, you can move to:
1. Up 2, Right 1: $(x + 1, y + 2)$
2. Up 2, Left 1: $(x - 1, y + 2)$
3. Down 2, Right 1: $(x + 1, y - 2)$
4. Down 2, Left 1: $(x - 1, y - 2)$
5. Right 2, Up 1: $(x + 2, y + 1)$
6. Right 2, Down 1: $(x + 2, y - 1)$
7. Left 2, Up 1: $(x - 2, y + 1)$
8. Left 2, Down 1: $(x - 2, y - 1)$
* All moves must stay within board boundaries (1 to $\{n\}$, 1 to $\{m\}$)
3. Victory Conditions:
You win if either:
- You move to Black Knight's position (capture)
- You reach ($\{tw\_x\}, \{tw\_y\}$) and Black Knight cannot attack this position
* A position is under attack if opponent's knight can move there next turn
Query Type:
Format: "My Move: $x$ $y$"
where $x, y$ are your new coordinates
Example: "My Move: 4 4"
Example Interaction:
You (at $\{x1\}, \{y1\}$): "My Move: 4 4"

- Moving to position (4,4)
Me: "6 3"
- Black Knight moves to (6,3)
You: "My Move: 5 6"
- Moving to position (5,6)
Me: "5 1"
- Black Knight moves to (5,1)
Instructions:
1. Make moves based on board state
2. Use exactly the format shown above
3. Explain your reasoning before each move
Remember:
- You are White Knight and move first
- Use L-shaped movements only
- Use exact format: "My Move: X Y"
- Stay within board boundaries
- Plan moves to either:
* Capture Black Knight
* Reach ($\{tw\_x\}, \{tw\_y\}$) safely
- Invalid move = immediate loss
- You have at most 15 rounds to defeat the Black Knight
Ready to start? Make your first move!

### Case N.62: KnightBattle Difficulty Levels

Easy: $n = 6$, Medium: $n = 8$, Hard: $n = 16$

**ZigzagGraph** In this task, models need to win a strategic graph traversal game where players follow opposite weight constraints. Success requires careful planning of moves while considering both immediate options and future path availability.

### Case N.63: ZigzagGraph Problem Template

Let's play the Zigzag Graph Game! Your task is to win this game by strategically moving through the graph while following increasing or decreasing edge weights.
Rules:
1. Game Setup:
- Graph: $\{n\}*\{n\}$ bipartite graph
- Left nodes: $\{','.join(str(x) for x in range(1, n + 1))\}$
- Right nodes: $\{','.join(str(x) for x in range(n + 1, 2 * n + 1))\}$
- Edge weights:
$\{chr(10).join(edge\_desc)\}$
- All edge weights are distinct
2. Game Mechanics:
- You choose "decreasing" mode and I choose "increasing" mode
- You place token on one node and then I place token on one node
- Players take turns moving token to adjacent unvisited nodes:
* Must move from opponent's last chosen node
* Edge weight must be less than last used edge (for you)
* Edge weight must be greater than last used edge (for me)
- Cannot visit same node twice
3. Victory Conditions:
- Player loses if unable to make a valid move from opponent's node
- Game ends when no legal moves remain
Query Type:
Format: "My Choice: $X$"
where $X$ is the node number $(1 - \{2 * n\})$
Example Round:

Initial placement:
You: "My Choice: 2"
- Placing token at node 2
I: "My Choice: 5"
- Moving from node 2 to node 5 with edge weight 8
You: "My Choice: 3"
- Moving from node 5 to node 3 with edge weight 6
- Following decreasing rule: 6 < 8
I: "My Choice: 6"
- Moving from node 3 to node 6 with edge weight 9
- Following increasing rule: 9 > 6
Instructions:
1. Make moves based on graph state
2. Use exactly the format shown above
3. Explain your reasoning before each move
Remember:
- Use exact format: "My Choice: $X$"
- Must move from opponent's last node
- Follow decreasing weight rule
- Invalid move = automatic loss
Ready to start? Make your first query!

## Case N.64: ZigzagGraph Difficulty Levels

Easy: $n = 5$, Medium: $n = 8$, Hard: $n = 12$

**XORBreaking**    In this task, models need to win a strategic game by breaking numbers using XOR operations. Success requires careful selection and breaking of numbers while forcing the opponent into unbreakable positions.

## Case N.65: XORBreaking Problem Template

Let's play the XOR Break Game! Your task is to win this game by strategically breaking numbers and forcing your opponent into a position where they can't make a valid move.
Rules:
1. Game Setup:
- Initial number: $\{k\}$ ($2 =< k =< n$)
- You play first
- I play second
- Maximum 20 moves allowed
2. Game Mechanics:
First Turn:
- You break initial number $p$ into two numbers $p_1$ and $p_2$
- Must satisfy: $0 < p_1, p_2 < p$ and $p_1 \oplus p_2 = p$
Subsequent Turns:
- Active player does two actions:
1. Choose one number ($p_1$ or $p_2$) from opponent's break
2. Try to break chosen number into two new numbers
- If player cannot break their chosen number, they lose
- Game continues until someone can't break their number
3. XOR Calculation Example:
Breaking 13:
- Can choose 10 and 7 because:
* 10 = 1010 in binary
* 7 = 0111 in binary
* $10 \oplus 7 = 1101 = 13$
- Both numbers are less than 13
- Both numbers are positive

Query Types:
First Turn Format:
- Your move: "Breaking into: $p_1$ $p_2$"
- Example: "Breaking into: 10 7"
Other Turns Format:
- Your move: "Choosing: $p$ Breaking into: $p_1$ $p_2$"
- My response: Either
* "Choosing: $x$ Breaking into: $y$ $z$"
or
* "Choosing: $x$ Cannot break further"
Example Round:
Initial number: 13
You: "Breaking into: 10 7"
- Breaking 13 into $10 \oplus 7$
- Both numbers less than 13
Me: "Choosing: 7 Breaking into: 3 4"
- Selected 7 and broke it into $3 \oplus 4$
You: "Choosing: 3 Breaking into: 2 1"
- Selected 3 and broke it into $2 \oplus 1$
Me: "Choosing: 1 Cannot break further"
- You win! 1 cannot be broken
Instructions:
1. Make moves based on XOR properties
2. Use exactly the format shown above
3. Explain your reasoning before each move
Remember:
- Use exact format for moves
- Numbers must satisfy: * Less than current number * Greater than 0 * XOR to current number
- Invalid break = automatic loss
- More than 20 moves = loss
Ready to start? Make your first query!

**Case N.66: XORBreaking Difficulty Levels**

Easy: $n = 100000$, Medium: $n = 10000000$, Hard: $n = 100000000$

**PizzaSlicing** In this task, models need to win a strategic game by choosing vertices that minimize the total area of triangular slices eaten. Success requires careful calculation of areas while considering both immediate and future slice configurations.

**Case N.67: PizzaSlicing Problem Template**

Let's play the Pizza Slice Game! Your task is to eat as little spinach pizza as possible by strategically choosing vertices. The player who eats less total area wins!
Rules:
1. Game Setup:
- Pizza shape: $\{n\}$-vertex convex polygon
- Vertices: $\{points\_str\}$
- You play first, I play second
- Total $\{(n-2)\}$ turns to complete
2. Game Mechanics:
- Players take turns choosing vertices
- When chosen, player eats triangle formed by:
* The chosen vertex
* Its two neighboring edges
- After each choice, pizza loses one vertex
- Game ends when all pizza is eaten
- Each vertex can only be chosen once

3. Area Calculation Example:

If you choose vertex 1 $(x_1, y_1)$:

- Triangle area = $|(x_2 - x_1)(y_3 - y_1) - (x_3 - x_1)(y_2 - y_1)|/2$
- Where $(x_2, y_2)$ and $(x_3, y_3)$ are neighboring vertices
- Area adds to your total eaten amount
- Player with smaller total area wins!

Query Type:

Format: "My Choice: $X$"

where $X$ is vertex index (1 to $\{n\}$)

Example Round:

You: "My Choice: 1"

Me: "3"

You: "My Choice: 2"

Me: "4"

Result: Add up areas of your triangles and compare with mine to determine winner!

Instructions:

1. Make choices based on area calculations
2. Use exactly the format shown above
3. Explain your reasoning before each choice

Remember:

- Use exact format: "My Choice: $X$"
- Choose only available vertices
- Aim to eat LESS total area than opponent
- Invalid move = automatic loss
- Victory = eating smaller total area than opponent

Ready to start? Make your first query!

### Case N.68: PizzaSlicing Difficulty Levels

Easy: $n = 6$, Medium: $n = 8$, Hard: $n = 12$

**PaperNumbering**  In this task, models need to arrange numbers in non-decreasing order on a line of papers through strategic placement and overwriting. Success requires careful planning of number positions while adapting to new values each round.

### Case N.69: PaperNumbering Problem Template

Let's play the Paper Number Game!

Rules:

1. Game Setup:
- $n$ blank papers in a line, numbered 1 to $n$ from left to right
- Game lasts $\{turns\}$ rounds
- Numbers range from 1 to $\{max\_number\}$

2. Game Mechanics:
- System provides one number (1 to $\{max\_number\}$) each round
- You must choose which paper to write the number on
- You can overwrite existing numbers on papers
- Game ends after $\{turns\}$ rounds or when winning condition is met

3. Winning Condition:
- All papers must have numbers written
- Numbers must be in non-decreasing order from left to right
- Win immediately when condition is met
- Lose if not achieved after $\{turns\}$ rounds

Query Type:

Format: "My Choice: $X$"

where $X$ is paper position (1 to $n$)

Example Round:

Given:

Me: "2"
You: "I'll place 2 on first paper to leave room for larger numbers"
"My Choice: 1"
- Paper state: [2,_,_...]
Me: "1"
You: "I'll place 1 on second paper temporarily"
"My Choice: 2"
- Paper state: [2,1,_...]
Me: "3"
You: "I'll replace 1 with 3 to achieve non-decreasing order"
"My Choice: 2"
- Paper state: [2,3,_...]
Instructions:
1. Make choices based on number sequence
2. Use exactly the format shown above
3. Explain your reasoning before each choice
Remember:
- Use exact format: "My Choice: $X$"
- Choose valid paper positions (1 to $n$)
- Aim for non-decreasing sequence
- Invalid move = automatic loss
Ready to start? Make your first query!
The first number I give you is: $\{initial\_value\}$

### Case N.70: PaperNumbering Difficulty Levels

Easy: $n = 5$, Medium: $n = 10$, Hard: $n = 15$

**GridGame** In this task, models need to win a strategic game by selecting grid cells that minimize their sum while following adjacency rules. Success requires careful planning of cell selections while considering both immediate values and future path availability.

### Case N.71: GridGame Problem Template

Let's play the Grid Game! Your task is to choose cells strategically to win.
Rules:
1. Game Setup:
- Grid size: $\{n\}$*$\{m\}$
- Grid already filled with numbers 1 to $\{n*m\}$
- Each number appears exactly once $\{grid\_str\}$
2. Game Mechanics:
- Players take turns selecting unselected cells
- You move first
- Any cell chosen after first turn must be adjacent to a previously selected cell
- Cells are adjacent if they share an edge (up/down/left/right)
- Game ends when all cells are selected
- You win if your selected numbers sum < my sum
3. Adjacency Example:
For cell $(2, 2)$:
- Adjacent cells: $(1, 2), (2, 1), (2, 3), (3, 2)$
- Diagonal cells like $(1, 1)$ are not adjacent
- Must choose a cell adjacent to any previously selected cell
Query Type:
Format: "My Choice: $x \ y$"
where $x$ is row (1 to $\{n\}$) and $y$ is column (1 to $\{m\}$)
Example Interaction:
You: "My Choice: 2 2"
- Selecting cell at row 2, column 2

Me: "My Choice: 2 3"
- Cell is adjacent to $(2, 2)$
You: "My Choice: 1 2"
- Cell is adjacent to $(2, 2)$
Instructions:
1. Make choices based on grid values
2. Use exactly the format shown above
3. Explain your reasoning before each choice
Remember:
- Use exact format: "My Choice: $x\ y$"
- Choose only adjacent cells after first turn
- First move can be any cell
- Keep track of both sums
- Plan moves to keep your sum smaller
- Invalid move = automatic loss
Ready to start? Make your first choice!

## Case N.72: GridSumGame Difficulty Levels

Easy: $n = 3$, Medium: $n = 5$, Hard: $n = 8$

**GridColoring**   In this task, models need to discover a special rectangular pattern on a grid through strategic cell selection. Success requires finding four differently colored cells that form a rectangle with sides parallel to grid lines.

## Case N.73: GridColoring Problem Template

Let's play the Grid Coloring Game! Find a special rectangular pattern on the grid.
Rules:
1. Game Setup:
- I have a $n$*$n$ grid
- $\{coloring\_description\}$
2. Game Mechanics:
- You can make up to 10 moves
- Each move: Choose an uncolored cell by specifying coordinates $(x, y)$
- I will respond by coloring that cell with a color of my choice (1 to $\{2 * n\}$)
- Your goal: Find 4 colored cells that form a valid rectangle
3. Victory Conditions:
A valid rectangle must:
- Have all 4 cells colored
- Have different colors in all 4 cells
- Form a rectangle with sides parallel to grid lines
Query Types:
1. To choose a cell:
Format: "My Choice: $x\ y$"
where $1 \leq x, y \leq n$
2. To submit answer:
Format: "My Answer: $x_1\ x_2\ y_1\ y_2$"
where $(x_1, y_1), (x_1, y_2), (x_2, y_1), (x_2, y_2)$ are rectangle corners
Example Interaction:
Given a 3*3 grid with initial coloring:
- Cell at position $(1, 1)$ has color 1
- Cell at position $(2, 2)$ has color 2
- Cell at position $(3, 3)$ has color 3
You: "My Choice: 1 2"
Me: "Cell $(1, 2)$ colored with color 4"
You: "My Choice: 2 1"
Me: "Cell $(2, 1)$ colored with color 5"

You: "My Answer: 1 2 1 2"
Me: "Win!"
Instructions:
1. Make choices based on grid state
2. Use exactly the formats shown above
3. Explain your reasoning before each move
Remember:
- Use exact format: "My Choice: $x$ $y$" for queries
- Use exact format: "My Answer: $x_1$ $x_2$ $y_1$ $y_2$" for final answer
- Explain your reasoning before making a choice
- Wait for my color response before next move
- Choosing already colored cell = invalid move = immediate loss
- All 4 cells in rectangle must have different colors
Ready to start? Make your first query!

## Case N.74: GridColoring Difficulty Levels

Easy: $n = 10$, Medium: $n = 20$, Hard: $n = 30$

**GeometricDistance**    In this task, models need to win a strategic game by choosing points that control the parity of cumulative squared distances. Success requires careful calculation of distances while planning moves to achieve an even final sum.

## Case N.75: GeometricDistance Problem Template

Let's play Geometric Distance Game! Your task is to win this game by choosing points and controlling the sum's parity.
Rules:
1. Game Setup:
- Starting point: $(\{sx\}, \{sy\})$
- Available points:
Point 1: $(\{x_1\}, \{y_1\})$
Point 2: $(\{x_2\}, \{y_2\})$
Point 3: $(\{x_3\}, \{y_3\})$
Point 4: $(\{x_4\}, \{y_4\})$
...
Point n: $(\{x_n\}, \{y_n\})$
2. Game Mechanics:
- Players take turns choosing one point
- Each point can only be chosen once
- After each choice, add the squared distance to sum:
* First turn: distance from $(\{sx\}, \{sy\})$ to your choice
* Later turns: distance from opponent's last choice to your choice
- Game ends when all points are chosen
- You win if final sum is even
3. Distance Calculation Example:
If you choose $(0, 1)$:
- From $(0, 0)$: distance squared = $(0 - 0)^2 + (1 - 0)^2 = 0 + 1 = 1$
- Sum becomes 1
Query Type:
Format: "My Choice: $X$"
where $X$ is point index (1 to $n$)
Example Round:
Given:
- Starting point: $(0, 0)$
- Points: $(1, 0)$, $(0, 1)$, $(1, 1)$, $(1, 2)$
You: "My Choice: 4"
- Distance from $(0, 0)$ to $(1, 2)$: $(1 - 0)^2 + (2 - 0)^2 = 1 + 4 = 5$

- Sum = 5
Me: "My Choice: 2"
- Distance from $(1, 2)$ to $(0, 1)$: $(0 - 1)^2 + (1 - 2)^2 = 1 + 1 = 2$
- Sum = 5 + 2 = 7
You: "My Choice: 3"
- Distance from $(0, 1)$ to $(1, 1)$: $(1 - 0)^2 + (1 - 1)^2 = 1 + 0 = 1$
- Sum = 7 + 1 = 8
Me: "My Choice: 1"
- Distance from $(1, 1)$ to $(1, 0)$: $(1 - 1)^2 + (0 - 1)^2 = 0 + 1 = 1$
- Sum = 8 + 1 = 9
Result: You lose! (Final sum = 9 is odd)
Instructions:
1. Make choices based on distance calculations
2. Use exactly the format shown above
3. Explain your reasoning before each choice
Remember:
- Use exact format: "My Choice: $X$"
- Choose only available points (1-$n$)
- Plan moves to make final sum even
- Invalid move = automatic loss
Ready to start? Make your first query!

---

**Case N.76: GeometricDistance Difficulty Levels**

Easy: $n = 4$, Medium: $n = 6$, Hard: $n = 8$

---

**BeeChase** In this task, models need to catch a moving target on a special honeycomb graph by coordinating three bees' movements. Success requires strategic positioning and understanding of graph topology to trap the target.

---

**Case N.77: BeeChase Problem Template**

Let's play the Bee Chase Game! Your task is to catch Nastya by strategically moving three bees on a special honeycomb graph.
Rules:
1. Game Setup:
- Graph: $\{n\}$ vertices connected by $\{len(edges)\}$ edges
- Edges: $\{edge\_desc\}$
- You control 3 bees
- I control Nastya
- Each vertex connects to at most 3 others
- Each edge is part of a cycle of length $\leq 5$
2. Game Mechanics:
- First round:
* You place 3 bees on any vertices
* I place Nastya on a different vertex
- Each subsequent round:
* You move each bee (or keep in place)
* I move Nastya along one edge
- Movement rules:
* Can only move along edges
* Multiple bees can share same vertex
* Nastya must move each turn
* All moves must be valid graph moves
3. Victory Conditions:
- You win if any bee reaches same vertex as Nastya
- You lose if not caught after $\{n\}$ moves
- Game ends immediately upon catch

Query Type:
Format: "My Choice: $X$ $Y$ $Z$"
where $X, Y, Z$ are vertex numbers for three bees
Example Round:
Initial placement:
You: "My Choice: 1 2 3"
- Placing bees at vertices 1,2,3
Me: "5"
- Nastya appears at vertex 5
You: "My Choice: 2 3 4"
- Moving bees to surround Nastya
Me: "6"
- Nastya moves to vertex 6
Result: You catch Nastya!
Instructions:
1. Make moves based on graph structure
2. Use exactly the format shown above
3. Explain your reasoning before each move
Remember:
- Use exact format: "My Choice: $X$ $Y$ $Z$"
- Choose only valid vertex numbers
- Plan moves to trap Nastya
- Invalid move = immediate loss
- Maximum $\{n\}$ moves to win
Ready to start? Make your first query!

## Case N.78: BeeChase Difficulty Levels

Easy: $n = 10$, Medium: $n = 20$, Hard: $n = 40$

**AssiutChess**   In this task, models need to trap a hidden king using a queen on a chessboard. Success requires strategic queen placement and movement while responding to the king's reported directions.

## Case N.79: AssiutChess Problem Template

Let's play Assiut Chess! Your task is to win this game by controlling a queen to trap the hidden king.
Rules:
1. Game Setup:
- $\{n\}*\{n\}$ chessboard (rows and columns from 1 to $\{n\}$)
- You control the queen, I control the hidden king
- First, you place the queen anywhere on the board
2. Game Mechanics:
- On each turn:
* I move the king first (in one of 8 directions)
* I tell you which direction the king moved
* You move the queen to any cell in straight or diagonal line
- King's possible moves:
* "Right", "Left", "Up", "Down"
* "Down-Right", "Down-Left", "Up-Left", "Up-Right"
- King's restrictions:
* Cannot move out of the board
* Cannot move to cells attacked by queen (same row, column, or diagonal)
- Queen's restrictions:
* Must move to a different cell each turn
* Must move in straight or diagonal lines
3. Victory Conditions:
- You win if the king has no valid moves

- Game ends when "Done" is received
Query Type:
Format: "My Choice: $x\ y$"
where $1 \le x, y \le \{n\}$
Example Round:
Initial queen placement:
You: "My Choice: 3 2"
Me: "Left"
You: "My Choice: 3 3"
Me: "Right"
You: "My Choice: 3 4"
Me: "Done"
Result: You win! King is trapped!
Instructions:
1. Make moves based on king's direction
2. Use exactly the format shown above
3. Explain your reasoning before each move
Remember:
- Use exact format: "My Choice: $x\ y$"
- Choose valid queen moves only
- Plan moves to trap the king
- Invalid move = immediate loss
- You have maximun 20 moves
 Ready to start? Make your first query!

**Case N.80: AssiutChess Difficulty Levels**

Easy: $n = 4$, Medium: $n = 6$, Hard: $n = 7$

## O  PER-TASK RESULTS

We list the experimental results for each of the 40 tasks in this section.

| Model | Easy | Medium | Hard |
|---|---|---|---|
| o3-mini | 93.33 | 73.33 | 60.00 |
| R1 | 96.67 | 56.67 | 50.00 |
| QwQ-32B | 86.67 | 46.67 | 20.00 |
| R1-Distill-Llama-70B | 90.00 | 46.67 | 33.33 |
| R1-Distill-Qwen-32B | 40.00 | 10.00 | 6.67 |
| R1-Distill-Qwen-7B | 0.00 | 0.00 | 0.00 |
| R1-Distill-Qwen-1.5B | 0.00 | 0.00 | 0.00 |
| GPT-4o | 83.33 | 43.33 | 50.00 |
| Qwen-Max | 93.33 | 80.00 | 43.33 |
| gemma-3-27b-IT | 50.00 | 3.33 | 6.67 |
| gemma-3-12b-IT | 26.67 | 0.00 | 0.00 |
| gemma-3-4b-IT | 46.67 | 0.00 | 3.33 |
| Qwen2.5-72B-IT | 93.33 | 66.67 | 50.00 |
| Qwen2.5-32B-IT | 90.00 | 73.33 | 56.67 |
| Qwen2.5-7B-IT | 66.67 | 46.67 | 36.67 |
| Qwen2.5-1.5B-IT | 0.00 | 0.00 | 0.00 |
| Llama-3.1-70B-IT | 90.00 | 56.67 | 23.33 |
| Llama-3.1-8B-IT | 40.00 | 13.33 | 3.33 |
| Mistral-Small-24B-IT-2501 | 70.00 | 10.00 | 3.33 |
| Ministral-8B-IT-2410 | 63.33 | 10.00 | 0.00 |

Table 8: Average accuracy for AssiutChess across different difficulties.

| Model | Easy | Medium | Hard |
|---|---|---|---|
| o3-mini | 42.22 | 26.67 | 15.56 |
| R1 | 52.22 | 37.78 | 28.89 |
| QwQ-32B | 50.00 | 42.22 | 25.56 |
| R1-Distill-Llama-70B | 37.78 | 13.33 | 14.44 |
| R1-Distill-Qwen-32B | 14.44 | 7.78 | 3.33 |
| R1-Distill-Qwen-7B | 0.00 | 0.00 | 0.00 |
| R1-Distill-Qwen-1.5B | 0.00 | 0.00 | 0.00 |
| GPT-4o | 31.11 | 13.33 | 10.00 |
| Qwen-Max | 27.78 | 12.22 | 2.22 |
| gemma-3-27b-IT | 24.44 | 15.56 | 10.00 |
| gemma-3-12b-IT | 16.67 | 7.78 | 4.44 |
| gemma-3-4b-IT | 34.44 | 8.89 | 7.78 |
| Qwen2.5-72B-IT | 30.00 | 11.11 | 3.33 |
| Qwen2.5-32B-IT | 11.11 | 7.78 | 2.22 |
| Qwen2.5-7B-IT | 24.44 | 13.33 | 3.33 |
| Qwen2.5-1.5B-IT | 2.22 | 1.11 | 0.00 |
| Llama-3.1-70B-IT | 27.78 | 13.33 | 10.00 |
| Llama-3.1-8B-IT | 22.22 | 5.56 | 7.78 |
| Mistral-Small-24B-IT-2501 | 33.33 | 6.67 | 5.56 |
| Ministral-8B-IT-2410 | 13.33 | 2.22 | 0.00 |

Table 9: Average accuracy for AttendanceCheck across different difficulties.

| Model | Easy | Medium | Hard |
|---|---|---|---|
| o3-mini | 100.00 | 100.00 | 86.67 |
| R1 | 100.00 | 86.67 | 86.67 |
| QwQ-32B | 90.00 | 56.67 | 66.67 |
| R1-Distill-Llama-70B | 56.67 | 46.67 | 30.00 |
| R1-Distill-Qwen-32B | 73.33 | 40.00 | 6.67 |
| R1-Distill-Qwen-7B | 3.33 | 0.00 | 0.00 |
| R1-Distill-Qwen-1.5B | 0.00 | 0.00 | 0.00 |
| GPT-4o | 3.33 | 0.00 | 0.00 |
| Qwen-M a z | 43.33 | 36.67 | 30.00 |
| gemma-3-27b-IT | 30.00 | 3.33 | 3.33 |
| gemma-3-12b-IT | 3.33 | 0.00 | 0.00 |
| gemma-3-4b-IT | 6.67 | 0.00 | 0.00 |
| Qwen2.5-72B-IT | 26.67 | 6.67 | 3.33 |
| Qwen2.5-32B-IT | 23.33 | 26.67 | 0.00 |
| Qwen2.5-7B-IT | 16.67 | 6.67 | 3.33 |
| Qwen2.5-1.5B-IT | 53.33 | 70.00 | 53.33 |
| Llama-3.1-70B-IT | 23.33 | 3.33 | 6.67 |
| Llama-3.1-8B-IT | 13.33 | 0.00 | 0.00 |
| Mistral-Small-24B-IT-2501 | 30.00 | 3.33 | 0.00 |
| Ministral-8B-IT-2410 | 20.00 | 10.00 | 0.00 |

Table 10: Average accuracy for BeeChase across different difficulties.

| Model | Easy | Medium | Hard |
|---|---|---|---|
| o3-mini | 60.00 | 42.22 | 23.33 |
| R1 | 67.78 | 60.00 | 20.00 |
| QwQ-32B | 95.56 | 97.78 | 74.44 |
| R1-Distill-Llama-70B | 42.22 | 38.89 | 11.11 |
| R1-Distill-Qwen-32B | 60.00 | 45.56 | 15.56 |
| R1-Distill-Qwen-7B | 25.56 | 18.89 | 4.44 |
| R1-Distill-Qwen-1.5B | 0.00 | 6.67 | 2.22 |
| GPT-4o | 21.11 | 10.00 | 13.33 |
| Qwen-Max | 42.22 | 31.11 | 17.78 |
| gemma-3-27b-IT | 52.22 | 38.89 | 28.89 |
| gemma-3-12b-IT | 21.11 | 10.00 | 4.44 |
| gemma-3-4b-IT | 16.67 | 10.00 | 6.67 |
| Qwen2.5-72B-IT | 52.22 | 58.89 | 16.67 |
| Qwen2.5-32B-IT | 55.56 | 42.22 | 15.56 |
| Qwen2.5-7B-IT | 76.67 | 65.56 | 11.11 |
| Qwen2.5-1.5B-IT | 6.67 | 1.11 | 0.00 |
| Llama-3.1-70B-IT | 73.33 | 53.33 | 30.00 |
| Llama-3.1-8B-IT | 60.00 | 30.00 | 5.56 |
| Mistral-Small-24B-IT-2501 | 22.22 | 18.89 | 8.89 |
| Ministral-8B-IT-2410 | 3.33 | 15.56 | 0.00 |

Table 11: Average accuracy for BitCompare across different difficulties.

| Model | Easy | Medium | Hard |
|---|---|---|---|
| o3-mini | 78.89 | 54.44 | 40.00 |
| R1 | 77.78 | 41.11 | 24.44 |
| QwQ-32B | 32.22 | 13.33 | 10.00 |
| R1-Distill-Llama-70B | 13.33 | 1.11 | 0.00 |
| R1-Distill-Qwen-32B | 8.89 | 0.00 | 0.00 |
| R1-Distill-Qwen-7B | 0.00 | 0.00 | 0.00 |
| R1-Distill-Qwen-1.5B | 0.00 | 0.00 | 0.00 |
| GPT-4o | 25.56 | 12.22 | 1.11 |
| Qwen-Max | 31.11 | 17.78 | 6.67 |
| gemma-3-27b-IT | 32.22 | 5.56 | 2.22 |
| gemma-3-12b-IT | 7.78 | 3.33 | 1.11 |
| gemma-3-4b-IT | 0.00 | 1.11 | 0.00 |
| Qwen2.5-72B-IT | 31.11 | 6.67 | 2.22 |
| Qwen2.5-32B-IT | 7.78 | 2.22 | 1.11 |
| Qwen2.5-7B-IT | 4.44 | 1.11 | 3.33 |
| Qwen2.5-1.5B-IT | 0.00 | 0.00 | 0.00 |
| Llama-3.1-70B-IT | 18.89 | 12.22 | 3.33 |
| Llama-3.1-8B-IT | 7.78 | 1.11 | 0.00 |
| Mistral-Small-24B-IT-2501 | 12.22 | 2.22 | 0.00 |
| Ministral-8B-IT-2410 | 7.78 | 2.22 | 0.00 |

Table 12: Average accuracy for BitGuessing across different difficulties.

| Model | Easy | Medium | Hard |
|---|---|---|---|
| o3-mini | 26.67 | 25.56 | 16.67 |
| R1 | 100.00 | 100.00 | 100.00 |
| QwQ-32B | 93.33 | 96.67 | 94.44 |
| R1-Distill-Llama-70B | 18.89 | 15.56 | 15.56 |
| R1-Distill-Qwen-32B | 7.78 | 4.44 | 5.56 |
| R1-Distill-Qwen-7B | 5.56 | 3.33 | 2.22 |
| R1-Distill-Qwen-1.5B | 0.00 | 0.00 | 0.00 |
| GPT-4o | 23.33 | 24.44 | 23.33 |
| Qwen-Max | 22.22 | 22.22 | 15.56 |
| gemma-3-27b-IT | 21.11 | 32.22 | 10.00 |
| gemma-3-12b-IT | 45.56 | 25.56 | 20.00 |
| gemma-3-4b-IT | 30.00 | 18.89 | 13.33 |
| Qwen2.5-72B-IT | 11.11 | 7.78 | 14.44 |
| Qwen2.5-32B-IT | 66.67 | 67.78 | 60.00 |
| Qwen2.5-7B-IT | 24.44 | 21.11 | 28.89 |
| Qwen2.5-1.5B-IT | 15.56 | 6.67 | 4.44 |
| Llama-3.1-70B-IT | 50.00 | 48.89 | 53.33 |
| Llama-3.1-8B-IT | 35.56 | 23.33 | 22.22 |
| Mistral-Small-24B-IT-2501 | 48.89 | 16.67 | 23.33 |
| Ministral-8B-IT-2410 | 72.22 | 55.56 | 20.00 |

Table 13: Average accuracy for CactusSearch across different difficulties.

| Model | Easy | Medium | Hard |
|---|---|---|---|
| o3-mini | 34.44 | 25.56 | 7.78 |
| R1 | 6.67 | 11.11 | 6.67 |
| QwQ-32B | 2.22 | 3.33 | 1.11 |
| R1-Distill-Llama-70B | 3.33 | 1.11 | 1.11 |
| R1-Distill-Qwen-32B | 8.89 | 3.33 | 1.11 |
| R1-Distill-Qwen-7B | 0.00 | 0.00 | 0.00 |
| R1-Distill-Qwen-1.5B | 0.00 | 0.00 | 0.00 |
| GPT-4o | 0.00 | 0.00 | 1.11 |
| Qwen-Max | 4.44 | 2.22 | 3.33 |
| gemma-3-27b-IT | 3.33 | 1.11 | 0.00 |
| gemma-3-12b-IT | 0.00 | 0.00 | 0.00 |
| gemma-3-4b-IT | 0.00 | 0.00 | 0.00 |
| Qwen2.5-72B-IT | 2.22 | 2.22 | 1.11 |
| Qwen2.5-32B-IT | 5.56 | 1.11 | 0.00 |
| Qwen2.5-7B-IT | 1.11 | 0.00 | 0.00 |
| Qwen2.5-1.5B-IT | 0.00 | 0.00 | 0.00 |
| Llama-3.1-70B-IT | 0.00 | 0.00 | 0.00 |
| Llama-3.1-8B-IT | 0.00 | 0.00 | 0.00 |
| Mistral-Small-24B-IT-2501 | 1.11 | 0.00 | 0.00 |
| Ministral-8B-IT-2410 | 0.00 | 0.00 | 0.00 |

Table 14: Average accuracy for ChemicalSynthesis across different difficulties.

| Model | Easy | Medium | Hard |
|---|---|---|---|
| o3-mini | 25.00 | 4.44 | 0.00 |
| R1 | 24.44 | 3.33 | 2.22 |
| QwQ-32B | 7.78 | 0.00 | 0.00 |
| R1-Distill-Llama-70B | 1.11 | 0.00 | 0.00 |
| R1-Distill-Qwen-32B | 2.22 | 0.00 | 0.00 |
| R1-Distill-Qwen-7B | 0.00 | 0.00 | 0.00 |
| R1-Distill-Qwen-1.5B | 0.00 | 0.00 | 0.00 |
| GPT-4o | 1.11 | 0.00 | 0.00 |
| Qwen-Max | 1.11 | 0.00 | 0.00 |
| gemma-3-27b-IT | 1.11 | 0.00 | 0.00 |
| gemma-3-12b-IT | 0.00 | 0.00 | 0.00 |
| gemma-3-4b-IT | 1.11 | 0.00 | 0.00 |
| Qwen2.5-72B-IT | 0.00 | 0.00 | 0.00 |
| Qwen2.5-32B-IT | 8.89 | 0.00 | 0.00 |
| Qwen2.5-7B-IT | 0.00 | 0.00 | 0.00 |
| Qwen2.5-1.5B-IT | 0.00 | 0.00 | 0.00 |
| Llama-3.1-70B-IT | 2.22 | 0.00 | 0.00 |
| Llama-3.1-8B-IT | 3.33 | 0.00 | 0.00 |
| Mistral-Small-24B-IT-2501 | 0.00 | 0.00 | 0.00 |
| Ministral-8B-IT-2410 | 0.00 | 0.00 | 0.00 |

Table 15: Average accuracy for ColorMagic across different difficulties.

| Model | Easy | Medium | Hard |
|---|---|---|---|
| o3-mini | 78.89 | 46.67 | 26.67 |
| R1 | 80.00 | 45.56 | 15.56 |
| QwQ-32B | 62.22 | 34.44 | 8.89 |
| R1-Distill-Llama-70B | 24.44 | 2.22 | 3.33 |
| R1-Distill-Qwen-32B | 11.11 | 2.22 | 1.11 |
| R1-Distill-Qwen-7B | 0.00 | 2.22 | 0.00 |
| R1-Distill-Qwen-1.5B | 0.00 | 0.00 | 0.00 |
| GPT-4o | 81.11 | 51.11 | 13.33 |
| Qwen-Max | 70.00 | 42.22 | 15.56 |
| gemma-3-27b-IT | 6.67 | 2.22 | 0.00 |
| gemma-3-12b-IT | 11.11 | 0.00 | 0.00 |
| gemma-3-4b-IT | 13.33 | 11.11 | 0.00 |
| Qwen2.5-72B-IT | 13.33 | 2.22 | 0.00 |
| Qwen2.5-32B-IT | 13.33 | 6.67 | 2.22 |
| Qwen2.5-7B-IT | 13.33 | 11.11 | 0.00 |
| Qwen2.5-1.5B-IT | 50.00 | 0.00 | 0.00 |
| Llama-3.1-70B-IT | 7.78 | 0.00 | 0.00 |
| Llama-3.1-8B-IT | 26.67 | 1.11 | 1.11 |
| Mistral-Small-24B-IT-2501 | 37.78 | 3.33 | 1.11 |
| Ministral-8B-IT-2410 | 33.33 | 20.00 | 2.22 |

Table 16: Average accuracy for DarkMaze across different difficulties.

| Model | Easy | Medium | Hard |
|---|---|---|---|
| o3-mini | 11.11 | 10.00 | 10.00 |
| R1 | 54.44 | 47.78 | 38.89 |
| QwQ-32B | 32.22 | 36.67 | 14.44 |
| R1-Distill-Llama-70B | 4.44 | 16.67 | 1.11 |
| R1-Distill-Qwen-32B | 1.11 | 1.11 | 3.33 |
| R1-Distill-Qwen-7B | 0.00 | 0.00 | 0.00 |
| R1-Distill-Qwen-1.5B | 0.00 | 0.00 | 0.00 |
| GPT-4o | 5.56 | 3.33 | 2.22 |
| Qwen-Max | 4.44 | 10.00 | 2.22 |
| gemma-3-27b-IT | 4.44 | 7.78 | 2.22 |
| gemma-3-12b-IT | 4.44 | 6.67 | 0.00 |
| gemma-3-4b-IT | 0.00 | 0.00 | 0.00 |
| Qwen2.5-72B-IT | 11.11 | 13.33 | 6.67 |
| Qwen2.5-32B-IT | 4.44 | 2.22 | 1.11 |
| Qwen2.5-7B-IT | 7.78 | 10.00 | 2.22 |
| Qwen2.5-1.5B-IT | 0.00 | 0.00 | 0.00 |
| Llama-3.1-70B-IT | 12.22 | 13.33 | 2.22 |
| Llama-3.1-8B-IT | 7.78 | 6.67 | 1.11 |
| Mistral-Small-24B-IT-2501 | 16.67 | 17.78 | 4.44 |
| Ministral-8B-IT-2410 | 11.11 | 28.89 | 10.00 |

Table 17: Average accuracy for FindBiggest across different difficulties.

| Model | Easy | Medium | Hard |
|---|---|---|---|
| o3-mini | 82.00 | 6.67 | 16.67 |
| R1 | 74.00 | 3.33 | 2.22 |
| QwQ-32B | 75.33 | 6.67 | 5.56 |
| R1-Distill-Llama-70B | 54.44 | 0.00 | 0.00 |
| R1-Distill-Qwen-32B | 24.44 | 0.00 | 0.00 |
| R1-Distill-Qwen-7B | 0.00 | 0.00 | 0.00 |
| R1-Distill-Qwen-1.5B | 0.00 | 0.00 | 0.00 |
| GPT-4o | 49.33 | 2.22 | 4.44 |
| Qwen-Max | 47.33 | 0.00 | 5.56 |
| gemma-3-27b-IT | 4.44 | 1.11 | 0.00 |
| gemma-3-12b-IT | 4.44 | 0.00 | 0.00 |
| gemma-3-4b-IT | 26.67 | 0.00 | 0.00 |
| Qwen2.5-72B-IT | 45.56 | 0.00 | 0.00 |
| Qwen2.5-32B-IT | 48.89 | 0.00 | 0.00 |
| Qwen2.5-7B-IT | 37.78 | 0.00 | 0.00 |
| Qwen2.5-1.5B-IT | 0.00 | 0.00 | 0.00 |
| Llama-3.1-70B-IT | 33.33 | 0.00 | 0.00 |
| Llama-3.1-8B-IT | 12.22 | 0.00 | 0.00 |
| Mistral-Small-24B-IT-2501 | 1.11 | 0.00 | 0.00 |
| Ministral-8B-IT-2410 | 2.22 | 0.00 | 0.00 |

Table 18: Average accuracy for BitQuery across different difficulties.

| Model | Easy | Medium | Hard |
|---|---|---|---|
| o3-mini | 66.67 | 51.11 | 57.78 |
| R1 | 25.56 | 10.00 | 11.11 |
| QwQ-32B | 26.67 | 20.00 | 15.56 |
| R1-Distill-Llama-70B | 16.67 | 5.56 | 1.11 |
| R1-Distill-Qwen-32B | 11.11 | 2.22 | 0.00 |
| R1-Distill-Qwen-7B | 0.00 | 0.00 | 0.00 |
| R1-Distill-Qwen-1.5B | 0.00 | 0.00 | 0.00 |
| GPT-4o | 0.00 | 0.00 | 0.00 |
| Qwen-Max | 0.00 | 0.00 | 0.00 |
| gemma-3-27b-IT | 0.00 | 0.00 | 0.00 |
| gemma-3-12b-IT | 0.00 | 0.00 | 0.00 |
| gemma-3-4b-IT | 0.00 | 0.00 | 0.00 |
| Qwen2.5-72B-IT | 1.11 | 0.00 | 0.00 |
| Qwen2.5-32B-IT | 0.00 | 0.00 | 0.00 |
| Qwen2.5-7B-IT | 0.00 | 0.00 | 0.00 |
| Qwen2.5-1.5B-IT | 0.00 | 0.00 | 0.00 |
| Llama-3.1-70B-IT | 0.00 | 0.00 | 0.00 |
| Llama-3.1-8B-IT | 0.00 | 0.00 | 0.00 |
| Mistral-Small-24B-IT-2501 | 0.00 | 0.00 | 0.00 |
| Ministral-8B-IT-2410 | 0.00 | 0.00 | 0.00 |

Table 19: Average accuracy for CircleFinding across different difficulties.

| Model | Easy | Medium | Hard |
|---|---|---|---|
| o3-mini | 32.22 | 22.22 | 21.11 |
| R1 | 42.22 | 37.78 | 18.89 |
| QwQ-32B | 24.44 | 20.00 | 15.56 |
| R1-Distill-Llama-70B | 22.22 | 21.11 | 14.44 |
| R1-Distill-Qwen-32B | 7.78 | 4.44 | 2.22 |
| R1-Distill-Qwen-7B | 0.00 | 0.00 | 0.00 |
| R1-Distill-Qwen-1.5B | 0.00 | 0.00 | 0.00 |
| GPT-4o | 36.67 | 14.44 | 12.22 |
| Qwen-Max | 17.78 | 18.89 | 5.56 |
| gemma-3-27b-IT | 12.22 | 13.33 | 12.22 |
| gemma-3-12b-IT | 15.56 | 12.22 | 8.89 |
| gemma-3-4b-IT | 2.22 | 2.22 | 0.00 |
| Qwen2.5-72B-IT | 26.67 | 20.00 | 12.22 |
| Qwen2.5-32B-IT | 26.67 | 7.78 | 13.33 |
| Qwen2.5-7B-IT | 11.11 | 18.89 | 11.11 |
| Qwen2.5-1.5B-IT | 0.00 | 1.11 | 0.00 |
| Llama-3.1-70B-IT | 32.22 | 14.44 | 5.56 |
| Llama-3.1-8B-IT | 18.89 | 8.89 | 10.00 |
| Mistral-Small-24B-IT-2501 | 16.67 | 13.33 | 11.11 |
| Ministral-8B-IT-2410 | 15.56 | 8.89 | 6.67 |

Table 20: Average accuracy for FindHidden across different difficulties.

| Model | Easy | Medium | Hard |
|---|---|---|---|
| o3-mini | 61.11 | 42.22 | 14.44 |
| R1 | 62.22 | 32.22 | 5.56 |
| QwQ-32B | 71.11 | 51.11 | 21.11 |
| R1-Distill-Llama-70B | 41.11 | 10.00 | 2.22 |
| R1-Distill-Qwen-32B | 27.78 | 3.33 | 1.11 |
| R1-Distill-Qwen-7B | 2.22 | 0.00 | 0.00 |
| R1-Distill-Qwen-1.5B | 0.00 | 0.00 | 0.00 |
| GPT-4o | 31.11 | 8.89 | 0.00 |
| Qwen-Max | 36.67 | 13.33 | 1.11 |
| gemma-3-27b-IT | 55.56 | 20.00 | 4.44 |
| gemma-3-12b-IT | 51.11 | 18.89 | 6.67 |
| gemma-3-4b-IT | 2.22 | 0.00 | 0.00 |
| Qwen2.5-72B-IT | 53.33 | 11.11 | 4.44 |
| Qwen2.5-32B-IT | 38.89 | 21.11 | 1.11 |
| Qwen2.5-7B-IT | 25.56 | 7.78 | 1.11 |
| Qwen2.5-1.5B-IT | 0.00 | 0.00 | 0.00 |
| Llama-3.1-70B-IT | 58.89 | 14.44 | 0.00 |
| Llama-3.1-8B-IT | 10.00 | 0.00 | 0.00 |
| Mistral-Small-24B-IT-2501 | 40.00 | 10.00 | 0.00 |
| Ministral-8B-IT-2410 | 11.11 | 5.56 | 0.00 |

Table 21: Average accuracy for FindTheImpostors across different difficulties.

| Model | Easy | Medium | Hard |
|---|---|---|---|
| o3-mini | 72.17 | 46.83 | 25.83 |
| R1 | 70.00 | 53.33 | 53.33 |
| QwQ-32B | 6.67 | 3.33 | 0.00 |
| R1-Distill-Llama-70B | 61.11 | 25.56 | 14.44 |
| R1-Distill-Qwen-32B | 24.44 | 15.56 | 6.67 |
| R1-Distill-Qwen-7B | 0.00 | 0.00 | 0.00 |
| R1-Distill-Qwen-1.5B | 0.00 | 0.00 | 0.00 |
| GPT-4o | 35.56 | 33.33 | 25.56 |
| Qwen-Max | 11.11 | 17.78 | 18.89 |
| gemma-3-27b-IT | 23.11 | 11.11 | 18.33 |
| gemma-3-12b-IT | 32.67 | 32.22 | 20.67 |
| gemma-3-4b-IT | 23.33 | 3.33 | 0.00 |
| Qwen2.5-72B-IT | 46.00 | 46.67 | 43.33 |
| Qwen2.5-32B-IT | 45.78 | 31.11 | 28.00 |
| Qwen2.5-7B-IT | 33.33 | 20.00 | 13.33 |
| Qwen2.5-1.5B-IT | 70.00 | 46.67 | 56.67 |
| Llama-3.1-70B-IT | 49.33 | 41.33 | 41.78 |
| Llama-3.1-8B-IT | 36.67 | 30.00 | 23.33 |
| Mistral-Small-24B-IT-2501 | 41.11 | 36.89 | 46.89 |
| Ministral-8B-IT-2410 | 70.00 | 3.33 | 33.33 |

Table 22: Average accuracy for GeoGame across different difficulties.

| Model | Easy | Medium | Hard |
|---|---|---|---|
| o3-mini | 100.00 | 96.67 | 100.00 |
| R1 | 90.00 | 96.67 | 100.00 |
| QwQ-32B | 83.33 | 83.33 | 76.67 |
| R1-Distill-Llama-70B | 26.67 | 60.00 | 46.67 |
| R1-Distill-Qwen-32B | 6.67 | 3.33 | 0.00 |
| R1-Distill-Qwen-7B | 0.00 | 0.00 | 0.00 |
| R1-Distill-Qwen-1.5B | 0.00 | 0.00 | 0.00 |
| GPT-4o | 60.00 | 60.00 | 43.33 |
| Qwen-Max | 73.33 | 66.67 | 50.00 |
| gemma-3-27b-IT | 0.00 | 0.00 | 3.33 |
| gemma-3-12b-IT | 0.00 | 3.33 | 0.00 |
| gemma-3-4b-IT | 0.00 | 0.00 | 0.00 |
| Qwen2.5-72B-IT | 33.33 | 46.67 | 26.67 |
| Qwen2.5-32B-IT | 26.67 | 26.67 | 6.67 |
| Qwen2.5-7B-IT | 30.00 | 26.67 | 20.00 |
| Qwen2.5-1.5B-IT | 0.00 | 0.00 | 0.00 |
| Llama-3.1-70B-IT | 50.00 | 46.67 | 23.33 |
| Llama-3.1-8B-IT | 0.00 | 0.00 | 0.00 |
| Mistral-Small-24B-IT-2501 | 0.00 | 0.00 | 0.00 |
| Ministral-8B-IT-2410 | 0.00 | 0.00 | 0.00 |

Table 23: Average accuracy for GridColoring across different difficulties.

| Model | Easy | Medium | Hard |
|---|---|---|---|
| o3-mini | 93.33 | 96.67 | 80.00 |
| R1 | 80.00 | 83.33 | 46.67 |
| QwQ-32B | 83.33 | 73.33 | 33.33 |
| R1-Distill-Llama-70B | 83.33 | 56.67 | 26.67 |
| R1-Distill-Qwen-32B | 86.67 | 60.00 | 20.00 |
| R1-Distill-Qwen-7B | 0.00 | 0.00 | 0.00 |
| R1-Distill-Qwen-1.5B | 0.00 | 0.00 | 0.00 |
| GPT-4o | 70.00 | 46.67 | 40.00 |
| Qwen-Max | 93.33 | 46.67 | 30.00 |
| gemma-3-27b-IT | 0.00 | 0.00 | 0.00 |
| gemma-3-12b-IT | 0.00 | 0.00 | 0.00 |
| gemma-3-4b-IT | 3.33 | 0.00 | 0.00 |
| Qwen2.5-72B-IT | 43.33 | 60.00 | 63.33 |
| Qwen2.5-32B-IT | 76.67 | 43.33 | 40.00 |
| Qwen2.5-7B-IT | 23.33 | 0.00 | 3.33 |
| Qwen2.5-1.5B-IT | 16.67 | 3.33 | 0.00 |
| Llama-3.1-70B-IT | 20.00 | 13.33 | 0.00 |
| Llama-3.1-8B-IT | 0.00 | 0.00 | 0.00 |
| Mistral-Small-24B-IT-2501 | 0.00 | 0.00 | 0.00 |
| Ministral-8B-IT-2410 | 16.67 | 26.67 | 53.33 |

Table 24: Average accuracy for GridGame across different difficulties.

| Model | Easy | Medium | Hard |
|---|---|---|---|
| o3-mini | 60.00 | 55.56 | 55.56 |
| R1 | 14.44 | 2.22 | 7.78 |
| QwQ-32B | 3.33 | 2.22 | 2.22 |
| R1-Distill-Llama-70B | 4.44 | 1.11 | 3.33 |
| R1-Distill-Qwen-32B | 0.00 | 1.11 | 0.00 |
| R1-Distill-Qwen-7B | 0.00 | 0.00 | 0.00 |
| R1-Distill-Qwen-1.5B | 0.00 | 0.00 | 0.00 |
| GPT-4o | 27.78 | 10.00 | 1.11 |
| Qwen-Max | 28.89 | 12.22 | 5.56 |
| gemma-3-27b-IT | 1.11 | 2.22 | 4.44 |
| gemma-3-12b-IT | 6.67 | 1.11 | 0.00 |
| gemma-3-4b-IT | 0.00 | 0.00 | 0.00 |
| Qwen2.5-72B-IT | 46.67 | 22.22 | 6.67 |
| Qwen2.5-32B-IT | 33.33 | 7.78 | 6.67 |
| Qwen2.5-7B-IT | 1.11 | 0.00 | 0.00 |
| Qwen2.5-1.5B-IT | 0.00 | 0.00 | 0.00 |
| Llama-3.1-70B-IT | 24.44 | 8.89 | 5.56 |
| Llama-3.1-8B-IT | 0.00 | 0.00 | 0.00 |
| Mistral-Small-24B-IT-2501 | 3.33 | 1.11 | 0.00 |
| Ministral-8B-IT-2410 | 0.00 | 0.00 | 0.00 |

Table 25: Average accuracy for GuessMax across different difficulties.

| Model | Easy | Medium | Hard |
|---|---|---|---|
| o3-mini | 93.33 | 93.33 | 93.33 |
| R1 | 90.00 | 100.00 | 100.00 |
| QwQ-32B | 90.00 | 86.67 | 90.00 |
| R1-Distill-Llama-70B | 73.33 | 86.67 | 80.00 |
| R1-Distill-Qwen-32B | 63.33 | 60.00 | 60.00 |
| R1-Distill-Qwen-7B | 20.00 | 20.00 | 6.67 |
| R1-Distill-Qwen-1.5B | 3.33 | 6.67 | 0.00 |
| GPT-4o | 16.67 | 10.00 | 16.67 |
| Qwen-Max | 26.67 | 23.33 | 13.33 |
| gemma-3-27b-IT | 6.67 | 13.33 | 3.33 |
| gemma-3-12b-IT | 3.33 | 3.33 | 0.00 |
| gemma-3-4b-IT | 0.00 | 13.33 | 0.00 |
| Qwen2.5-72B-IT | 13.33 | 23.33 | 30.00 |
| Qwen2.5-32B-IT | 3.33 | 20.00 | 3.33 |
| Qwen2.5-7B-IT | 0.00 | 0.00 | 0.00 |
| Qwen2.5-1.5B-IT | 0.00 | 13.33 | 0.00 |
| Llama-3.1-70B-IT | 0.00 | 6.67 | 0.00 |
| Llama-3.1-8B-IT | 0.00 | 3.33 | 3.33 |
| Mistral-Small-24B-IT-2501 | 0.00 | 0.00 | 0.00 |
| Ministral-8B-IT-2410 | 0.00 | 0.00 | 0.00 |

Table 26: Average accuracy for KnightBattle across different difficulties.

| Model | Easy | Medium | Hard |
|---|---|---|---|
| o3-mini | 24.44 | 7.78 | 0.00 |
| R1 | 5.56 | 0.00 | 0.00 |
| QwQ-32B | 12.22 | 0.00 | 0.00 |
| R1-Distill-Llama-70B | 6.67 | 0.00 | 0.00 |
| R1-Distill-Qwen-32B | 2.22 | 0.00 | 0.00 |
| R1-Distill-Qwen-7B | 0.00 | 0.00 | 0.00 |
| R1-Distill-Qwen-1.5B | 0.00 | 0.00 | 0.00 |
| GPT-4o | 0.00 | 0.00 | 0.00 |
| Qwen-Max | 0.00 | 0.00 | 0.00 |
| gemma-3-27b-IT | 2.22 | 0.00 | 0.00 |
| gemma-3-12b-IT | 0.00 | 0.00 | 0.00 |
| gemma-3-4b-IT | 0.00 | 0.00 | 0.00 |
| Qwen2.5-72B-IT | 2.22 | 0.00 | 0.00 |
| Qwen2.5-32B-IT | 1.11 | 0.00 | 0.00 |
| Qwen2.5-7B-IT | 0.00 | 0.00 | 0.00 |
| Qwen2.5-1.5B-IT | 0.00 | 0.00 | 0.00 |
| Llama-3.1-70B-IT | 0.00 | 0.00 | 0.00 |
| Llama-3.1-8B-IT | 0.00 | 0.00 | 0.00 |
| Mistral-Small-24B-IT-2501 | 0.00 | 0.00 | 0.00 |
| Ministral-8B-IT-2410 | 0.00 | 0.00 | 0.00 |

Table 27: Average accuracy for LegendaryTree across different difficulties.

| Model | Easy | Medium | Hard |
|---|---|---|---|
| o3-mini | 76.67 | 61.11 | 38.89 |
| R1 | 80.00 | 67.78 | 51.11 |
| QwQ-32B | 92.22 | 68.89 | 50.00 |
| R1-Distill-Llama-70B | 86.67 | 50.00 | 32.22 |
| R1-Distill-Qwen-32B | 84.44 | 36.67 | 14.44 |
| R1-Distill-Qwen-7B | 2.22 | 2.22 | 5.56 |
| R1-Distill-Qwen-1.5B | 0.00 | 1.11 | 1.11 |
| GPT-4o | 70.00 | 42.22 | 30.00 |
| Qwen-Max | 72.22 | 50.00 | 38.89 |
| gemma-3-27b-IT | 78.89 | 20.00 | 50.00 |
| gemma-3-12b-IT | 75.56 | 44.44 | 32.22 |
| gemma-3-4b-IT | 63.33 | 30.00 | 15.56 |
| Qwen2.5-72B-IT | 75.56 | 57.78 | 44.44 |
| Qwen2.5-32B-IT | 87.78 | 66.67 | 67.78 |
| Qwen2.5-7B-IT | 54.44 | 30.00 | 22.22 |
| Qwen2.5-1.5B-IT | 8.89 | 0.00 | 0.00 |
| Llama-3.1-70B-IT | 77.78 | 65.56 | 60.00 |
| Llama-3.1-8B-IT | 80.00 | 52.22 | 32.22 |
| Mistral-Small-24B-IT-2501 | 70.00 | 44.44 | 36.67 |
| Ministral-8B-IT-2410 | 57.78 | 21.11 | 20.00 |

Table 28: Average accuracy for ListQuery across different difficulties.

| Model | Easy | Medium | Hard |
|---|---|---|---|
| o3-mini | 20.00 | 23.33 | 16.67 |
| R1 | 17.78 | 28.89 | 21.11 |
| QwQ-32B | 2.22 | 0.00 | 0.00 |
| R1-Distill-Llama-70B | 0.00 | 0.00 | 0.00 |
| R1-Distill-Qwen-32B | 0.00 | 0.00 | 0.00 |
| R1-Distill-Qwen-7B | 0.00 | 0.00 | 0.00 |
| R1-Distill-Qwen-1.5B | 0.00 | 0.00 | 0.00 |
| GPT-4o | 4.00 | 0.00 | 0.00 |
| Qwen-Max | 3.71 | 0.00 | 0.00 |
| gemma-3-27b-IT | 5.56 | 5.56 | 2.22 |
| gemma-3-12b-IT | 5.56 | 2.22 | 1.11 |
| gemma-3-4b-IT | 4.44 | 2.22 | 0.00 |
| Qwen2.5-72B-IT | 4.44 | 8.89 | 0.00 |
| Qwen2.5-32B-IT | 6.67 | 14.44 | 1.11 |
| Qwen2.5-7B-IT | 6.67 | 3.33 | 0.00 |
| Qwen2.5-1.5B-IT | 0.00 | 0.00 | 0.00 |
| Llama-3.1-70B-IT | 10.00 | 7.78 | 1.11 |
| Llama-3.1-8B-IT | 0.00 | 1.11 | 0.00 |
| Mistral-Small-24B-IT-2501 | 6.67 | 10.00 | 0.00 |
| Ministral-8B-IT-2410 | 2.22 | 0.00 | 0.00 |

Table 29: Average accuracy for MagneticField across different difficulties.

| Model | Easy | Medium | Hard |
|---|---|---|---|
| o3-mini | 4.44 | 2.22 | 1.11 |
| R1 | 31.11 | 18.89 | 1.11 |
| QwQ-32B | 16.67 | 6.67 | 0.00 |
| R1-Distill-Llama-70B | 0.00 | 0.00 | 0.00 |
| R1-Distill-Qwen-32B | 0.00 | 0.00 | 0.00 |
| R1-Distill-Qwen-7B | 0.00 | 0.00 | 0.00 |
| R1-Distill-Qwen-1.5B | 0.00 | 0.00 | 0.00 |
| GPT-4o | 1.11 | 0.00 | 0.00 |
| Qwen-Max | 6.67 | 2.22 | 0.00 |
| gemma-3-27b-IT | 0.00 | 0.00 | 0.00 |
| gemma-3-12b-IT | 0.00 | 0.00 | 0.00 |
| gemma-3-4b-IT | 0.00 | 0.00 | 0.00 |
| Qwen2.5-72B-IT | 0.00 | 0.00 | 0.00 |
| Qwen2.5-32B-IT | 0.00 | 0.00 | 0.00 |
| Qwen2.5-7B-IT | 1.11 | 0.00 | 0.00 |
| Qwen2.5-1.5B-IT | 0.00 | 0.00 | 0.00 |
| Llama-3.1-70B-IT | 2.22 | 0.00 | 0.00 |
| Llama-3.1-8B-IT | 0.00 | 0.00 | 0.00 |
| Mistral-Small-24B-IT-2501 | 0.00 | 0.00 | 0.00 |
| Ministral-8B-IT-2410 | 0.00 | 0.00 | 0.00 |

Table 30: Average accuracy for MahjongDetective across different difficulties.

| Model | Easy | Medium | Hard |
|---|---|---|---|
| o3-mini | 72.22 | 55.56 | 37.78 |
| R1 | 62.22 | 32.22 | 7.78 |
| QwQ-32B | 88.89 | 32.22 | 21.11 |
| R1-Distill-Llama-70B | 52.22 | 23.33 | 13.33 |
| R1-Distill-Qwen-32B | 31.11 | 11.11 | 1.11 |
| R1-Distill-Qwen-7B | 8.89 | 2.22 | 1.11 |
| R1-Distill-Qwen-1.5B | 5.56 | 0.00 | 0.00 |
| GPT-4o | 34.44 | 5.56 | 1.11 |
| Qwen-Max | 31.11 | 7.78 | 4.44 |
| gemma-3-27b-IT | 40.00 | 14.44 | 6.67 |
| gemma-3-12b-IT | 30.00 | 8.89 | 2.22 |
| gemma-3-4b-IT | 0.00 | 5.56 | 2.22 |
| Qwen2.5-72B-IT | 38.89 | 2.22 | 1.11 |
| Qwen2.5-32B-IT | 2.22 | 3.33 | 4.44 |
| Qwen2.5-7B-IT | 35.56 | 11.11 | 2.22 |
| Qwen2.5-1.5B-IT | 6.67 | 0.00 | 2.22 |
| Llama-3.1-70B-IT | 56.67 | 23.33 | 1.11 |
| Llama-3.1-8B-IT | 34.44 | 6.67 | 3.33 |
| Mistral-Small-24B-IT-2501 | 18.89 | 2.22 | 0.00 |
| Ministral-8B-IT-2410 | 3.33 | 0.00 | 0.00 |

Table 31: Average accuracy for MedianQuery across different difficulties.

| Model | Easy | Medium | Hard |
|---|---|---|---|
| o3-mini | 62.22 | 28.89 | 14.44 |
| R1 | 64.44 | 33.33 | 21.11 |
| QwQ-32B | 57.78 | 20.00 | 10.00 |
| R1-Distill-Llama-70B | 51.11 | 25.56 | 5.56 |
| R1-Distill-Qwen-32B | 12.22 | 1.11 | 0.00 |
| R1-Distill-Qwen-7B | 0.00 | 0.00 | 0.00 |
| R1-Distill-Qwen-1.5B | 0.00 | 10.00 | 1.11 |
| GPT-4o | 3.33 | 4.44 | 1.11 |
| Qwen-Max | 18.89 | 16.67 | 5.56 |
| gemma-3-27b-IT | 1.11 | 0.00 | 0.00 |
| gemma-3-12b-IT | 17.78 | 3.33 | 3.33 |
| gemma-3-4b-IT | 0.00 | 7.78 | 1.11 |
| Qwen2.5-72B-IT | 6.67 | 4.44 | 1.11 |
| Qwen2.5-32B-IT | 31.11 | 24.44 | 0.00 |
| Qwen2.5-7B-IT | 46.67 | 11.11 | 2.22 |
| Qwen2.5-1.5B-IT | 0.00 | 0.00 | 0.00 |
| Llama-3.1-70B-IT | 36.67 | 11.11 | 6.67 |
| Llama-3.1-8B-IT | 23.33 | 11.11 | 0.00 |
| Mistral-Small-24B-IT-2501 | 22.22 | 2.22 | 1.11 |
| Ministral-8B-IT-2410 | 13.33 | 7.78 | 6.67 |

Table 32: Average accuracy for MimicHunt across different difficulties.

| Model | Easy | Medium | Hard |
|---|---|---|---|
| o3-mini | 66.67 | 40.00 | 30.00 |
| R1 | 45.56 | 26.67 | 7.78 |
| QwQ-32B | 65.56 | 0.00 | 0.00 |
| R1-Distill-Llama-70B | 32.22 | 2.22 | 0.00 |
| R1-Distill-Qwen-32B | 26.67 | 1.11 | 0.00 |
| R1-Distill-Qwen-7B | 0.00 | 0.00 | 0.00 |
| R1-Distill-Qwen-1.5B | 1.11 | 0.00 | 0.00 |
| GPT-4o | 50.00 | 26.67 | 18.89 |
| Qwen-Max | 70.00 | 1.11 | 0.00 |
| gemma-3-27b-IT | 71.11 | 1.11 | 2.22 |
| gemma-3-12b-IT | 58.89 | 0.00 | 0.00 |
| gemma-3-4b-IT | 5.56 | 0.00 | 0.00 |
| Qwen2.5-72B-IT | 66.67 | 47.78 | 35.56 |
| Qwen2.5-32B-IT | 66.67 | 5.56 | 28.89 |
| Qwen2.5-7B-IT | 43.33 | 0.00 | 0.00 |
| Qwen2.5-1.5B-IT | 0.00 | 0.00 | 0.00 |
| Llama-3.1-70B-IT | 76.67 | 46.67 | 22.22 |
| Llama-3.1-8B-IT | 30.00 | 11.11 | 7.78 |
| Mistral-Small-24B-IT-2501 | 31.11 | 1.11 | 0.00 |
| Ministral-8B-IT-2410 | 11.11 | 0.00 | 0.00 |

Table 33: Average accuracy for MinMax across different difficulties.

| Model | Easy | Medium | Hard |
|---|---|---|---|
| o3-mini | 3.33 | 3.33 | 3.33 |
| R1 | 14.44 | 12.22 | 14.44 |
| QwQ-32B | 8.89 | 3.33 | 1.11 |
| R1-Distill-Llama-70B | 1.11 | 1.11 | 1.11 |
| R1-Distill-Qwen-32B | 0.00 | 0.00 | 2.22 |
| R1-Distill-Qwen-7B | 0.00 | 0.00 | 0.00 |
| R1-Distill-Qwen-1.5B | 0.00 | 0.00 | 0.00 |
| GPT-4o | 0.00 | 1.11 | 6.67 |
| Qwen-Max | 0.00 | 0.00 | 2.22 |
| gemma-3-27b-IT | 13.33 | 1.11 | 0.00 |
| gemma-3-12b-IT | 0.00 | 0.00 | 0.00 |
| gemma-3-4b-IT | 1.11 | 0.00 | 0.00 |
| Qwen2.5-72B-IT | 2.22 | 14.44 | 2.22 |
| Qwen2.5-32B-IT | 3.33 | 1.11 | 0.00 |
| Qwen2.5-7B-IT | 0.00 | 0.00 | 0.00 |
| Qwen2.5-1.5B-IT | 0.00 | 0.00 | 0.00 |
| Llama-3.1-70B-IT | 0.00 | 0.00 | 0.00 |
| Llama-3.1-8B-IT | 3.33 | 0.00 | 0.00 |
| Mistral-Small-24B-IT-2501 | 0.00 | 0.00 | 3.33 |
| Ministral-8B-IT-2410 | 16.67 | 7.78 | 1.11 |

Table 34: Average accuracy for SafepathFinder across different difficulties.

| Model | Easy | Medium | Hard |
|---|---|---|---|
| o3-mini | 6.67 | 1.11 | 0.00 |
| R1 | 30.00 | 30.00 | 24.44 |
| QwQ-32B | 33.33 | 16.67 | 7.78 |
| R1-Distill-Llama-70B | 23.33 | 12.22 | 8.89 |
| R1-Distill-Qwen-32B | 14.44 | 4.44 | 3.33 |
| R1-Distill-Qwen-7B | 1.11 | 0.00 | 0.00 |
| R1-Distill-Qwen-1.5B | 0.00 | 0.00 | 0.00 |
| GPT-4o | 7.78 | 6.67 | 2.22 |
| Qwen-Max | 8.89 | 3.33 | 5.56 |
| gemma-3-27b-IT | 11.11 | 3.33 | 3.33 |
| gemma-3-12b-IT | 6.67 | 3.33 | 2.22 |
| gemma-3-4b-IT | 4.44 | 0.00 | 2.22 |
| Qwen2.5-72B-IT | 4.44 | 2.22 | 0.00 |
| Qwen2.5-32B-IT | 8.89 | 11.11 | 1.11 |
| Qwen2.5-7B-IT | 14.44 | 7.78 | 11.11 |
| Qwen2.5-1.5B-IT | 0.00 | 0.00 | 0.00 |
| Llama-3.1-70B-IT | 6.67 | 2.22 | 4.44 |
| Llama-3.1-8B-IT | 1.11 | 0.00 | 0.00 |
| Mistral-Small-24B-IT-2501 | 4.44 | 2.22 | 0.00 |
| Ministral-8B-IT-2410 | 5.56 | 4.44 | 5.56 |

Table 35: Average accuracy for TrainPursuit across different difficulties.

| Model | Easy | Medium | Hard |
|---|---|---|---|
| o3-mini | 45.56 | 52.22 | 41.11 |
| R1 | 61.11 | 55.56 | 51.11 |
| QwQ-32B | 58.89 | 54.44 | 63.33 |
| R1-Distill-Llama-70B | 64.44 | 64.44 | 47.78 |
| R1-Distill-Qwen-32B | 23.33 | 17.78 | 6.67 |
| R1-Distill-Qwen-7B | 0.00 | 3.33 | 0.00 |
| R1-Distill-Qwen-1.5B | 0.00 | 0.00 | 0.00 |
| GPT-4o | 11.11 | 21.11 | 15.56 |
| Qwen-Max | 15.56 | 14.44 | 14.44 |
| gemma-3-27b-IT | 60.00 | 40.00 | 33.33 |
| gemma-3-12b-IT | 13.33 | 0.00 | 0.00 |
| gemma-3-4b-IT | 20.00 | 8.89 | 6.67 |
| Qwen2.5-72B-IT | 63.33 | 55.56 | 52.22 |
| Qwen2.5-32B-IT | 65.56 | 62.22 | 53.33 |
| Qwen2.5-7B-IT | 23.33 | 11.11 | 3.33 |
| Qwen2.5-1.5B-IT | 0.00 | 0.00 | 0.00 |
| Llama-3.1-70B-IT | 43.33 | 32.22 | 17.78 |
| Llama-3.1-8B-IT | 33.33 | 16.67 | 10.00 |
| Mistral-Small-24B-IT-2501 | 51.11 | 42.22 | 22.22 |
| Ministral-8B-IT-2410 | 0.00 | 0.00 | 0.00 |

Table 36: Average accuracy for TreasureHunt across different difficulties.

| Model | Easy | Medium | Hard |
|---|---|---|---|
| o3-mini | 86.67 | 78.89 | 80.00 |
| R1 | 83.33 | 81.11 | 77.78 |
| QwQ-32B | 70.00 | 70.00 | 70.00 |
| R1-Distill-Llama-70B | 8.89 | 6.67 | 8.89 |
| R1-Distill-Qwen-32B | 11.11 | 13.33 | 11.11 |
| R1-Distill-Qwen-7B | 1.11 | 2.22 | 0.00 |
| R1-Distill-Qwen-1.5B | 0.00 | 0.00 | 0.00 |
| GPT-4o | 43.33 | 50.00 | 53.33 |
| Qwen-Max | 46.67 | 45.56 | 54.44 |
| gemma-3-27b-IT | 13.33 | 10.00 | 8.89 |
| gemma-3-12b-IT | 11.11 | 11.11 | 14.44 |
| gemma-3-4b-IT | 1.11 | 1.11 | 8.89 |
| Qwen2.5-72B-IT | 11.11 | 5.56 | 2.22 |
| Qwen2.5-32B-IT | 26.67 | 21.11 | 26.67 |
| Qwen2.5-7B-IT | 6.67 | 3.33 | 14.44 |
| Qwen2.5-1.5B-IT | 2.22 | 2.22 | 8.89 |
| Llama-3.1-70B-IT | 12.22 | 12.22 | 13.33 |
| Llama-3.1-8B-IT | 0.00 | 4.44 | 3.33 |
| Mistral-Small-24B-IT-2501 | 4.44 | 10.00 | 13.33 |
| Ministral-8B-IT-2410 | 10.00 | 3.33 | 10.00 |

Table 37: Average accuracy for VladikMaze across different difficulties.

| Model | Easy | Medium | Hard |
|---|---|---|---|
| o3-mini | 42.22 | 53.33 | 7.78 |
| R1 | 27.78 | 18.89 | 0.00 |
| QwQ-32B | 13.33 | 3.33 | 0.00 |
| R1-Distill-Llama-70B | 1.11 | 0.00 | 0.00 |
| R1-Distill-Qwen-32B | 0.00 | 0.00 | 0.00 |
| R1-Distill-Qwen-7B | 0.00 | 0.00 | 0.00 |
| R1-Distill-Qwen-1.5B | 0.00 | 0.00 | 0.00 |
| GPT-4o | 3.33 | 0.00 | 0.00 |
| Qwen-Max | 11.11 | 0.00 | 0.00 |
| gemma-3-27b-IT | 4.44 | 0.00 | 0.00 |
| gemma-3-12b-IT | 0.00 | 0.00 | 0.00 |
| gemma-3-4b-IT | 0.00 | 0.00 | 0.00 |
| Qwen2.5-72B-IT | 0.00 | 0.00 | 0.00 |
| Qwen2.5-32B-IT | 0.00 | 0.00 | 0.00 |
| Qwen2.5-7B-IT | 0.00 | 0.00 | 0.00 |
| Qwen2.5-1.5B-IT | 0.00 | 0.00 | 0.00 |
| Llama-3.1-70B-IT | 0.00 | 0.00 | 0.00 |
| Llama-3.1-8B-IT | 0.00 | 0.00 | 0.00 |
| Mistral-Small-24B-IT-2501 | 0.00 | 0.00 | 0.00 |
| Ministral-8B-IT-2410 | 0.00 | 0.00 | 0.00 |

Table 38: Average accuracy for Wordle across different difficulties.

| Model | Easy | Medium | Hard |
|---|---|---|---|
| o3-mini | 80.00 | 90.00 | 66.67 |
| R1 | 63.33 | 60.00 | 86.67 |
| QwQ-32B | 56.67 | 86.67 | 80.00 |
| R1-Distill-Llama-70B | 13.33 | 10.00 | 6.67 |
| R1-Distill-Qwen-32B | 10.00 | 23.33 | 16.67 |
| R1-Distill-Qwen-7B | 3.33 | 3.33 | 3.33 |
| R1-Distill-Qwen-1.5B | 0.00 | 0.00 | 0.00 |
| GPT-4o | 76.67 | 53.33 | 16.67 |
| Qwen-Max | 30.00 | 20.00 | 10.00 |
| gemma-3-27b-IT | 3.33 | 0.00 | 0.00 |
| gemma-3-12b-IT | 0.00 | 0.00 | 0.00 |
| gemma-3-4b-IT | 0.00 | 0.00 | 0.00 |
| Qwen2.5-72B-IT | 10.00 | 0.00 | 0.00 |
| Qwen2.5-32B-IT | 3.33 | 0.00 | 10.00 |
| Qwen2.5-7B-IT | 6.67 | 0.00 | 6.67 |
| Qwen2.5-1.5B-IT | 0.00 | 0.00 | 0.00 |
| Llama-3.1-70B-IT | 3.33 | 13.33 | 16.67 |
| Llama-3.1-8B-IT | 0.00 | 0.00 | 0.00 |
| Mistral-Small-24B-IT-2501 | 20.00 | 0.00 | 0.00 |
| Ministral-8B-IT-2410 | 0.00 | 0.00 | 0.00 |

Table 39: Average accuracy for XORBreaking across different difficulties.

| Model | Easy | Medium | Hard |
|---|---|---|---|
| o3-mini | 58.57 | 21.84 | 30.22 |
| R1 | 57.78 | 42.22 | 27.78 |
| QwQ-32B | 81.11 | 54.44 | 28.89 |
| R1-Distill-Llama-70B | 31.11 | 7.78 | 4.44 |
| R1-Distill-Qwen-32B | 32.22 | 8.89 | 4.44 |
| R1-Distill-Qwen-7B | 2.22 | 0.00 | 0.00 |
| R1-Distill-Qwen-1.5B | 0.00 | 0.00 | 0.00 |
| GPT-4o | 46.67 | 21.11 | 13.33 |
| Qwen-Max | 71.11 | 40.00 | 26.67 |
| gemma-3-27b-IT | 30.00 | 11.11 | 14.44 |
| gemma-3-12b-IT | 27.78 | 13.33 | 4.44 |
| gemma-3-4b-IT | 3.33 | 3.33 | 1.11 |
| Qwen2.5-72B-IT | 37.78 | 18.89 | 11.11 |
| Qwen2.5-32B-IT | 45.56 | 21.11 | 14.44 |
| Qwen2.5-7B-IT | 11.11 | 4.44 | 0.00 |
| Qwen2.5-1.5B-IT | 0.00 | 0.00 | 0.00 |
| Llama-3.1-70B-IT | 34.44 | 16.67 | 7.78 |
| Llama-3.1-8B-IT | 10.00 | 5.56 | 3.33 |
| Mistral-Small-24B-IT-2501 | 6.67 | 0.00 | 1.11 |
| Ministral-8B-IT-2410 | 3.33 | 2.22 | 3.33 |

Table 40: Average accuracy for ZeroFinding across different difficulties.

| Model | Easy | Medium | Hard |
|---|---|---|---|
| o3-mini | 43.33 | 56.67 | 43.33 |
| R1 | 40.00 | 20.00 | 13.33 |
| QwQ-32B | 26.67 | 23.33 | 26.67 |
| R1-Distill-Llama-70B | 20.00 | 23.33 | 6.67 |
| R1-Distill-Qwen-32B | 30.00 | 13.33 | 23.33 |
| R1-Distill-Qwen-7B | 0.00 | 3.33 | 0.00 |
| R1-Distill-Qwen-1.5B | 0.00 | 0.00 | 0.00 |
| GPT-4o | 6.67 | 6.67 | 0.00 |
| Qwen-Max | 33.33 | 6.67 | 0.00 |
| gemma-3-27b-IT | 3.33 | 0.00 | 3.33 |
| gemma-3-12b-IT | 0.00 | 3.33 | 0.00 |
| gemma-3-4b-IT | 0.00 | 3.33 | 0.00 |
| Qwen2.5-72B-IT | 3.33 | 0.00 | 3.33 |
| Qwen2.5-32B-IT | 20.00 | 0.00 | 0.00 |
| Qwen2.5-7B-IT | 3.33 | 0.00 | 0.00 |
| Qwen2.5-1.5B-IT | 3.33 | 0.00 | 3.33 |
| Llama-3.1-70B-IT | 6.67 | 0.00 | 0.00 |
| Llama-3.1-8B-IT | 0.00 | 0.00 | 0.00 |
| Mistral-Small-24B-IT-2501 | 3.33 | 0.00 | 0.00 |
| Ministral-8B-IT-2410 | 10.00 | 3.33 | 0.00 |

Table 41: Average accuracy for ZigzagGraph across different difficulties.

| Model | Easy | Medium | Hard |
|---|---|---|---|
| o3-mini | 23.61 | 17.78 | 16.67 |
| R1 | 62.50 | 31.11 | 30.00 |
| QwQ-32B | 22.22 | 13.33 | 5.56 |
| R1-Distill-Llama-70B | 13.89 | 8.89 | 1.11 |
| R1-Distill-Qwen-32B | 2.78 | 2.22 | 0.00 |
| R1-Distill-Qwen-7B | 0.00 | 0.00 | 0.00 |
| R1-Distill-Qwen-1.5B | 0.00 | 0.00 | 0.00 |
| GPT-4o | 18.06 | 13.33 | 3.33 |
| Qwen-Max | 20.83 | 20.00 | 6.67 |
| gemma-3-27b-IT | 12.50 | 7.78 | 0.00 |
| gemma-3-12b-IT | 12.50 | 6.67 | 0.00 |
| gemma-3-4b-IT | 2.78 | 4.44 | 0.00 |
| Qwen2.5-72B-IT | 27.78 | 18.89 | 5.56 |
| Qwen2.5-32B-IT | 18.06 | 11.11 | 2.22 |
| Qwen2.5-7B-IT | 0.00 | 2.22 | 0.00 |
| Qwen2.5-1.5B-IT | 0.00 | 0.00 | 0.00 |
| Llama-3.1-70B-IT | 18.06 | 11.11 | 2.22 |
| Llama-3.1-8B-IT | 1.39 | 2.22 | 0.00 |
| Mistral-Small-24B-IT-2501 | 18.06 | 10.00 | 6.67 |
| Ministral-8B-IT-2410 | 6.94 | 2.22 | 0.00 |

Table 42: Average accuracy for PermutationDiscovery across different difficulties.

| Model | Easy | Medium | Hard |
|---|---|---|---|
| o3-mini | 96.67 | 96.67 | 83.33 |
| R1 | 83.33 | 76.67 | 73.33 |
| QwQ-32B | 76.67 | 76.67 | 56.67 |
| R1-Distill-Llama-70B | 93.33 | 83.33 | 70.00 |
| R1-Distill-Qwen-32B | 23.33 | 26.67 | 30.00 |
| R1-Distill-Qwen-7B | 10.00 | 0.00 | 0.00 |
| R1-Distill-Qwen-1.5B | 3.33 | 0.00 | 0.00 |
| GPT-4o | 36.67 | 26.67 | 30.00 |
| Qwen-Max | 80.00 | 46.67 | 33.33 |
| gemma-3-27b-IT | 36.67 | 3.33 | 0.00 |
| gemma-3-12b-IT | 20.00 | 3.33 | 0.00 |
| gemma-3-4b-IT | 16.67 | 3.33 | 3.33 |
| Qwen2.5-72B-IT | 46.67 | 40.00 | 10.00 |
| Qwen2.5-32B-IT | 43.33 | 23.33 | 33.33 |
| Qwen2.5-7B-IT | 0.00 | 0.00 | 0.00 |
| Qwen2.5-1.5B-IT | 16.67 | 13.33 | 6.67 |
| Llama-3.1-70B-IT | 33.33 | 40.00 | 53.33 |
| Llama-3.1-8B-IT | 10.00 | 10.00 | 0.00 |
| Mistral-Small-24B-IT-2501 | 60.00 | 30.00 | 3.33 |
| Ministral-8B-IT-2410 | 30.00 | 0.00 | 0.00 |

Table 43: Average accuracy for PizzaSlice across different difficulties.

| Model | Easy | Medium | Hard |
|---|---|---|---|
| o3-mini | 86.67 | 66.67 | 32.22 |
| R1 | 83.33 | 60.00 | 30.00 |
| QwQ-32B | 55.56 | 17.78 | 16.67 |
| R1-Distill-Llama-70B | 62.22 | 20.00 | 7.78 |
| R1-Distill-Qwen-32B | 12.22 | 3.33 | 3.33 |
| R1-Distill-Qwen-7B | 1.11 | 0.00 | 0.00 |
| R1-Distill-Qwen-1.5B | 0.00 | 0.00 | 0.00 |
| GPT-4o | 58.89 | 30.00 | 26.67 |
| Qwen-Max | 71.11 | 45.56 | 22.22 |
| gemma-3-27b-IT | 65.56 | 40.00 | 21.11 |
| gemma-3-12b-IT | 45.56 | 34.44 | 34.44 |
| gemma-3-4b-IT | 38.89 | 32.22 | 28.89 |
| Qwen2.5-72B-IT | 67.78 | 42.22 | 27.78 |
| Qwen2.5-32B-IT | 47.78 | 43.33 | 27.78 |
| Qwen2.5-7B-IT | 70.00 | 34.44 | 31.11 |
| Qwen2.5-1.5B-IT | 62.22 | 41.11 | 7.78 |
| Llama-3.1-70B-IT | 61.11 | 38.89 | 27.78 |
| Llama-3.1-8B-IT | 51.11 | 23.33 | 25.56 |
| Mistral-Small-24B-IT-2501 | 65.56 | 25.56 | 24.44 |
| Ministral-8B-IT-2410 | 71.11 | 26.67 | 28.89 |

Table 44: Average accuracy for RPD across different difficulties.

| Model | Easy | Medium | Hard |
|---|---|---|---|
| o3-mini | 54.44 | 0.00 | 0.00 |
| R1 | 34.44 | 0.00 | 0.00 |
| QwQ-32B | 28.89 | 0.00 | 0.00 |
| R1-Distill-Llama-70B | 28.89 | 0.00 | 0.00 |
| R1-Distill-Qwen-32B | 5.56 | 0.00 | 0.00 |
| R1-Distill-Qwen-7B | 0.00 | 0.00 | 0.00 |
| R1-Distill-Qwen-1.5B | 0.00 | 0.00 | 0.00 |
| GPT-4o | 27.78 | 0.00 | 0.00 |
| Qwen-Max | 33.33 | 0.00 | 0.00 |
| gemma-3-27b-IT | 31.11 | 0.00 | 0.00 |
| gemma-3-12b-IT | 31.11 | 0.00 | 0.00 |
| gemma-3-4b-IT | 18.89 | 0.00 | 0.00 |
| Qwen2.5-72B-IT | 28.89 | 0.00 | 0.00 |
| Qwen2.5-32B-IT | 35.56 | 0.00 | 0.00 |
| Qwen2.5-7B-IT | 13.33 | 0.00 | 0.00 |
| Qwen2.5-1.5B-IT | 26.67 | 0.00 | 0.00 |
| Llama-3.1-70B-IT | 30.00 | 0.00 | 0.00 |
| Llama-3.1-8B-IT | 15.56 | 0.00 | 0.00 |
| Mistral-Small-24B-IT-2501 | 28.89 | 0.00 | 0.00 |
| Ministral-8B-IT-2410 | 21.11 | 0.00 | 0.00 |

Table 45: Average accuracy for RainbowCandy across different difficulties.

| Model | Easy | Medium | Hard |
|---|---|---|---|
| o3-mini | 7.78 | 0.00 | 0.00 |
| R1 | 5.56 | 0.00 | 0.00 |
| QwQ-32B | 10.00 | 0.00 | 0.00 |
| R1-Distill-Llama-70B | 0.00 | 0.00 | 0.00 |
| R1-Distill-Qwen-32B | 0.00 | 0.00 | 0.00 |
| R1-Distill-Qwen-7B | 0.00 | 0.00 | 0.00 |
| R1-Distill-Qwen-1.5B | 0.00 | 0.00 | 0.00 |
| GPT-4o | 0.00 | 0.00 | 0.00 |
| Qwen-Max | 0.00 | 0.00 | 0.00 |
| gemma-3-27b-IT | 0.00 | 0.00 | 0.00 |
| gemma-3-12b-IT | 0.00 | 0.00 | 0.00 |
| gemma-3-4b-IT | 0.00 | 0.00 | 0.00 |
| Qwen2.5-72B-IT | 0.00 | 0.00 | 0.00 |
| Qwen2.5-32B-IT | 0.00 | 0.00 | 0.00 |
| Qwen2.5-7B-IT | 0.00 | 0.00 | 0.00 |
| Qwen2.5-1.5B-IT | 0.00 | 0.00 | 0.00 |
| Llama-3.1-70B-IT | 0.00 | 0.00 | 0.00 |
| Llama-3.1-8B-IT | 0.00 | 0.00 | 0.00 |
| Mistral-Small-24B-IT-2501 | 0.00 | 0.00 | 0.00 |
| Ministral-8B-IT-2410 | 0.00 | 0.00 | 0.00 |

Table 46: Average accuracy for RotaryLock across different difficulties.

| Model | Easy | Medium | Hard |
|---|---|---|---|
| o3-mini | 93.33 | 46.67 | 23.33 |
| R1 | 86.67 | 46.67 | 20.00 |
| QwQ-32B | 100.00 | 26.67 | 10.00 |
| R1-Distill-Llama-70B | 93.33 | 40.00 | 3.33 |
| R1-Distill-Qwen-32B | 66.67 | 10.00 | 0.00 |
| R1-Distill-Qwen-7B | 0.00 | 0.00 | 0.00 |
| R1-Distill-Qwen-1.5B | 0.00 | 0.00 | 0.00 |
| GPT-4o | 46.67 | 20.00 | 0.00 |
| Qwen-Max | 83.33 | 23.33 | 3.33 |
| gemma-3-27b-IT | 3.33 | 0.00 | 0.00 |
| gemma-3-12b-IT | 0.00 | 0.00 | 0.00 |
| gemma-3-4b-IT | 10.00 | 0.00 | 0.00 |
| Qwen2.5-72B-IT | 90.00 | 10.00 | 3.33 |
| Qwen2.5-32B-IT | 80.00 | 10.00 | 0.00 |
| Qwen2.5-7B-IT | 46.67 | 0.00 | 0.00 |
| Qwen2.5-1.5B-IT | 16.67 | 0.00 | 0.00 |
| Llama-3.1-70B-IT | 63.33 | 0.00 | 0.00 |
| Llama-3.1-8B-IT | 10.00 | 0.00 | 0.00 |
| Mistral-Small-24B-IT-2501 | 0.00 | 0.00 | 0.00 |
| Ministral-8B-IT-2410 | 3.33 | 0.00 | 0.00 |

Table 47: Average accuracy for PaperNumber across different difficulties.

# P GRADING CASE STUDY OF A HARD TASK (CODEFORCES 3500)

To provide an intuitive understanding of how EvolArena handles high-difficulty reasoning tasks, we detail the implementation of **GridGame**. This task is adapted from a Codeforces problem (e.g., "Grid Game") with a difficulty rating of 3500, representing the peak of competitive programming challenges.

## P.1 TASK LOGIC

**Objective:** Two players take turns selecting numbers from an $N \times M$ grid. The game ends when all cells are selected. The player wins if their total sum is strictly less than the opponent's sum. **Constraint:** After the first move, every selected cell must be orthogonally adjacent to a previously selected cell (by either player). This requires deep strategic lookahead to force the opponent into selecting high-value cells.

---

**Original Codeforces Problem: Grid Game (Difficulty 3500)**

**Problem Description:** You are given a grid with $n$ rows and $m$ columns. You need to fill each cell with a unique integer from 1 to $n \cdot m$. After filling the grid, you will play a game on this grid against the interactor.
Players take turns selecting one of the previously unselected cells from the grid, with the interactor going first.

- On the first turn, the interactor can choose any cell from the grid.
- After that, any chosen cell must be **orthogonally adjacent** to at least one previously selected cell. (Two cells are adjacent if they share an edge).

The game continues until all cells have been selected. Your goal is to let the sum of numbers in the cells selected by you be **strictly less** than the sum of numbers in the cells selected by the interactor.

**Input/Output constraints:** The first line contains $t$ ($1 \le t \le 100$) test cases. Each test case contains $n$ and $m$ ($4 \le n, m \le 10$). If the sum of numbers in the cells selected by you is not strictly less than the interactor's sum, you receive a Wrong Answer verdict.

**Tags:** constructive algorithms, games, graph matchings, greedy, interactive.

---

## P.2 GENERATOR: PROCEDURAL PROBLEM CONSTRUCTION

The Generator is responsible for creating a unique, randomized game instance for each evaluation to prevent data contamination and memorization. For the GridSum task, simply memorizing a strategy is insufficient; the model must analyze the specific numerical layout.

The generator performs two key functions:

1. **State Randomization:** It generates a random permutation of numbers from 1 to $N \times M$ and maps them to the grid coordinates. This ensures that every game instance presents a novel numerical landscape, forcing the model to perform calculation and planning dynamically.

2. **Prompt Synthesis:** It embeds this generated grid into a standardized natural language template (similar to a Codeforces problem statement), explicitly defining the grid size, the specific numbers in each cell, and the adjacency rules.

## P.3 MONITOR: DETERMINISTIC GAME ENGINE AND OPPONENT

The Monitor acts as the deterministic game engine that enforces rules and simulates the opponent. Unlike simple format checkers, it maintains the global game state—including the set of selected cells $S$, the grid values $G$, and the cumulative scores ($Sum_{Player}, Sum_{System}$)—and executes the following critical functions:

- **Strict Adjacency Enforcement:** The defining constraint of this Codeforces 3500 task is that every newly selected cell (after the first move) must be orthogonally adjacent to *at least one* cell in the set of previously selected cells $S$. The Monitor strictly validates this topological constraint at every turn, rejecting any move that violates it.

- **System Strategy Execution:** The Monitor acts as the opponent (System). It calculates the set of all currently valid moves based on the updated $S$ and selects one (randomly in this baseline implementation) to expand the territory, dynamically updating the system's score.

- **Outcome Determination:** Upon game completion (when all cells are filled or maximum turns are reached), the Monitor compares the final sums to deterministically judge the winner.

## P.4 EVALUATOR: MULTI-DIMENSIONAL GRADING

The Evaluator acts as a comprehensive diagnostic tool. Instead of simply checking the final output, it parses the entire interaction history $H$ to compute four critical metrics, providing a holistic view of the model's reasoning capability:

- **Accuracy:** It determines success by checking if the player achieved a strictly lower sum than the system ($Sum_{Player} < Sum_{System}$) upon game completion, or if the system was forced into a stalemate.

- **Invalid Rate:** It rigorously counts every instance where the model attempted an illegal move (e.g., violating the adjacency constraint), distinguishing between "strategy failure" and "rule violation."

- **Pattern Analysis:** It integrates an LLM-based analyzer to scan the model's "Thought" trace, quantifying specific reasoning behaviors such as *Planning* (looking ahead for low-value cells) and *Verifying* (checking adjacency).

- **Efficiency:** It records the number of turns taken to achieve victory, penalizing failed attempts with the maximum turn count.

---

**Case P.1: GridGame Problem Template**

Let's play the Grid Game! Your task is to choose cells strategically to win.
Rules:
1. Game Setup:
- Grid size: $\{n\}*\{m\}$
- Grid already filled with numbers 1 to $\{n*m\}$
- Each number appears exactly once $\{grid\_str\}$
2. Game Mechanics:
- Players take turns selecting unselected cells
- You move first
- Any cell chosen after first turn must be adjacent to a previously selected cell
- Cells are adjacent if they share an edge (up/down/left/right)
- Game ends when all cells are selected
- You win if your selected numbers sum < my sum
3. Adjacency Example:
For cell $(2, 2)$:
- Adjacent cells: $(1, 2), (2, 1), (2, 3), (3, 2)$
- Diagonal cells like $(1, 1)$ are not adjacent
- Must choose a cell adjacent to any previously selected cell
Query Type:
Format: "My Choice: $x$ $y$"
where $x$ is row (1 to $\{n\}$) and $y$ is column (1 to $\{m\}$)
Example Interaction:
You: "My Choice: 2 2"
- Selecting cell at row 2, column 2
Me: "My Choice: 2 3"
- Cell is adjacent to $(2, 2)$

---

You: "My Choice: 1 2"
- Cell is adjacent to $(2, 2)$
Instructions:
1. Make choices based on grid values
2. Use exactly the format shown above
3. Explain your reasoning before each choice
Remember:
- Use exact format: "My Choice: $x\ y$"
- Choose only adjacent cells after first turn
- First move can be any cell
- Keep track of both sums
- Plan moves to keep your sum smaller
- Invalid move = automatic loss
Ready to start? Make your first choice!

---

**Algorithm 14** Monitor for GridGame

---

**Require:** User Input $I$, Selected Set $S$, Grid $G$, Scores $Sum_P, Sum_S$, Turn $t$, MaxTurns $K$

1: **Regex:** r"My Choice:\s*(\d+)\s+(\d+)"
2: **if** $I$ matches Regex with $(x, y)$ **then**
3:    $Cell \leftarrow (x, y)$
4:                                      ▷ 1. Basic Validity Checks
5:    **if** $Cell \notin G$ **or** $Cell \in S$ **then**
6:       **return** "Invalid", "Invalid cell choice"
7:    **end if**
8:                               ▷ 2. Enforce Adjacency (Critical Constraint)
9:    **if** $S \neq \emptyset$ **and** $\neg \exists s \in S, \text{IsOrthogonallyAdjacent}(Cell, s)$ **then**
10:       **return** "Invalid", "Cell must be adjacent to previous selection"
11:    **end if**
12:                                     ▷ 3. Update Player State
13:    $S.\text{add}(Cell)$
14:    $Sum_P \leftarrow Sum_P + G[Cell]$
15:                             ▷ 4. System Turn: Calculate Valid Moves
16:    $ValidMoves \leftarrow \{c \mid c \in G \setminus S \text{ and } \exists s \in S, \text{IsAdjacent}(c, s)\}$
17:    **if** $ValidMoves = \emptyset$ **then**
18:       **return** "My Choice: $x\ y$", "I have no valid moves. You win!"
19:    **end if**
20:    $SysMove \leftarrow \text{RandomChoice}(ValidMoves)$
21:    $S.\text{add}(SysMove); Sum_S \leftarrow Sum_S + G[SysMove]$
22:                                   ▷ 5. Check End Condition
23:    **if** $t == K$ **then**
24:       **if** $Sum_P < Sum_S$ **then**
25:          **return** "My Choice: $x\ y$", "My Choice: $SysMove$\n You win!"
26:       **else**
27:          **return** "My Choice: $x\ y$", "My Choice: $SysMove$\n You lose!"
28:       **end if**
29:    **end if**
30:    **return** "My Choice: $x\ y$", "My Choice: $SysMove$"
31: **else**
32:    **return** "Invalid", "Invalid Format"
33: **end if**

---

**Algorithm 15** Generator for GridGame

---

**Require:** Complexity Parameters $N$ (Rows), $M$ (Cols)

1:                                    ▷ 1. Construct Randomized Game State

2: $Values \leftarrow$ RandomPermutation($[1, \ldots, N \times M]$)

3: $Grid \leftarrow$ MapToCoordinates($Values, N, M$)

4: $TotalTurns \leftarrow (N \times M)/2$

5:                                 ▷ 2. Synthesize Natural Language Prompt

6: $GridDescription \leftarrow$ "Initial grid state:\n"

7: **for** $i \leftarrow 1$ **to** $N$ **do**

8:     **for** $j \leftarrow 1$ **to** $M$ **do**

9:         $GridDescription \leftarrow GridDescription +$ f"Cell $(i, j)$: $Grid[i, j]$\n"

10:     **end for**

11: **end for**

12: $ProblemPrompt \leftarrow$ FillTemplate(TaskDescription, $GridDescription, N, M$)

13: **return** ($ProblemPrompt, Grid, TotalTurns$)

---

**Algorithm 16** Evaluator for GridGame

---

**Require:** Interaction History $H$, MaxTurns $K$, Final Scores $Sum_P, Sum_S$

1: **Initialize Metrics:**

2: $Success \leftarrow$ False

3: $TurnCount \leftarrow$ Length($H$)

4: $InvalidCount \leftarrow 0$

5: $Patterns \leftarrow \{$Associate $: 0,$ Verify $: 0,$ Plan $: 0,$ Feedback $: 0\}$

6: **for** each turn $t$ in $H$ **do**

7:     $Feedback \leftarrow H[t]$.Feedback

8:     $Thought \leftarrow H[t]$.ModelThought

9:                             ▷ 1. Robustness: Count Rule Violations

10:     **if** $Feedback$ contains "Invalid" **then**

11:         $InvalidCount \leftarrow InvalidCount + 1$

12:     **end if**

13:                        ▷ 2. Cognitive Diagnosis: Extract Reasoning Steps

14:                       ▷ Calls external LLM analyzer to classify thought process

15:     $Patterns \leftarrow Patterns +$ AnalyzeReasoningPatterns($Thought$)

16:                               ▷ 3. Outcome Verification

17:     **if** $Feedback$ contains "You win!" **then**

18:         $Success \leftarrow$ True

19:     **else if** $t == K$ **then**                       ▷ Game ended normally

20:         **if** $Sum_P < Sum_S$ **then**

21:             $Success \leftarrow$ True

22:         **end if**

23:     **end if**

24: **end for**

25:                                   ▷ 4. Efficiency Calculation

26: $Efficiency \leftarrow TurnCount$ if $Success$ else $K$

27: $InvalidRate \leftarrow InvalidCount/TurnCount$

28: **return** $\{Success, Efficiency, InvalidRate, Patterns\}$

---

