# OpenReview forum: "EvolArena: An Evolving Arena for Multi-Turn Reasoning in LLMs"
_ICLR.cc/2026/Conference — ICLR 2026 Conference Withdrawn Submission_

### Official Review · Reviewer_P7mg · 2025-10-20

**Soundness:** 2
**Presentation:** 3
**Contribution:** 2
**Rating:** 2
**Confidence:** 4

**Summary:**

# Summary

The authors present EvolArena, a benchmark set of reasoning, planning, and game playing tasks for evaluating LLMs capabilities in sequential decision making.

## What is the problem solved? Is it a known problem? How is it solved in the literature and how is the current approach different?

Sequential decision making either in a single-player (e.g., planning) or muti-player (e.g., general game playing) setting is a central problem in computer science in general and in AI in particular. There are several communities within AI that focus on creating solvers for a variety of settings, for many decades now. Each of these communities of course have their own benchmarks to evaluate these solvers on. Designing new benchmarks is often a time consuming process that also requires an understanding of a particular setting. For instance, benchmarks for partially observable setting for a single player games would be very different from the multi-player setting, and from so-called classical planning setting (single player, full observability, deterministic action outcomes, closed world).

In the era of LLMs, when a single "solver" is assumed to be able to tackle all possible problems, these separate benchmarks are grouped together to check the abilities of LLMs across the spectrum of possible decision making tasks. I find this aggregation to be ill-motivated, and done for convenience only, also preventing from comparing to the established solvers in each setting. In most cases, the aggregation is also performed over existing benchmarks, which requires some engineering work to put them into the same framework.

The authors follow the same trend, aggregating over various types of sequential decision making problems, including information acquisition games, path planning, and multi agent planning.
Differently from most work in the area, they create new benchmarks based mostly on two sources, popular logic puzzles published on the New York Times website and 32 multi-agent planning tasks originating from algorithmic competition problems on Codeforces.

The authors position these tasks as partitioned into 4 categories: Inductive Reasoning, Abductive Reasoning, Deductive Reasoning, and Planning Reasoning. I find this partitioning puzzling. Information acquisition games are not about inductive reasoning, path planning is not about deductive reasoning, and Planning Reasoning is not a common term, not sure what it means in your case. The example task in that category is a general game playing or multi-agent planning task, in an adversarial setting. This positioning sounds more like PR than science, I would strongly encourage you to drop it in the next iterations of this work.

While the authors mention some related work, they miss some of the most relevant ones, specifically AgentBench [1] and AgentBoard [2]. I would be very interested to hear how the authors differentiate themselves from these works.

# Significance

The existence of a large collection of similar benchmarks reduces the significance of this work. Also, no justification is given for the benefits of these sequential decision making tasks over other existing ones.

# Soundness

The idea seems sound, but since there is no formal definition of these tasks, what constitutes a solution, how a solution is validated, etc, it is hard to judge.

For task generation, the authors divide the tasks into three categories, "easy", "medium", and "hard" based on the value to the input parameter to the generator. This hint on (a) the generators having only one parameter, and (b) the larger the value to that parameter, the harder the problem is.
(a) is often an indication of a very limited generator, stabilizing many decisions and focusing on only a very limited set of possible instances, which in future might lead to memorization concerns, much like with the BlocksWorld domain in planning.
(b) is not necessarily true in sequential decision making. For instance, adding more resources or increasing the maze size without scaling the obstacles makes the tasks easier, not harder.

The evaluation is done on several axes:
1. Accuracy, the standard measure of success
2. The Efficiency is a pairwise comparison of models on the solutions lengths of their commonly solved tasks. This can be misleading, as it only measures in absolute terms, so if A solved 3 tasks finding solutions of length 5, 5, 5, while B found solutions 4, 4, 100, B is considered to be better than A under the efficiency measure.
3. Invalid Rate measures how many tasks did the model generate invalid actions for. Related datasets that explicitly measure action applicability are [3,4,5,6].
4. Pattern Analysis measures how much does the "reasoning" of the language models follow the Associate, Verify, Plan, and Feedback pattern. This measure needs at least some justification, but none is given.


# Novelty

Maze navigation and path planning in particular benchmarks for LLMs already exist, for instance in [7].
The novelty is somewhat limited by the abundance of similar benchmarks.
Having said that, some of the tasks presented are novel in this context.

# Scholarship

I am missing the formal definitions for the problems tackled in this paper and the references to the respective formalisms that are used to formally capturing them.

Further, the missing references that are mentioned above:
[1] Liu et al., ICLR 2024, AgentBench: Evaluating LLMs as Agents
[2] Ma et al., NeurIPS 2024, AgentBoard: An Analytical Evaluation Board of Multi-turn LLM Agents
[3] Handa et al,. ICLR 2025. ActionReasoningBench: Reasoning about Actions with and without Ramification Constraints
[4] He et al., ACL 2023, Exploring the Capacity of Pretrained Language Models for Reasoning about Actions and Change
[5] Kokel et al., AAAI 2025, ACPBench: Reasoning about Action, Change, and Planning
[6] Kokel et al., LM4Plan@AAAI 2025, ACPBench Hard: Unrestrained Reasoning about Action, Change, and Planning
[7] Palod et al., Arxiv 2025, Performative Thinking? The Brittle Correlation Between CoT Length and Problem Complexity


# Clarity

The paper is relatively easy to follow.

# Evaluation and Reproducibility

As the tasks themselves are not described, only examples are given, and are aggregated into 4 categories, we are only shown the evaluation results for a category. These numbers are not very informative, as we do not know what tasks do they correspond to. The comparison between models is done on the aggregated scores and not on the per-task basis, and therefore I find it somewhat less justifiable.

**Strengths:**

1. Non static benchmark, with dynamically generated tasks
2. Large collection of various tasks
3. Diverse set of sequential decision making problems

**Weaknesses:**

1. The generator has only one parameter, it seems to generate only a single task per parameter, so the variability is actually quite limited.
2. There is no formal analysis of the classes the tasks belong to.
3. The novelty is somewhat limited by the existence of several similar benchmarks.
4. The benchmark is not sufficiently justified - why is it important to measure the abilities of language models to solve sequential decision making tasks if they are notoriously bad at it and not designed to solve such tasks.
5. The evaluation is not sufficiently informative, some of the evaluation metrics are somewhat questinable and not sufficiently motivated.
6. There is no description or measure of the effort that took to manually implement all these tasks in the benchmark set. In other words, there is no discussion of how large of an effort it is to extend the benchmark set with additional tasks.

**Questions:**

1. What is the motivation of aggregating together various sequential decision making tasks under the same benchmark? What is the benefit over separate evaluations (e.g., path planning, classical planning, contingent planning, general game-playing) beyond the ease of conducting experiments?
2. How do you differentiate your work from AgentBench [1] and AgentBoard [2]?
3. How do you validate solutions? Did you have to implement a separate validator for each task?

---

> ### Author Response · Authors · 2025-11-21
> **Reply to Weakness 1**
>
> Dear reviewer,
>
> Thanks for your feedback. We address your questions point by point as follows. If we have any misunderstanding, please feel free to let us know and we will reply quickly.
>
> > **Weakness 1: Mechanism of the Generator**
>
> We must clarify a fundamental misunderstanding regarding the working principle of our Generator. Your assumption that "one parameter equals one task" is incorrect.
>
> Your criticism rests on a flawed premise that equates the complexity parameter $n$ with a single unique task instance. Our Generator is not a simple "template filler"; it is a **Procedural Generation Engine**. The parameter $n$ serves solely to control the level of complexity (e.g., the number of players in *Find the Impostors*). For any fixed value of $n$, our Generator is capable of programmatically producing a massive amount of distinct, unique instances.
>
> This is explicitly confirmed in Section 3.2 of our paper. Our final dataset was created by generating "30 distinct problems" for every difficulty level of each of the 40 tasks. For instance, in a "Simple" task scenario (assuming $n=6$), the model confronts not a single task, but 30 distinct, procedurally generated task instances. Our total dataset comprises 3,600 instances (40 tasks × 3 difficulty levels × 30 instances/level).
>
> Therefore, your concerns regarding "very limited variability" and "potential for memorization" are unfounded. On the contrary, our framework possesses extremely high variability. Models cannot solve problems through "memorization"; they must master generalizable reasoning logic to tackle the countless new instances that our Generator can produce (even under a fixed $n$).

---

> ### Author Response · Authors · 2025-11-21
> **Reply to Weakness 2**
>
> > **Weakness 2: Formal Analysis of Task Categories**
>
> We disagree with the claim that there is "no formal analysis of the classes."
>
> One of the core contributions of our paper is precisely the detailed theoretical grounding and formal analysis provided for these four reasoning categories (Inductive, Abductive, Deductive, Planning) in the latter half of Section 3.1 (DATA CLASSIFICATION), starting from "By leveraging four distinct task categories...".
>
> Your criticism that "path planning is not deductive reasoning" is based on the classical AI classification of "problem domains." However, our classification is grounded in the **"cognitive processes"** required for the model to solve the problem.
>
> Our analysis and classification rationale are as follows:
>
> ### 1. Inductive Reasoning (Information Probing)
> * **Analysis:** We define this as "forming general conclusions by identifying patterns from specific observations."
> * **Task Mapping:** "Information Probing" (IP) tasks (e.g., *Find the Impostors*) map directly to this. The model must collect evidence through limited, local queries (observations) and synthesize these observations to infer the complete, hidden global distribution (conclusion).
>
> ### 2. Abductive Reasoning (Dynamic Adaptation)
> * **Analysis:** We define this as "inferring the most plausible explanation from limited or incomplete evidence."
> * **Task Mapping:** In "Dynamic Adaptation" (DA) tasks (e.g., *Password Breaker*), the answer evolves dynamically according to rules. The model cannot directly observe the state; it must reverse-engineer the current most probable hidden state based on limited interactive feedback.
>
> ### 3. Deductive Reasoning (State Operation)
> * **Analysis:** We define this as "deriving specific conclusions through the application of known rules or logical implications."
> * **Task Mapping:** We classify "State Operation" (SO) tasks (e.g., *Maze Navigation*) as deductive because these tasks introduce hidden mechanisms. The model must first infer these hidden rules (e.g., "U/D keys are swapped") through interaction, and then apply these known rules to perform subsequent deductive reasoning to plan a correct path.
>
> ### 4. Planning (Strategic Gaming)
> * **Analysis:** We define this as "constructing action sequences by anticipating future states and considering opponent reactions."
> * **Task Mapping:** "Strategic Gaming" (SG) tasks (e.g., *Knight Battle*) explicitly require the model to perform multi-step planning to outmaneuver the system.
>
> Therefore, our classification is far from "PR"; it is a rigorous, top-down design based on cognitive science theory, aiming to systematically isolate and evaluate specific capabilities across different cognitive dimensions.

---

> ### Author Response · Authors · 2025-11-21
> **Reply to Weakness 3**
>
> > **Weakness 3: Novelty Analysis**
>
> We respect your opinion and acknowledge the rapid development in the field of interactive evaluation. However, we must firmly point out that classifying EvolArena as "one of many similar benchmarks" is a misinterpretation of the core novelty of our work. This view is based on a superficial similarity (e.g., both involve "interaction" or "reasoning"). In reality, EvolArena differs fundamentally from other benchmarks in terms of evaluation objectives, interaction paradigms, and evaluation mechanisms, filling a unique and critical gap.
>
> We provide a detailed comparison of EvolArena with the related papers you mentioned:
>
> ### 1. Evaluation Objective: Pure Logic vs. Real-World/Formal Skills
> EvolArena is positioned as **"pure, abstract logical reasoning."** This is distinct from the goal of other benchmarks that test "applied skills."
>
> * **vs. Real-World Agents (AgentBench, AgentBoard):**
>     AgentBench and AgentBoard evaluate LLMs' "agent" capabilities in noisy, open-ended, unstructured real-world environments. They test whether models can interact with real software (e.g., operating systems, databases, web browsers).
>     **EvolArena's Distinction:** We deliberately strip away all this real-world "noise." EvolArena provides a closed, deterministic, rule-clear "logic laboratory." We do not test whether a model can use bash commands or browse the web; we only test its ability to perform multi-step reasoning in pure logic games like *Find the Impostors* or *Password Breaker*.
>
> * **vs. Classical AI Planning (TRAC, ACPBench, ActionReasoningBench):**
>     These benchmarks are constructed by the classical AI Planning community to test whether LLMs understand formal planning theories. Their core involves PDDL, STRIPS, Preconditions, Effects, and "Ramifications."
>     **EvolArena's Distinction:** Our tasks (e.g., strategic games, dynamic puzzles) do not require models to possess any formal knowledge of classical AI planning. EvolArena tests a more general, intuitive logical reasoning capability rather than proficiency in a specific academic domain (like PDDL).
>
> ### 2. Interaction Paradigm: Dynamic Multi-Turn vs. Static QA
> This is the core novelty of EvolArena: we evaluate **"dynamic multi-turn conversational reasoning."**
>
> * **Paradigms of TRAC, ACPBench, ActionReasoningBench:**
>     These benchmarks are essentially static, single-turn QA. They provide the LLM with a complete description of the world state and then ask a complex question about that state (e.g., "Is this plan valid?", "List all available actions", "Will the 'clear' property change due to Ramifications?").
> * **EvolArena's Paradigm:** Our paradigm is **fundamentally multi-turn**. The model cannot see the complete world state; it must go through a 15-turn interaction process, like playing a "guessing game," actively proposing exploratory queries to the Monitor, observing feedback, and then using this feedback to decide its next action.
>
> ### 3. Evaluation Mechanism: Evolving vs. Static
> EvolArena's **"Evolving"** characteristic is its core mechanism against benchmark saturation, which other benchmarks lack.
>
> * **Static Benchmarks (AgentBench, ACPBench, etc.):**
>     These benchmarks (whether from PDDL or not) are based on a fixed set of problem instances. This inevitably exposes them to risks of **data contamination** and **performance saturation**.
> * **EvolArena's "Evolving" Mechanism:**
>     Our core is the parametric Generator. By adjusting the complexity parameter $n$, we can programmatically generate infinite, completely new instances with controllable difficulty. This gives EvolArena an extremely long lifecycle, capable of "evolving" harder challenges as model capabilities improve, fundamentally solving the data contamination issue of static benchmarks.
>
> **Conclusion:**
> EvolArena's novelty is clear and unique. It is the first benchmark to simultaneously feature:
> 1. **Objective:** Focus on pure, abstract logical reasoning (rather than real-world skills or formal planning theory).
> 2. **Paradigm:** Must be solved through dynamic, incremental **multi-turn interaction (dialogue)**.
> 3. **Mechanism:** Fully automated, deterministic (non-LLM judge), and evolvable (procedural generation) to counter data contamination.
>
> **P.S.:** As for *PERFORMATIVE THINKING?*, it is not a competing benchmark. Instead, it is a paper critiquing "Chain-of-Thought (CoT)," supporting our core motivation: that passive, monologue-style "Chain-of-Thought" is unreliable, which is why we need an interactive benchmark based on action and feedback like EvolArena to truly evaluate reasoning.

---

> ### Author Response · Authors · 2025-11-21
> **Reply to Weakness 4**
>
> > **Weakness 4: LLM Suitability for Sequential Reasoning Tasks**
>
> We must respectfully but firmly push back against this critique. We believe that the view that "LLMs are not designed for this" and "they are bad at it" is precisely the **fundamental reason** for the existence of EvolArena and similar benchmarks like AgentBench and ACPBench.
>
> Our motivation is not grounded in the current perfection of LLMs, but rather in the chasm between their demonstrated massive potential and their severe deficiencies.
>
> ### 1. Assessing an Undeniable Emerging Paradigm
> The assertion that LLMs are "not designed for this" overlooks a critical reality: LLMs are being widely repurposed across both industry and academia as the "brain" of general-purpose agents. This is already an established fact.
>
> * **New Paradigm:** Classical AI focused on building specialized solvers for specific problems (e.g., path planning, game theory). The new paradigm of LLM agents tests whether a single, general-purpose model can autonomously understand rules and make decisions across diverse interactive environments.
> * **Filling the Gap:** As our paper points out, existing evaluations (like MMLU, GSM8K) are predominantly static and single-turn. The core motivation behind works like EvolArena, AgentBench, and AgentBoard is to fill this massive void in "interactive evaluation."
>
> ### 2. "Notoriously Bad" Requires Quantification
> You claim that LLMs are "notoriously bad" at sequential decision-making. This is exactly **why** this benchmark is valuable. Scientific progress relies on precise diagnosis, not just "notorious" intuition.
>
> * **Diagnosing Failure:** The purpose of EvolArena is not to showcase high scores, but to diagnose failure. Our experimental data (along with ACPBench Hard, ActionReasoningBench, etc.) collectively proves that even frontier models (like GPT-4o, o1-preview) collapse completely when facing long-horizon reasoning, causal chains (Ramifications), or complex composite problems.
> * **Providing Signals:** Without benchmarks like EvolArena to systematically and automatically reveal these defects (e.g., o3-mini's high accuracy accompanied by extremely low efficiency), the research community would lack clear signals to guide the improvement of the next generation of models.
>
> ### 3. Aggregation Is a Scientific Goal in Itself
> You question the motivation for "aggregation." We must emphasize that aggregation itself is a scientific objective. Classical AI strictly divides problems into "path planning," "game theory," etc. But real-world problems are fuzzy and hybrid. By aggregating 40 tasks across 4 categories (and AgentBench's 8 environments), EvolArena tests the **General-purpose Reasoning** capability of a single model. We are not evaluating a model's depth in a narrow track, but its generalization ability and adaptability when facing entirely new rules and diverse scenarios.
>
> In summary, because LLMs are **being used** as agents, they **must** be evaluated. Because they perform poorly on these tasks, we have an even greater need for diagnostic benchmarks like EvolArena to pinpoint the reasons for their failure.

---

> ### Author Response · Authors · 2025-11-21
> **Reply to Weakness 5**
>
> > **Weakness 5: Concerns Regarding Evaluation Metrics**
>
> We firmly dispute the criticism that our evaluation metrics are "questionable" and "insufficiently informative." We believe this critique stems from a misunderstanding of our multi-dimensional evaluation framework and overlooks the supplementary data provided in the Appendix.
>
> EvolArena employs a diagnostic evaluation composed of four independent axes (Accuracy, Efficiency, Invalid Rate, Pattern Analysis), which provides significantly more information than a single aggregated score.
>
> ### 1. Clarifying the Efficiency Metric: Win Rate vs. Average Turns
> Your criticism of the "Efficiency" metric is partial, as it only considers the first layer of our analysis.
> The example you provided (A=[5,5,5] vs B=[4,4,100]) correctly identifies the characteristic of our "pairwise comparison" metric in Figure 4. This metric (defined in Section 2) measures "head-to-head win rate" (B wins on 2/3 tasks), not "average turns" (where A's average is much lower than B's). We use this metric to demonstrate the consistency of models in finding better paths on *individual problems*.
>
> However, to provide a complete global picture unaffected by outliers (like B=[100]), we also reported the "average interaction turns" you expected in **Table 4 of Appendix I**.
>
> * Table 4 shows the average turns for o3-mini, R1, and QwQ-32B on all jointly solved problems.
> * The data indicates (e.g., R1 averages 6.94 turns, while o3-mini averages 8.97) that R1 is indeed more efficient on average, confirming the trend in Figure 4.
>
> By providing **two complementary efficiency metrics** (win rate consistency and average turns), our evaluation offers a richer and more robust analysis than a single metric.
>
> ### 2. Clarifying the Motivation for Pattern Analysis
> You claim that "Pattern Analysis" lacks motivation. On the contrary, this metric is well-justified as our core tool for diagnosing *why* models fail.
> We selected these four patterns (Associate, Verify, Plan, Feedback) because they directly map to the core cognitive cycle required for solving multi-turn interaction problems:
> * **Associate:** Does the model remember the original problem? (Context Retrieval)
> * **Plan:** Is the model formulating a strategy for the next step? (Forward Thinking)
> * **Feedback:** Is the model utilizing the Monitor's feedback from the previous round? (State Update)
> * **Verify:** Is the model checking its own reasoning process? (Metacognition/Self-Correction)
>
> The validity of this metric is empirically demonstrated in Section 4.6. The data shows that stronger reasoning models (QwQ-32B and R1) score significantly higher on "Verify" and "Feedback" than distilled models (R1-Distill-32B). This proves that our pattern analysis successfully captures the key capabilities distinguishing strong from weak reasoners, making its motivation and informational value sufficient.
>
> ### 3. Sufficiency of Evaluation
> We firmly believe that our evaluation framework is one of the most informative available. Instead of relying on a single, potentially misleading metric, it provides a diagnostic profile containing four orthogonal dimensions:
> * **Can it solve the problem?** (Accuracy)
> * **How fast does it solve it?** (Efficiency - Win Rate + Average)
> * **Does it follow the rules?** (Invalid Rate)
> * **How does it think?** (Pattern Analysis)
>
> This multi-dimensional analysis (Accuracy + Efficiency + Robustness + Process) provides a complete, interpretable view of model capabilities, which is by no means "insufficiently informative."

---

> ### Author Response · Authors · 2025-11-21
> **Reply to Weakness 6**
>
> > **Weakness 6: Workload and Scalability of the Benchmark**
>
> We thank the reviewer for raising this important question regarding workload and scalability. We have divided our work into two phases:
> 1.  One-time framework development (building the EvolArena pipeline).
> 2.  Reproducible task implementation (adding new tasks to the framework).
>
> ### 1. Initial Implementation Workload
> The overall framework development required approximately 120 hours. Integrating the 40 seed problems into our benchmark involved a **"manual implementation" workload of about 3 hours per task, totaling 120 hours.** It is crucial to emphasize that this work is not data annotation, but **software development**. As stated in Appendix E, we did not "copy" the problems. Our core work involved **fundamentally transforming** static logic puzzles from Codeforces into novel dynamic interactive tasks.
>
> For each of these 40 tasks, we manually executed the following:
> 1.  Designing a new set of interactive rules and feedback mechanisms.
> 2.  Implementing a **Generator** script capable of procedurally creating new instances and answers for the task.
> 3.  Implementing a **Monitor** script that acts as a deterministic "game engine" to validate model inputs (format and logic) and return feedback based on rules.
> 4.  Implementing an **Evaluator** script to define the evaluation metrics and success conditions for that specific task.
>
> This upfront investment was indeed substantial, but it is precisely the core contribution of our benchmark: we have built 40 reusable, programmable "reasoning environments."
>
> ### 2. Workload for Extending the Benchmark
> Your question about scalability precisely highlights the value of the EvolArena framework. Thanks to the aforementioned "one-time framework development," the marginal cost of extending EvolArena is fixed, one-time, and significantly lower than creating a static benchmark of equivalent scale.
>
> To add a new task (e.g., the 41st task) to EvolArena:
> * **Required Effort:** Researchers do not need to manually create 30 (or 3000) instances of varying difficulty. They only need to implement three new programmatic scripts (Generator, Monitor, Evaluator) to define the logic of this new task, just as we did.
> * **Scalability:** Once these three scripts are added to the framework, our pipeline can automatically generate an infinite number of new instances (as described in Section 2) and automatically run evaluations on all models.
>
> **Conclusion:** Our approach replaces **"continuous, large-scale manual annotation costs"** (manually writing thousands of problems) with **"one-time, high-skill programming costs"** (implementing logic for a new task). We firmly believe that although the initial investment is high, this method offers superior **scalability, reproducibility, and resistance to contamination**, making it a better solution for future benchmarking.

---

> ### Author Response · Authors · 2025-11-21
> **Reply to Question 1**
>
> > **Question 1: Aggregation as Convenience**
>
> We firmly push back against this critique. We believe that the view that "aggregation is for convenience" and that LLMs are "not designed for this" stems from a fundamental misunderstanding, conflating the evaluation goals of "classical AI solvers" with those of "general-purpose LLM agents."
>
> Our motivation is precisely not "convenience," but to evaluate the capabilities of a novel, singular **"General Problem Solver."**
>
> ### 1. Evaluation Goal: "Generalist Agent" vs. "Specialized Solver"
> The "separate evaluations" (such as classical planning or game playing) you advocate for are designed to assess the performance of a **specialized solver** on *known* problem categories.
>
> EvolArena's motivation is the exact opposite. Our goal is to evaluate the **adaptability** of a single, **generalist agent** when facing unknown and diverse challenges.
>
> Real-world problems do not come pre-labeled as "this is a path planning problem" or "this is an abductive reasoning problem." Therefore, aggregating these 40 tasks from different domains is specifically intended to simulate this real-world challenge: an LLM must, as a single agent, autonomously understand a completely new interactive environment and formulate strategies solely by reading the rules (Prompt).
>
> This is a novel "meta-reasoning" capability that "separate evaluations" cannot measure.
>
> ### 2. Diagnostic Value: "Why" It Failed, Not "Where" It Failed
> You point out that LLMs are "notoriously bad" at these tasks. This precisely reinforces the necessity of our benchmark. Scientific progress relies on precise diagnosis, not just "notorious" intuition.
>
> If we were to test on 40 separate domains, we would at best derive 40 isolated conclusions (e.g., "the model is bad at playing Knight Battle").
>
> The true benefit of aggregated evaluation lies in **"Diagnosing Cross-Domain Defects"**: By aggregating 40 tasks, we can identify the common, fundamental cognitive bottlenecks behind model failures. Our analysis (along with AgentBench's analysis) has proven that model failure is not because they are bad at a *specific* game, but because they exhibit ubiquitous issues:
> * **Poor long-horizon reasoning and planning capabilities**
> * **Inability to use environmental feedback to update internal states**
> * **Weak basic instruction following capabilities (leading to invalid operations)**
>
> In summary, we aggregate these tasks not for "convenience," but to scientifically evaluate the generalization ability and adaptability of LLMs as general-purpose agents. This approach allows us to distill common, actionable reasons for failure (such as "insufficient long-horizon planning capability") from 40 different tasks, something separate evaluations cannot provide.

---

> ### Author Response · Authors · 2025-11-21
> **Reply to Question 2**
>
> > **Question 2: Differentiation from Related Works**
>
> EvolArena differs fundamentally from AgentBench and AgentBoard in terms of evaluation objectives, core mechanisms, and interaction paradigms.
>
> ### 1. Evaluation Objective: Pure Logic vs. Real-World Agents
> The core objective of AgentBench and AgentBoard is to evaluate LLMs as **"General-purpose Agents"** in **simulated real-world** applications.
> * Their environments are noisy, open-ended, and complex:
>     * **AgentBench** includes operating systems (Ubuntu), databases (MySQL), and web browsing.
>     * **AgentBoard** includes Embodied AI, Games, Web, and Tool use.
> * In these benchmarks, models must master **"application skills"** such as writing bash commands, executing SQL queries, or parsing HTML.
>
> **EvolArena's Distinction:** Our goal is to evaluate pure, abstract, intrinsic logical reasoning capabilities.
> * We deliberately strip away all real-world noise and dependencies on specific skills (like coding).
> * Our environments are closed, deterministic, rule-clear "logic laboratories" (e.g., "Find the Impostors"). We do not test whether a model can use bash; we only test its ability to perform multi-step induction, deduction, and planning in pure logic games.
>
> ### 2. Evaluation Mechanism: Evolvable Generation vs. Static Human Annotation
> AgentBoard and EvolArena take two completely different technological routes in how they evaluate "process."
> * **AgentBoard's** core innovation is the **"Fine-grained Progress Rate."** The implementation of this mechanism relies heavily on expensive and subjective "human-annotated subgoals."
> * **EvolArena's** core innovation is **"Evolvability."** Our framework is fully automated, requiring no human annotation. Our **Generator** can programmatically and infinitely generate new instances with parameterized difficulty. This fundamentally solves the problems of **data contamination** and **performance saturation** faced by static benchmarks (including AgentBench and AgentBoard).
>
> ### 3. Interaction Paradigm: Dynamic Querying vs. Open-World Action
> * In **AgentBench** and **AgentBoard**, interaction typically involves the model performing high-risk actions in complex, partially observable environments (such as clicking links in WebArena or moving in ALFWorld).
> * In **EvolArena**, the interaction paradigm is fundamentally multi-turn dialogue/querying. The model must go through a "probe-observe-deduce" cycle, actively proposing exploratory queries to the Monitor to gain information, and then using this feedback to update its internal state and decide on the next action.
>
> **Conclusion:** EvolArena's novelty lies in being the first benchmark to systematically isolate and measure pure abstract logic through a fully automated, deterministic, and evolvable framework.

---

> ### Author Response · Authors · 2025-11-21
> **Reply to Question 3**
>
> > **Question 3: Did you implement a separate validator for each task?**
>
> **Yes, we implemented separate, customized validation logic for each of the 40 tasks.** Within our framework, this "validation" logic is decoupled into two key task-specific components: the **Monitor** and the **Evaluator**.
>
> ### 1. Monitor: Real-time Verification During Interaction
> The Monitor acts as the "referee" for each task. As detailed in Section 3.2, we "tailored monitors to each task's interactive rules."
>
> * **Mechanism:** In every interaction turn, the Monitor receives the model's output (e.g., "My Query: 1,2,3").
> * **Validation:** It immediately verifies the output's (1) format validity and (2) logical validity based on the specific rules of that task (e.g., rules for "Find the Impostors").
> * **Feedback:** It calculates and returns deterministic feedback (e.g., "1" or "0") based on the hidden "reasoning objective" (i.e., the answer).
>
> ### 2. Evaluator: Final Verification After Interaction
> The Evaluator performs the final "solution verification" after the entire interaction concludes. As stated in Section 3.2, we designed "task-specific evaluators."
>
> * **Mechanism:** When the Generator creates a problem, it also generates the corresponding "reasoning objective" (ground truth).
> * **Validation:** The Evaluator receives the complete dialogue history and checks whether the model's final state $s_{T}$ (e.g., its "My Answer: ...") matches the "reasoning objective" $s$ provided by the Generator.
> * **Result:** The task is judged as "Successful" (Accuracy=1) only if the two match. The Evaluator also conducts a comprehensive analysis based on the definitions of other metrics.
>
> **Conclusion:** We implemented separate Monitor and Evaluator scripts for each of the 40 tasks because every task possesses unique interactive rules, format requirements, and success conditions.

---

> ### Author Response · Authors · 2025-11-21
> **Reply to Question 1 in Summary**
>
> > **Question 1: Misconceptions Regarding the Generator**
>
> We need to clarify that your criticism of our Generator stems from two core misunderstandings: first, conflating the "complexity parameter $n$" with being the "sole parameter," and second, overlooking our "difficulty calibration" process.
>
> ### 1. Clarification on (a) "Limited Generator" and "Memorization"
> You worry that our generator is "very limited" and might lead to "memorization." This concern is based on a flawed premise that equates the "complexity parameter $n$" with a single, unique task instance.
>
> * **$n$ Is Not the Sole Parameter:** As described in Section 2 of the paper, our generator is formalized as $p,s=P(t,n,g_{n})$. Here, $n$ is merely a meta-parameter controlling the level of complexity (e.g., the number of players in *Find the Impostors*). In contrast, $g_n$ represents all "corresponding problem parameters," such as the specific locations of impostors, the actual layout of a maze, etc.
> * **Procedural Generation vs. Memorization:** Our generator is procedural. For any fixed value of $n$, our generator is capable of programmatically producing a massive number of distinct, unique instances.
> * **Empirical Evidence:** This is explicitly confirmed in Section 3.2 of our paper. Our final dataset was created by generating "30 distinct problems" for every difficulty level of each of the 40 tasks. If $n=6$ corresponded to only one instance, this would have been impossible. This design (where one $n$ corresponds to a vast instance space) is specifically intended to prevent the "memorization" issues you fear.
>
> ### 2. Clarification on (b) "Difficulty Scaling Fallacy"
> You raise a valid point: in some cases, a "larger $n$" does not necessarily imply "harder" (e.g., expanding a maze without adding more obstacles).
> However, this criticism does not apply to EvolArena because we implemented rigorous "empirical calibration" to ensure the validity of our difficulty gradients.
>
> 1. **Parameter Co-scaling:** Our generator $P(t,n,g_{n})$ synergistically scales all relevant difficulty parameters $g_n$ as $n$ increases. We do not commit errors such as enlarging a maze without increasing obstacles. For example, in *Find the Impostors*, increasing $n$ (total players) simultaneously increases both the search space and the number of targets.
> 2. **Empirical Verification:** We did not *assume* that a larger $n$ implies greater difficulty. As described in Section 3.2, we **"iteratively refined"** the values of $n$ and used o3-mini to **"evaluate task solvability"** until the difficulty gradient **"exhibits meaningful progression."**
> 3. **Proof in Final Results:** Our calibration was successful. As stated in Section 4.2: "Across all models, performance decreases progressively from 'Easy' to 'Medium' to 'Hard.' This demonstrates the rationality of our dataset's difficulty stratification."
>
> **Conclusion:** The Generator in EvolArena is a sophisticated procedural engine, not a simple template filler. Through the separation of complexity parameters ($n$) from instance parameters ($g_n$) and rigorous empirical calibration, we ensure both high variability to prevent memorization and valid difficulty scaling.

---

> ### Author Response · Authors · 2025-11-21
> **Reply to Question 2 in Summary**
>
> > **Question 2: Issues Regarding Problem Definition**
>
> We firmly refute the criticism regarding a "lack of formal definitions."
>
> We have explicitly provided formal definitions for our evaluation framework and interactive reasoning dynamics in **Section 2** and **Appendix B** of the paper.
>
> The formalisms you might be seeking are likely those of traditional "classical planning" (such as PDDL or STRIPS). However, EvolArena has a different evaluation objective, and thus our formalization differs. Our formalization is based on **interactive decision-making processes**, rather than static state-space search.
>
> Specifically:
>
> ### 1. Framework Formalization (Section 2)
> In Section 2, we have mathematically formalized the three core components of EvolArena:
> * **Generator (P):** $p,s=P(t,n,g_{n})$
> * **Monitor (M):** $(m_{i},s_{i})= M(t,q_{i},s_{i-1},s,I)$
> * **Evaluator (E):** $e= E(t,\{(q_{1},m_{1}),...,(q_{T},m_{T})\} )$
>
> This set of definitions provides a rigorous, reproducible mathematical foundation for our automated evaluation pipeline.
>
> ### 2. Formalization of Reasoning Dynamics (Appendix B)
> In Appendix B, we further provide a dynamic model for "multi-turn reasoning" itself. We define the model's query $q_i$ as the output of its policy function $f_{\theta}$ given the initial problem $p$ and interaction history $\mathcal{H}_{i-1}$:
>
> $q\_{i}=f\_{\theta}(p,\mathcal{H}\_{i-1})$
>
> This explicitly formalizes our tasks as a **partially observable, history-dependent decision process**, which is precisely the core capability we aim to evaluate.
>
> **Conclusion:** Therefore, we believe the paper provides sufficient formal definitions. The reason they may have appeared "missing" is likely due to an expectation for literature citations and formalisms associated with classical AI planning (such as ACPBench or TRAC), whereas our work is built upon a different, **dynamic interactive** formalization paradigm.

---

> ### Author Response · Authors · 2025-11-21
> **Reply to Question 3 in Summary**
>
> > **Question 3: Misconception Regarding Task Descriptions**
>
> We must firmly refute this criticism as it rests on a completely erroneous premise. Your assertion that "the tasks themselves are not described, only examples are given" is factually incorrect.
>
> We have provided exhaustively detailed task descriptions within the paper. It appears that our **Appendix N: TASK INTRODUCTION** was completely overlooked.
>
> * **Extensive Coverage:** This appendix spans nearly 40 pages, extending from page 30 to page 68 of the manuscript.
> * **Comprehensive Templates:** It does not merely provide "examples"; instead, it supplies complete **"Problem Templates"** for every single one of the 40 tasks.
> * **Rigorous Specifications:** These templates explicitly list the **interaction rules, query format requirements, and example interactions** for each task.
> * **Depth of Definition:** For instance, tasks such as "Find The Impostors" (Case K.1) and "KnightBattle" (Case K.61) each have more than a full page dedicated to detailing their rules and interaction mechanics.
>
> **Conclusion:** Therefore, the assertion that tasks are undescribed is directly contradicted by the facts presented in the manuscript.

---

> ### Author Response · Authors · 2025-11-21
> **Reply to Question 4 in Summary**
>
> > **Question 4: Motivation for Aggregated Scores**
>
> You argue that our comparison based on "aggregated scores" rather than a "per-task basis" is unjustifiable. We respectfully contend that this critique reverses the logical priority.
>
> At the scale of 40 diverse tasks, a primary "per-task" analysis would actually be **unreasonable** and **less informative**. A raw score matrix involving 40 tasks and 20 models would result in a massive table filled with high-variance "noise." Deriving any statistically reliable conclusions regarding a model's **"general reasoning capability"** from such granular results would be precarious.
>
> The core analytical value of EvolArena lies precisely in our **aggregation**. Our four categories (IP, DA, SO, SG) are not arbitrary groupings; they represent specific cognitive capabilities we aim to measure (e.g., "dynamic adaptation" or "strategic planning"). By aggregating, we filter out task-specific noise, enabling us to diagnose **systemic, cross-domain cognitive defects** in models. Our conclusion—for example, that models universally fail in the "Dynamic Adaptation (DA)" category—is a powerful, generalizable insight that a fragmented per-task analysis could not provide.
>
> Nevertheless, to ensure our evaluation is maximally informative and unimpeachable, we have added a new **Appendix O**, presenting the detailed accuracy of all models across all 40 independent tasks. This ensures our analysis possesses both the high-level insights of aggregation in the main text and the fine-grained transparency of per-task results in the appendix.

---

### Official Review · Reviewer_7c9C · 2025-10-28

**Soundness:** 3
**Presentation:** 3
**Contribution:** 3
**Rating:** 4
**Confidence:** 4

**Summary:**

EvolArena is a new benchmark + automated evaluation framework for multi-turn interactive reasoning of LLMs. The authors (i) define 4 task classes (Information Probing, Dynamic Adaptation, State Operation, Strategic Gaming), (ii) convert 40 seed problems into structured interactive templates, (iii) implement a Generator / Monitor / Evaluator pipeline that automatically produces instances at three difficulty levels (easy/medium/hard) and runs deterministic interactions, and (iv) evaluate ~20 models (reasoning-specialized and general) reporting accuracy, efficiency (turns), invalid-operation rates, and pattern analyses. Main claims: multi-turn interactive reasoning remains hard for SOTA models, their framework scales and can be customized via tunable generators, and it provides diagnostic metrics for real weaknesses.

**Strengths:**

1. The paper provides an interesting way of sourcing multi-turn problems from interactive games / puzzles.
2. The framework provides diverse evaluation metrics and identify specific reasoning patterns used in the model response.
3. Though the generated tasks are closed-world and contrived, the authors address this limitation directly and reasonably.

**Weaknesses:**

1. The paper can benefit from more details about the generator, monitor, and evaluator. E.g. the monitor is rule-based (what are these rules) and checks for formatting (what kind); the evaluator evaluates for pattern analysis (how). These information would be helpful in the appendix.
2. The paper could elaborate more on how this benchmark can be considered "evolving". Based on the work, it seems like here “evolving” = procedurally extensible (generate from seed problems) + parameterized difficulty scaling (control difficulty parameter). However, it seems like up to the choice of seed problem and difficulty parameter, the benchmark is rather static and not adaptive to the agent's capabilities. Could the authors please clarify this word choice? I think a more substantial explanation of the evolving nature of this framework would better highlight the novelty of this work.
3. Several formatting issues in the paper (e.g., figure 4 caption is cut off, inconsistent bolding in table 1 caption, etc.

**Questions:**

1. The seed tasks are obtained from publicly available sources. Could you speak to whether this creates contamination and whether this could be an issue?
2. See "evolving" comment in weaknesses
3. Could the authors clarify which of the generator and evaluator are LLM-based. For those that are, could the authors provide an estimate on token costs for the generator?

---

> ### Author Response · Authors · 2025-11-21
> **Reply to Weakness 1**
>
> Dear reviewer,
>
> Thank you for your valuable suggestions. We are encouraged that you found our paper interesting. We address your questions point by point as follows. If we have any misunderstanding, please feel free to let us know and we will reply quickly.
>
> > **Weakness 1: Details about the generator, monitor, and evaluator.**
>
> We sincerely appreciate this constructive feedback. We fully agree that enhancing the technical transparency of the framework components (Generator, Monitor, Evaluator) is crucial for reproducibility.
>
> We would like to first clarify that the paper already contains some of the core details requested, although we acknowledge they could be more exhaustive and systematic.
>
> ### 1. Details Already Present in the Paper
> * **Monitor (Rules & Formats):** In Section 3.3, we detailed the Monitor's mechanism using the "Find the Impostors" task as an example. We explicitly stated that the Monitor validates two specific response patterns: "My Query: a, b, c" and "My Answer: $x_{1},x_{2},...,x_{k}$". Any non-matching response is rejected. Furthermore, it provides deterministic feedback ("1" or "0") based on the hidden answer sequence (the rule).
> * **Evaluator (Pattern Analysis):** Regarding how Pattern Analysis is conducted, we defined the four patterns (Associate, Verify, Plan, Feedback) and their calculation formulas in Section 2. More importantly, in Section 4.6, we explicitly stated that this analysis is measured "using Qwen2.5-72B as an analyzer."
> * **Generator:** Similarly, in Section 3.3, we explained how the generator for "Find the Impostors" creates problems by varying parameter $n$ and generating binary target sequences.
>
> ### 2. Supplementary Detailed Implementation in Appendix
> Although we provided the above examples, we fully agree with your view that providing more systematic and comprehensive details would significantly enhance the paper's quality.
>
> To achieve full technical transparency, we have added a new **Appendix L** in the revised paper. In this section, we provide detailed implementation logic or pseudocode for one representative task from each category (IP, DA, SO, SG), such as "Password Breaker," "Maze Navigation," and "Knight Battle." This clearly demonstrates:
>
> * **Generator:** How it programmatically constructs task instances and answers using parameter $n$.
> * **Monitor:** The complete rule set, including all Regular Expression (Regex) formats it checks, and the logic for returning different feedback based on internal states.
> * **Evaluator:** Specific evaluation criteria (e.g., how "Victory" is determined in "Knight Battle").
>
> We believe this addition fully addresses your concerns regarding framework transparency.

---

> ### Author Response · Authors · 2025-11-21
> **Reply to Weakness 2**
>
> > **Weakness 2: Concerns About the "Evolving" Mechanism**
>
> We appreciate this critical inquiry. Your observation is keen: in our current experiments, "Evolving" is primarily manifested as "programmatic extensibility" and "parametric difficulty scaling". However, we must clarify that this design not only addresses the flaws of current static benchmarks but also serves as the necessary infrastructure to realize the "true adaptability".
>
> ### 1. Direct Meaning of "Evolving": Anti-Saturation
> Traditional benchmarks (e.g., GSM8K) face the critical issues of being **Static** and **Finite**. Once model capabilities improve or data contamination occurs, these benchmarks rapidly reach **Saturation**, losing their evaluative value.
>
> EvolArena's "Evolving" characteristic is designed to fundamentally solve this problem:
> * Our core is the parametric **Generator (P)**. We do not provide a fixed dataset but an engine capable of generating infinite new instances via the complexity parameter $n$.
> * The three levels in the current paper (e.g., $n=6, 9, 12$) are merely to demonstrate this capability. When community models (like o4 or o5) fully master the current "Hard" level, EvolArena can **immediately "evolve"** to $n=15, 20$ or higher difficulties without requiring any new human annotation or data collection.
> * Thus, "Evolving" here accurately means **"Evolvability"**—endowing the benchmark with an infinite lifecycle that grows alongside community model capabilities, never saturating.
>
> ### 2. Future Potential of "Evolving": Realizing True Adaptability
> We wish to clarify that EvolArena's current architecture (the parametric Generator and deterministic Monitor) is precisely the infrastructure built to realize this advanced adaptability. We detail this potential across three progressive levels:
>
> #### **2.1 Automated Curriculum Learning**
> This represents the most direct future application of "Evolving": transforming EvolArena from an **"examination venue" into a "gymnasium" for RL training.**
> * **Current Problem:** One of the major challenges in training reasoning agents is "Reward Sparsity." If tasks are too difficult, the model rarely succeeds and learns nothing; if too easy, learning is inefficient.
> * **EvolArena's Solution:** Our framework perfectly addresses this.
>     1. **Environment:** 40 tasks provide a diverse training ground.
>     2. **Reward:** The Monitor provides immediate, deterministic feedback (Success, Failure, Invalid), serving as a perfect reward signal.
>     3. **Curriculum:** The Generator provides a tunable "knob" for difficulty (parameter $n$).
> * **Implementation:** An external "Curriculum Controller" can be constructed to observe the model's win rate at current difficulty $n$. If the win rate $> 90\%$, the controller calls the Generator to increase difficulty to $n+1$; if $< 10\%$, it decreases to $n-1$. This ensures the model always trains within its "Zone of Proximal Development," maximizing training efficiency and the upper bound of reasoning capabilities.
>
> #### **2.2 High-Resolution Adaptive Evaluation**
> You mentioned "adapting to the agent's capabilities." This applies not only to training but also to evaluation.
> * **Current Problem:** Static, "one-size-fits-all" benchmarks suffer from low resolution. They struggle to differentiate subtle differences between two strong models (e.g., o3-mini vs. R1) that both solve "Hard" tasks, or distinguish between two weak models that both fail.
> * **EvolArena's Solution:** Our "Evolving" architecture supports **Model-Contingent Evaluation**.
> * **Implementation:** Instead of testing on fixed levels (e.g., $n=6, 9, 12$), the Evaluator can dynamically adjust $n$.
>     * For a 7B weak model, the Evaluator starts at $n=4$ and incrementally tests its **"capability inflection point"** (e.g., failure at $n=7$).
>     * For a strong model like o3-mini, the Evaluator starts at $n=12$ and **"evolves upward"** to $n=13, 14, 15...$, probing its capability ceiling.
>
> #### **2.3 Adversarial \& Strategic Evolution**
> This represents the frontier of the "Evolving" concept: the evolution of the **benchmark itself**.
> * **Current Problem:** Models may "overfit" or "game" a benchmark by learning specific task heuristics rather than general reasoning abilities.
> * **EvolArena's Solution:** Our Generator is controlled not only by the difficulty parameter $n$ but also by problem parameters $g_n$.
> * **Implementation:** We envision an "Adversarial Generator" that analyzes the failure logs of a specific model (e.g., o3-mini) to identify specific strategic blind spots (e.g., consistent failure in specific opening configurations of Knight Battle). The Generator then "evolves" to specifically produce more of these instances that effectively counter the model's current strategy.
>
> Moreover, we have clarified this distinction in Appendix C, expanding the definition of "Evolving" from "parametric scaling" to "infrastructure for adaptive assessment, training, and adversarial evolution."

---

> ### Author Response · Authors · 2025-11-21
> **Reply to Weakness 3**
>
> > **Weakness 3: Formatting Issues**
>
> We sincerely appreciate your keen observation and meticulous reading. We apologize for the formatting issues identified (such as the truncated caption in Figure 4 and the inconsistent bolding in Table 1) . We have corrected these issues in the revised manuscript and have conducted a thorough review to ensure no other formatting errors remain. We are grateful for this feedback, as it significantly contributes to enhancing the presentation quality of our work.

---

> ### Author Response · Authors · 2025-11-21
> **Reply to Question 1 & 2**
>
> > **Question 1: Potential Data Contamination**
>
> This is a crucial question that we considered a core design priority from the outset. Our conclusion is that while the risk of contamination exists, we have methodologically mitigated this issue through (1) fundamental task adaptation and (2) dynamic instance generation. Our experimental results (specifically the uniformly low scores across all models) also empirically refute the possibility of severe contamination.
>
> ### 1. Core Mitigation Strategy: Fundamental Shift from "Static Problems" to "Interactive Tasks"
> You expressed concern that models might have "seen" problems from Codeforces or NYT during pre-training. We must emphasize that we did not use these seed problems in their "static form."
>
> As detailed in Appendix E, every seed problem underwent a manual, radical adaptation (manually transformed), converting it into a completely new task that requires multi-turn interaction. This comprehensive adaptation process involved three key steps:
> 1. **Designing New Interaction Rules:** We designed a new set of interaction rules for each original problem, transforming it into a dynamic task that necessitates multi-turn interaction for resolution.
> 2. **Creating Problem Templates:** We manually created standardized templates, including clear rule descriptions, strict format requirements, and interaction examples.
> 3. **Developing Generators:** We developed corresponding generators based on these templates.
>
> Therefore, even if a model "memorized" the static solution to the original Codeforces problem during pre-training, that knowledge is insufficient to solve our interactive version with entirely new rules requiring multi-turn queries and feedback.
>
> ### 2. Generator Mechanism: Countering Specific Instance Contamination
> Secondly, our evaluation does not rely on a few specific samples from the original websites. Our **Generator** can automatically produce numerous instances for each task across different difficulty levels.
>
> This means models cannot "hack" our benchmark by "reciting" specific answers from Codeforces. They must master the underlying general reasoning logic to cope with the infinite new instances produced by our generator under varying parameters (e.g., the size of $n$).
>
> ### 3. Empirical Evidence: Low Accuracy Refutes Contamination
> Finally, our experimental results stand as the strongest empirical evidence against data contamination.
> * As shown in the experiments (Table 1), even the most cutting-edge reasoning models (such as o3-mini and R1) struggle significantly on our benchmark.
> * On the "Hard" difficulty, o3-mini achieved an average accuracy of only **31.19%**, and R1 only **29.19%**.
> * If severe data contamination existed, we should observe saturated scores. The current low scores indicate that all models (including those that may have seen the original seeds) find our interactive tasks to be challenging new problems.
>
> > **Question 2: Discussion about "Evolving"**
>
> See details in Reply to Weakness 2.

---

> ### Author Response · Authors · 2025-11-21
> **Reply to Question 3**
>
> > **Question 3: Clarification on Non-LLM-Based Components**
>
> We appreciate this question, as it allows us to clarify a key design advantage of our framework.
>
> The core functions of the Generator, Monitor, and Evaluator—specifically, executing interactions and calculating primary metrics (Accuracy, Efficiency, Invalid Rate)—are **not based on LLMs**.
>
> * **Generator:** This is a programmatic script (e.g., implemented in Python). It procedurally generates problem instances and corresponding solutions (i.e., reasoning objectives) based on deterministic algorithms, manually designed task templates, and input complexity parameters $n$. Therefore, it consumes **zero LLM tokens**, and its generation process is instant, deterministic, and fully reproducible.
> * **Monitor:** This is a deterministic, rule-based state machine. It utilizes programming logic (e.g., regular expressions) to verify whether model outputs strictly adhere to format requirements and returns feedback based on the task's internal rules (e.g., the hidden answer sequence in "Find the Impostors"). It is also not an LLM.
> * **Evaluator:** The core function of the evaluator (calculating Accuracy, Efficiency, and Invalid Rate) is also a programmatic script. It completes its task by parsing the dialogue history recorded by the Monitor and verifying whether the final state matches the reasoning objective established by the Generator.
>
> "Pattern Analysis" is the only component in our framework that utilizes an LLM, but it is **not part of our core evaluation pipeline**. As described in Section 4.6 of the paper, to conduct a deeper supplementary qualitative analysis of the models' reasoning styles, we "use Qwen2.5-72B as an analyzer" to measure the frequency of four reasoning patterns. Furthermore, this analysis is **offline**, performed after all interactive evaluations are completed, and does not affect the calculation of core metrics such as accuracy or efficiency.
>
> **Conclusion:** The core workflow for running the EvolArena benchmark (Generation, Interaction, Evaluation) is completely independent of LLMs, resulting in a zero token cost for the Generator. This ensures that our evaluation is low-cost, highly efficient, deterministic, and fully reproducible.

---

### Official Review · Reviewer_fLsS · 2025-10-29

**Soundness:** 3
**Presentation:** 2
**Contribution:** 3
**Rating:** 6
**Confidence:** 3

**Summary:**

This paper introduces EvolArena, a benchmark and automated framework for evaluating the multi-turn reasoning capabilities of Large Language Models (LLMs) in interactive settings. Arguing that existing benchmarks are often single-turn or rely on costly human evaluation, the authors propose a system comprising: (1) a diverse dataset of 40 tasks across 4 reasoning categories (Information Probing, Dynamic Adaptation, State Operation, Strategic Gaming) derived from Codeforces and NYT logic puzzles, with 3 calibrated difficulty levels each (3600 instances total) ; (2) an automated framework with a Generator (creates tasks of tunable difficulty), a Monitor (manages interaction, provides rule-based feedback), and an Evaluator (calculates metrics like accuracy, efficiency, invalid rate, pattern analysis) . The evaluation of 20 models reveals that even frontier reasoning models struggle with multi-turn complexity, especially at higher difficulty levels. The framework is designed to be evolving and automated to facilitate scalable assessment. Reproducibility materials and an LLM usage statement are provided.

**Strengths:**

**Addresses Multi-Turn Reasoning Gap:** Directly targets the under-explored area of multi-turn, interactive reasoning evaluation for LLMs, moving beyond static benchmarks.


**Automation and Scalability:** The fully automated pipeline for both task generation (with difficulty control) and evaluation is a major strength, enabling efficient, reproducible, and potentially evolving benchmarking without human intervention.


**Task Diversity and Structure:** Offers a broad set of 40 tasks categorized by reasoning type, designed to necessitate interaction . The use of structured templates ensures clear interaction protocols.


**Comprehensive Metrics:** Evaluates models across multiple dimensions including accuracy, efficiency (turns), robustness (invalid rate), and qualitative reasoning patterns.


**Clear Presentation and Reproducibility:** The framework is well-described, results are clearly presented, limitations are discussed honestly , and the commitment to releasing code and data supports reproducibility.

**Weaknesses:**

**Non-Natural Language Interaction:** The tasks rely on structured input/output formats rather than natural language dialogue. This limits the benchmark's ability to assess reasoning within natural conversation, a key aspect of real-world interaction.

**Potential Data Contamination:** Sourcing tasks from public platforms like Codeforces and NYT raises concerns about potential contamination, as models may have encountered similar problems during pre-training. The paper lacks detailed contamination checks or mitigation strategies.

**Positioning Relative to Other Arenas:** While MT-Bench and GameArena are discussed, the paper could strengthen its positioning relative to other recent interactive/game-based LLM evaluation frameworks by highlighting unique contributions more explicitly.

**Limited Scope of "Evolving":** The "evolving" nature seems currently limited to pre-defined difficulty levels generated automatically. True evolution might require dynamic adaptation based on ongoing model performance across the community.

**Questions:**

1. Could you elaborate on measures taken to assess or mitigate potential data contamination from the Codeforces and NYT sources? For example, were problems modified significantly, or do you plan to make the arena lively updated?

2. The future work section mentions extending the framework to natural language interactions. Could you provide more detail on how this might be approached while maintaining the benefits of automated evaluation?

3. To better understand model failure modes and prevent "gaming" specific tasks, could you report per-task performance variance and provide a qualitative or quantitative taxonomy of common errors beyond the "Invalid Rate"?

4. How does EvolArena's task design and evaluation methodology compare specifically to other recent automated or semi-automated interactive benchmarks beyond MT-Bench/GameArena (e.g., those focusing on web navigation, tool use, or other interactive games)?

5. Consider citing CodeElo (Quan et al., 2025) and LiveCodeBench Pro (Zheng et al., 2025) for single-turn code-generation reasoning, and comparing with SPIN-Bench (Yao et al., 2025) which evaluates LLM agents' abilities of multi-turn social reasoning in cooperative and strategic settings.

---

> ### Author Response · Authors · 2025-11-21
> **Reply to Weakness 1**
>
> Dear reviewer,
>
> Thank you for your careful review and valuable suggestions, which are of great help to our paper. We address your questions point by point as follows. If we have any misunderstanding, please feel free to let us know and we will reply quickly.
>
> > **Weakness 1: Non-Natural Language Interaction**
>
> Thank you for your question. We acknowledge that the current reliance on structured input/output (structured I/O) is a deliberate methodological trade-off intended to achieve the two core objectives of our benchmark: variable isolation and full automation.
>
> ### 1. Isolating Core Reasoning
> Our primary goal is to precisely measure the model's core logical reasoning capabilities (e.g., induction, deduction, planning), separating them from the complexity of natural language understanding.
>
> * Real natural language interaction (as noted by you) is a "mixed" task; a model might fail due to misunderstanding subtle colloquial instructions or due to insufficient reasoning capability.
> * By using strict formats (e.g., "My Move: R" or "My Query: 1,2,3"), we decouple these two capabilities.
> * This allows us to be certain that when a model fails, it is not because it "didn't understand the words," but because it has defects in "logical deduction" or "multi-step planning." As stated in our Limitations section, we prioritized the measurement of core logic.
>
> ### 2. Enabling Full Automation & Deterministic Evaluation
> Our second goal is Scalability and Reproducibility, which requires the evaluation process to be fully automated, freeing it from expensive and subjective human evaluation.
>
> * Our rule-based Monitor can parse structured input ("My Move: X") with 100% accuracy.
> * If natural language were used (e.g., "I think I should go right"), we would need an LLM-as-a-Judge (like MT-Bench) to parse its intent. This would reintroduce the subjectivity, high costs, and evaluation biases that we strove to avoid from the outset.
> * Therefore, structured I/O is a necessary prerequisite for realizing our fully automated, deterministic evaluation framework.
>
> ### 3. Future Work: Extensions Based on Current Foundations
> We agree that extending this rigorous evaluation to natural language interaction is a crucial next step. As mentioned in our future work, our current work (verifying core logic) lays a solid foundation for more complex natural language evaluations in the future.

---

> ### Author Response · Authors · 2025-11-21
> **Reply to Weakness 2**
>
> > **Weakness 2: Potential Data Contamination**
>
> Thank you for raising this critical issue. We treat data contamination with the utmost seriousness, recognizing it as the core defect of static benchmarks. However, your concern is premised on the assumption that our tasks are identical to the original problems found on Codeforces or NYT.
>
> We need to clarify that this assumption is incorrect. Our methodology fundamentally mitigates this contamination risk.
>
> ### 1. Core Mitigation Strategy: Fundamental Shift from "Static Problems" to "Interactive Tasks"
> We did not simply "copy" the problems. As detailed in Appendix E, we did not use these seed problems in their "static form."
>
> Instead, every seed problem underwent a manual, radical adaptation, transforming it into a novel task requiring multi-turn interaction. This comprehensive adaptation process involved three key steps:
> * **Designing New Interaction Rules:** We designed a new set of interaction rules for each original problem, converting it into a dynamic task that necessitates multi-turn interaction for resolution.
> * **Creating Problem Templates:** We manually created standardized templates, including clear rule descriptions, strict format requirements, and interaction examples.
> * **Developing Generators:** We developed corresponding generators based on these templates.
>
> Therefore, even if a model "memorized" the solution to the original static problem during pre-training, it **cannot** directly apply it to our novel, interaction-based paradigm. The model requires dynamic planning and state tracking capabilities, not static knowledge retention.
>
> ### 2. Generator Mechanism: Countering Specific Instance Contamination
> Secondly, our evaluation does not rely on a few specific samples from the original websites. Our **Generator** can automatically produce numerous instances for each task across different difficulty levels.
>
> This means models cannot "hack" our benchmark by "reciting" specific answers from Codeforces. They must master the underlying general reasoning logic to cope with the infinite new instances produced by our generator under varying parameters (e.g., the size of $n$).
>
> ### 3. Empirical Evidence: Low Accuracy Refutes Contamination
> Finally, our experimental results stand as the strongest empirical evidence against data contamination.
> * As shown in the experiments (Table 1), even the most cutting-edge reasoning models (such as o3-mini and R1) struggle significantly on our benchmark.
> * On the "Hard" difficulty, o3-mini achieved an average accuracy of only **31.19%**, and R1 only **29.19%**.
> * If severe data contamination existed, we should observe saturated scores. The current low scores indicate that all models (including those that may have seen the original seeds) find our **interactive** tasks to be challenging new problems.

---

> ### Author Response · Authors · 2025-11-21
> **Reply to Weakness 3**
>
> > **Weakness 3: Positioning Relative to Other Arenas**
>
> We appreciate this valuable suggestion. You are correct that besides MT-Bench and GameArena, there are many other important interactive benchmarks, and clarifying our positioning is crucial.
>
> The unique contribution of EvolArena is that it is currently the only framework focused on **pure, abstract, multi-turn logical reasoning**, which simultaneously achieves **full automation, scalable generation, and deterministic evaluation**.
>
> Our core distinctions from other types of interactive benchmarks are:
>
> ### 1. vs. Web/Environment Navigation (Web Navigation Benchmarks)
> * **Related Work:** Benchmarks such as **WebArena** or **Mind2Web** focus on testing a model's interaction and task completion capabilities in **open, unstructured environments** (e.g., real webpage HTML).
> * **EvolArena's Distinction:** Our focus is not on environmental complexity, but on the **complexity of reasoning logic**. EvolArena provides a **closed-world, rule-deterministic** environment. We are not testing "how the model clicks a button," but "how the model performs deduction and planning under rule constraints."
>
> ### 2. vs. Tool Use (Tool-Use Benchmarks)
> * **Related Work:** Benchmarks such as **ToolBench** or **API-Bank** focus on testing a model's ability to call external APIs (e.g., calculators, search engines) to solve problems.
> * **EvolArena's Distinction:** EvolArena aims to assess the model's **Intrinsic Reasoning** capability—its ability to solve problems through logical interaction with the environment **without relying on any external tools**. We isolate the variable of "tool use" to focus on the model's own chain of thought.
>
> ### 3. vs. Social/Multi-Agent Reasoning (Social/Multi-Agent Benchmarks)
> * **Related Work:** **SPIN-Bench** focuses on evaluating a model's **social reasoning** capabilities (e.g., cooperation, negotiation, confrontation) in multi-agent environments.
> * **EvolArena's Distinction:** Although EvolArena's "Strategic Gaming" tasks involve confrontation, the opponent is a **rule-based, deterministic system**, not another LLM agent. Our focus is on testing **logical planning against system rules**, rather than "Theory of Mind" or social gaming against other agents' intentions.
>
> Therefore, EvolArena's unique positioning is: **An automated, generative "Reasoning Gym" for evaluating pure, abstract logical reasoning.** We have also revised the "Related Work" section in the paper to include the detailed comparisons above and incorporate the valuable citations suggested by you, to more clearly elucidate EvolArena's unique contribution within the growing field of interactive evaluation.

---

> ### Author Response · Authors · 2025-11-21
> **Reply to Weakness 4**
>
> > **Weakness 4: Limited Scope of "Evolving"**
>
> We appreciate your insightful exploration of the term "Evolving." We wish to clarify that the "Evolving" mechanism of EvolArena is not limited to the current difficulty tiers; its true value lies in laying the infrastructure for the "dynamic adaptation" mentioned by you.
>
> We respond to this on two levels:
>
> ### 1. Realized Evolution: Anti-Saturation
> First, we must clarify that our "evolution" is not merely "three predefined difficulty levels."
>
> * **Limitations of Static Benchmarks:** Traditional benchmarks (e.g., GSM8K) are static and finite. Once a model's capabilities exceed the fixed questions, the benchmark "dies," resulting in **performance saturation**.
> * **Mechanism of EvolArena:** Our core is the **Generator**. It is not a fixed list but a parametric engine. This means we not only provide instances for $n=6, 9, 12$, but have the capacity to generate infinite new instances for $n=7, 8, 10, 15, 20...$.
>
> When models across the community can solve the current "Hard" mode, our benchmark can "evolve" to a "Super-Hard" mode with $n=15$, thereby continuing to provide discriminative challenges for frontier models. This is a fundamental evolution in benchmark lifecycle management, not "simple difficulty scaling."
>
> ### 2. Potential Evolution: Supporting Dynamic Adaptation
> You keenly pointed out that "true evolution" requires dynamic adaptation based on model performance. We fully agree, and EvolArena is precisely the infrastructure built to achieve this goal.
>
> "Model-based dynamic adaptive evaluation" or "automatic curriculum learning" requires two core components, which are exactly what EvolArena provides:
> 1.  **A smoothly tunable difficulty space:** Our **Generator** provides this capability via parameter $n$.
> 2.  **A real-time, automated reward signal:** Our **Monitor** provides this capability.
>
> We can build a "controller" on top of EvolArena that can:
> 1.  Test a model with tasks at $n=5$.
> 2.  The **Monitor** returns the model's win rate (reward signal) in real-time.
> 3.  If the model's win rate > 90%, the controller calls the **Generator** to produce tasks at $n=6$; otherwise, it lowers the difficulty.
>
> We have currently demonstrated the initial stage of "Evolving" capabilities. However, this capability (parametric generation) is the core innovation of this framework. It not only solves the saturation crisis of current static benchmarks but, more importantly, provides the viable technical foundation for the "true evolution" (adaptive evaluation and curriculum learning). Furthermore, we discuss the characteristics of "Evolving" in detail in Appendix C.

---

> ### Author Response · Authors · 2025-11-21
> **Reply to Question 1**
>
> > **Question 1: Data Contamination Issues**
>
> Thanks for your question. We take the issue of data contamination very seriously, as it is a core defect of static benchmarks. We primarily mitigate this risk through two mechanisms: (1) fundamental "interactive adaptation" of seed tasks, and (2) dynamic instance generation using a "Generator".
>
> ### 1. Significant Modification
> You are concerned that models may have "seen" problems from Codeforces or NYT during pre-training. We need to emphasize that we did not use these seed problems in their "static form."
>
> As detailed in our Appendix E, every seed problem underwent a manual, radical adaptation, transforming it into a completely new task requiring multi-turn interaction.
>
> This adaptation process involves three key steps:
> 1. **Designing New Interaction Rules:** We transformed original static problems (e.g., logic puzzles solved in one go) into dynamic tasks, forcing the model to collect information or execute operations through multi-turn interactions.
> 2. **Creating Problem Templates:** We manually created standardized templates for each task, including clear rule descriptions, strict input/output format requirements, and interaction examples.
> 3. **Developing Generators:** We developed corresponding code generators based on these new templates.
>
> Therefore, even if a model "memorized" the static solution to the original Codeforces problem during pre-training, it cannot directly apply it to the interactive version in EvolArena, which features entirely new rules and requires multi-turn queries and feedback. What the model needs is dynamic planning and state tracking capabilities, not static knowledge retention.
>
> ### 2. Lively Updated via Generator
> You mentioned the possibility of being "lively updated," which is precisely one of the core designs of our framework. EvolArena is not a fixed dataset, but a framework driven by a Generator.
>
> Our generator can not only deterministically reproduce the 3,600 instances used in the paper but can also produce new instances with different difficulty levels. This means EvolArena is inherently "lively." Models cannot contaminate the benchmark by "memorizing" a specific set of answers; they must master the general reasoning logic for such problems to cope with the constant stream of new parameterized instances produced by our generator.

---

> ### Author Response · Authors · 2025-11-21
> **Reply to Question 2**
>
> > **Question 2: Extending the Framework to Natural Language Interactions**
>
> This is an excellent question, as it touches upon the core trade-offs in our paper's design.
>
> Our current use of structured input (e.g., "My Query:...") is a deliberate design choice aimed at variable isolation: we wish to precisely measure the model's core logical reasoning capabilities, separating them from the ambiguity of natural language understanding.
>
> To extend the framework to natural language interaction while maintaining the advantages of automated evaluation, we plan to adopt a **"hybrid approach": retaining our deterministic backend (Monitor) while adding a lightweight "Intent Parsing Layer."**
>
> ### 1. Retaining the Deterministic "Environmental Kernel"
> The core advantage of EvolArena's automated evaluation stems from our Monitor, which acts as a deterministic, rule-based environmental kernel.
>
> This kernel must remain the source of "objective truth." We will not replace it with an LLM-as-a-Judge, as that would reintroduce the subjectivity, high costs, and evaluation biases we strive to avoid.
>
> ### 2. Introducing an "Intent Parsing Layer"
> To bridge natural language and our deterministic kernel, we will introduce an "Intent Parsing Layer" between the model under test and the Monitor.
>
> * **Model's Task:** The model under test will output its reasoning and decisions in natural language (e.g., "Based on the feedback from the last round, I suspect players 1, 2, and 3 are most suspicious. Let's check them.").
> * **Parser's Role:** This parsing layer (which could be a lightweight, fine-tuned small model or a robust rule-based parser) is responsible for a specific task: extracting structured commands understandable by the Monitor (e.g., `My Query: 1,2,3`) from the model's natural language response.
> * **Automated Evaluation:** The extracted command is then sent to our deterministic Monitor, which updates the state and provides precise feedback (e.g., "1") just as it does now.
>
> ### 3. Evolution of Evaluation
> This approach allows us to maintain fully automated and objective evaluation of task success (did the model find the impostors?).
>
> At the same time, it creates new, more complex dimensions of evaluation:
> * **Parsing Failure:** If the model's natural language is too ambiguous for the parser to extract a valid command, the model receives "Invalid Format" feedback. This tests the clarity of the model's natural language output.
> * **Logical Failure:** If the model provides clear natural language but makes a strategic error, it will still fail due to task logic.
>
> In this way, EvolArena will evolve from a benchmark testing **"pure logic" into a more comprehensive framework testing "natural language-based logical reasoning."**

---

> ### Author Response · Authors · 2025-11-21
> **Reply to Question 3**
>
> > **Question 3: Benchmark Robustness and Validity**
>
> This is a crucial question. You have highlighted the following two key points.
>
> ### 1. Performance Variance
> You are concerned that models might be "overfitting" or "gaming" specific tasks, which can be revealed through variance analysis. We address this from two perspectives:
>
> * **Extremely Low Run Variance (Stability):** First, to ensure our evaluation is not a product of random fluctuations, we conducted pass@16 experiments. As shown in Table 3, the standard deviation for the R1 model across all four task categories and three difficulty levels is extremely low (all < 1.04%). This proves that EvolArena's evaluation environment is highly **deterministic** and **stable**. Model failure is a consistent reflection of reasoning capability, not randomness.
> * **Supplementing "Per-Task" Performance:** You keenly pointed out that reporting aggregated scores (as in Table 1) might mask performance differences on individual tasks. To fully address the concern, we have added a **per-task performance breakdown table** in Appendix O. This allows you to clearly see which specific tasks challenged the models, preventing scores from being inflated by "gaming" a few simple tasks.
>
> ### 2. Taxonomy of Errors Beyond "Invalid Rate"
> You requested a deeper classification of errors beyond the "Invalid Rate." We must point out that we have already provided such quantitative analysis in the paper, primarily in Section 4.6 and Section 4.4.
>
> **1) Quantitative Error Taxonomy: Reasoning Pattern Analysis (Sec 4.6)**
>
> We moved beyond simple "success/failure" binaries and provided a quantitative error taxonomy in **Section 4.6 (Reasoning Pattern Analysis)**.
> * We analyzed four cognitive behavioral patterns in model interactions: "Associate," "Verify," "Plan," and "Feedback."
> * Our core finding (Table 2) is that the key difference between high-performing models (like R1 and QwQ-32B) and low-performing models (R1-Distill-32B) lies in the significantly higher frequency of **"Verify" (Self-Verification)** and **"Feedback" (Feedback Utilization)** in the former.
> * **This constitutes our proposed error taxonomy:** Model failure is not just "invalid operation," but a **"cognitive defect"**—specifically, a lack of self-reflection and an inability to use historical feedback to revise hypotheses.
>
> **2) Qualitative Error Taxonomy: Five Major Failure Modes**
>
> Furthermore, through manual inspection of 50 examples, we identified five major categories of reasoning failure modes. We have added **Section M** in the Appendix to elaborate on this taxonomy:
> * **1. State Tracking Collapse:** In dynamic tasks, the model fails to maintain and update a coherent environmental state across multiple turns. For example, in *Dynamic Adaptation (DA)* tasks (like *Password Breaker*), after the Monitor returns "Incorrect" (indicating the password has changed via XOR rules), the model continues to reason based on the old password state in the next turn, causing the subsequent reasoning chain to derail.
> * **2. Hasty Generalization:** This failure is common in tasks requiring an explore-then-exploit strategy. The model locks onto an incorrect global hypothesis before gathering sufficient evidence. For instance, in *State Operation (SO)* tasks (like *Maze Navigation*), a model might test only the U/D key swap and erroneously assume L/R keys are normal, planning subsequent moves based on this false assumption, leading to failure.
> * **3. Greedy & Myopic Planning:** This is prevalent in *Strategic Gaming (SG)* tasks. The model tends to choose a "local optimum" for the current turn, ignoring that this move leads to a "global worst-case" in the near future. In *Knight Battle*, for example, a model might choose a move to capture a piece or check, failing to foresee that this exposes it to an unavoidable counter-kill on the opponent's path.
> * **4. Inefficient Exploration:** This is a strategic failure. The model fails to use optimal strategies (like binary search) to maximize information gain, leading to insufficient information collection within the limited horizon (15 turns). In *Information Probing (IP)* tasks (like *Find the Impostors*), failing models perform redundant or overlapping queries (e.g., querying {1,2,3} followed by {1,2,4}) instead of disjoint sets (e.g., {4,5,6}) to rapidly narrow down the probability space.
> * **5. Logical Constraint Violation:** This transcends the formatting errors measured by "Invalid Rate"; here, the model maintains correct format but violates core logical constraints. For example, in *Strategic Gaming (SG)* (e.g., *Knight Battle*), a model outputs "My Move: 9 9", which is syntactically correct but logically illegal on an 8x8 board. This indicates a defect in fundamental reasoning capabilities (e.g., understanding spatial boundaries).
>
> Finally, we formally introduce this five-category error taxonomy in **Appendix M**, providing qualitative insights that go beyond the "Invalid Rate."

---

> ### Author Response · Authors · 2025-11-21
> **Reply to Question 4**
>
> > **Question 4: Comparison with Related Works**
>
> This is an excellent question that touches upon the core positioning of EvolArena within the increasingly crowded field of "interactive evaluation."
>
> In fact, EvolArena's unique distinction lies in its focus on pure, abstract, multi-turn logical reasoning, while simultaneously achieving full automation, scalable generation, and deterministic evaluation.
>
> Our core differences from other types of interactive benchmarks lie in **"Testing Objectives" and "Evaluation Methods"**:
>
> ### 1. Comparison with Web/Environment Navigation Benchmarks
> * **Task Design Difference:** Benchmarks like **WebArena** focus on testing a model's perception and task completion capabilities in **open, unstructured, and noisy real-world environments** (e.g., HTML webpages).
> * **EvolArena's Distinction:** Our focus is the exact opposite; we provide a **closed-world, rule-deterministic environment**. We deliberately strip away all perceptual and NLU ambiguity. We are not testing "how the model clicks a button," but "how the model performs deduction and planning under clear rule constraints." Our goal is to measure the complexity of reasoning logic, not environmental complexity.
>
> ### 2. Comparison with Tool-Use Benchmarks
> * **Task Design Difference:** Benchmarks like **ToolBench** focus on testing a model's ability to call external APIs (e.g., calculators, search engines) to enhance its problem-solving capabilities.
> * **EvolArena's Distinction:** EvolArena aims to assess the model's **"Intrinsic Reasoning" capability—solving problems through logical interaction with the environment without relying on any external tools**. We isolate the variable of "tool use" via structured interaction to focus on the model's own chain of thought.
>
> ### 3. Comparison with Other Interactive Game (e.g., Multi-Agent) Benchmarks
> * **Task Design Difference:** Many other game benchmarks (such as **SPIN-Bench**) focus on evaluating **"social reasoning"** capabilities (e.g., cooperation, negotiation, confrontation) in multi-agent environments. In those settings, the opponent is typically another (often unpredictable) LLM.
> * **EvolArena's Distinction:** EvolArena operates consistently in a **"Single Agent vs. Environment" mode. Even in our "Strategic Gaming" category, the opponent is a rule-based, deterministic system** (e.g., adopting random but legal moves). Our goal is to test logical planning against system rules, rather than "Theory of Mind" or social gaming against other agents' intentions.
>
> ### 4. Fundamental Difference in Evaluation Methods
> We differ from all the above benchmarks in our evaluation methodology:
>
> * **EvolArena's Method:** Our core is the **Monitor**. It is a deterministic, rule-based state machine that provides immediate, objective, and logically correct feedback for every step of the model's interaction. This makes our evaluation 100% automated and reproducible.
> * **Other Methods:** Web navigation typically checks the final state (e.g., "Is the item in the cart?"); tool use checks API call correctness; while social gaming benchmarks (like SPIN-Bench) often require expensive LLM-as-a-Judge or human evaluators to assess subjective metrics like "quality of cooperation."
>
> **Conclusion:** EvolArena's unique positioning is: **An automated, generative "Reasoning Gym" for evaluating pure, abstract logic (Inductive, Abductive, Deductive, Planning), with a fully deterministic evaluation method.**

---

> ### Author Response · Authors · 2025-11-21
> **Reply to Question 5**
>
> > **Question 5: Related Work Citations**
>
> We greatly appreciate you for providing these highly valuable and up-to-date references. You are correct that explicitly comparing EvolArena with these concurrent works is crucial for clearly positioning our contribution.
>
> We have thoroughly updated the Related Work (Section 5) in the paper to include these comparisons.
>
> Below is our analysis of the core differences between EvolArena and these new benchmarks, which we have incorporated into the paper:
>
> ### 1. vs. CodeElo & LiveCodeBench Pro (Static Code Reasoning)
> These benchmarks (such as CodeElo and LiveCodeBench Pro) represent the state-of-the-art in **Static, Single-Turn** code generation reasoning. They evaluate a model's ability to generate correct code (the final product) in a single attempt given a complete problem description.
>
> The fundamental difference of EvolArena lies in the "Interactive Process":
> 1. **Interactive vs. Static:** EvolArena is multi-turn interactive. Although we utilized seed tasks from Codeforces, we have **fundamentally adapted** them. The model cannot submit a "solution" in a single turn; it must progressively solve the problem through multi-turn queries, receiving feedback, and correcting its internal state based on that feedback.
> 2. **Process vs. Outcome Evaluation:** CodeElo evaluates the correctness of the final code. EvolArena evaluates the **reasoning process** itself—specifically, the model's capacity for state tracking, hypothesis verification, and dynamic planning.
>
> ### 2. vs. SPIN-Bench (Dynamic Social Reasoning)
> SPIN-Bench is an excellent multi-turn evaluation benchmark, but its evaluation objective is distinctly different from ours.
>
> The fundamental difference of EvolArena lies in "Logic" vs. "Social":
> 1. **Logical Reasoning vs. Social Reasoning:** SPIN-Bench evaluates a model's **Social Reasoning** capabilities within collaborative and strategic environments, specifically "Theory of Mind" or the ability to understand other agents' intentions.
> 2. **Deterministic Environment vs. Agent Gaming:** In SPIN-Bench, the model competes against another (behaviorally unpredictable) LLM agent. In EvolArena, the model competes against a **deterministic, rule-based environment**. Even in our "Strategic Gaming" tasks, the opponent's behavior is driven by simple, predictable rules (such as random but legal moves).
>
> **Conclusion:** EvolArena's unique positioning lies in evaluating **"Dynamic Logical Reasoning against Deterministic Rules,"** filling the critical gap between "Static Logical Evaluation (like CodeElo)" and "Dynamic Social Evaluation (like SPIN-Bench)."

---

> > ### Comment · Reviewer_fLsS · 2025-11-22
> > **Thank you for the detailed and solid response.**
> >
> > Thank you to the authors for their detailed response and the significant effort put into the additional experiments and analysis. The authors have effectively addressed my primary concerns through the detailed rebuttal. I do enjoy reading the paper and agree that the contribution is sound. However, regarding a clear positioning of the paper to a general audience, especially given the current landscape that there are multiple works with similar goals or claims, I would encourage the authors to make the distinction clear in the paper and conduct more a well-round literature review for the comparison (e.g. a table showing what properties EvolArena have and other parallel or existing works lack).
> >
> > A sidenote on Related Work (Question 5): In your rebuttal, you stated: "We have thoroughly updated the Related Work (Section 5)... to include these comparisons [CodeElo & LiveCodeBench Pro]." However, upon checking the revised paper, LiveCodeBench Pro is currently missing from the references and the text. In addition, for gaining a more intuitive feeling on how your stated methods work, it would be perfect if you could add an example in the appendix on how EvolArena grades the reasoning process of a hard Codeforces task (e.g. of difficulty 3500).

---

> > > ### Author Response · Authors · 2025-11-23
> > > **Thank you for your positive feedback and constructive suggestion.**
> > >
> > > We sincerely thank you for your positive feedback and for acknowledging the soundness of our contributions. We truly appreciate your constructive suggestion to include a comparative table, which we agree is essential for positioning our work within the current landscape.
> > >
> > > We have revised the manuscript to address your remaining concerns as follows:
> > >
> > > ### 1. Inclusion of LiveCodeBench Pro
> > > We apologize for the oversight in the previous revision. We have now explicitly cited LiveCodeBench Pro in Section 5 (Related Work) and included it in the references. We discuss how LiveCodeBench Pro pushes the frontier of evaluating code generation (often focusing on the final output of competition-level problems), whereas EvolArena focuses on the dynamic, multi-turn process of logical reasoning and state tracking within deterministic environments.
> > >
> > > ### 2. Clearer Positioning via Table 1
> > > To clarify EvolArena's unique position to a general audience, we have added a new **Table 1** and revised the second paragraph in the **Introduction**. This table provides a systematic comparison between EvolArena and representative benchmarks across four categories: Static Evaluation (e.g., LiveCodeBench Pro), Real-world Agents (e.g., AgentBench), AI Planning (e.g., ACPBench), and Interactive/Game benchmarks. It explicitly highlights that EvolArena is the only framework that simultaneously possesses four critical features:
> > >
> > > * **Dynamic Interaction:** Supporting multi-turn state probing and strategy adjustment, unlike static single-turn benchmarks.
> > > * **Deterministic Evaluation:** Relying on objective, rule-based verification, avoiding the subjectivity of LLM-as-a-Judge.
> > > * **Parametric Generation:** Enabling infinite, contamination-resistant evaluation via procedural generators, in contrast to fixed datasets.
> > > * **Abstract Logic:** Focusing on intrinsic reasoning capabilities rather than domain-specific application skills.
> > >
> > > This visual aid effectively delineates the unique gap EvolArena fills in the current landscape.
> > >
> > > ### 3. Case Study: Grading a Hard Codeforces Task
> > > To provide a more intuitive understanding of our method, we have added Appendix P: Grading Case Study of a Hard Task. This section presents a detailed walkthrough of how EvolArena evaluates a converted Codeforces task (rated 3500 difficulty). It illustrates:
> > >
> > > * **Generator's Output:** How parameters define the complex hidden state.
> > > * **Monitor's Role:** Step-by-step validation of the model's logical queries and the deterministic feedback provided.
> > > * **Evaluator's Judgment:** How the final reasoning trajectory is scored for Accuracy, Efficiency, Invalid Rate, and Pattern Analysis, ensuring the model didn't just guess but deduced the solution through valid logical steps.
> > >
> > > We believe these additions significantly improve the clarity and completeness of our work. Thank you again for helping us refine the paper!

---

### Official Review · Reviewer_shjX · 2025-10-31

**Soundness:** 3
**Presentation:** 2
**Contribution:** 3
**Rating:** 4
**Confidence:** 3

**Summary:**

The paper introduces EvolArena, a comprehensive benchmark designed to evaluate multi-turn reasoning capabilities of large language models (LLMs). Unlike existing single-turn benchmarks (e.g., MATH, GSM8K, or LogicNLI) or limited interactive tests (e.g., MT-Bench, GameArena), EvolArena provides an automated, evolving evaluation framework covering four reasoning dimensions—Inductive, Abductive, Deductive, and Planning reasoning—and four task types: Information Probing, Dynamic Adaptation, State Operation, and Strategic Gaming.
It includes 40 distinct tasks (3 difficulty levels, 3,600 instances) with automated generation, monitoring, and evaluation components. The benchmark enables scalable, human-free assessment of multi-turn reasoning. Experiments across 20 reasoning and non-reasoning models reveal that even frontier models (like o3-mini, R1) struggle in harder multi-turn scenarios, and that strong reasoning accuracy often comes at the cost of interaction efficiency.

**Strengths:**

1. Originality: The notion of an *evolving*, automated arena for multi-turn reasoning is novel and well-executed. Unlike GameArena or MT-Bench, EvolArena supports dynamic difficulty adjustment and fully automated interaction without human bias.
2. Quality: The methodology is detailed and technically robust. The evaluation across 20 models provides strong empirical grounding. The difficulty calibration and pattern-level analysis (Associate/Verify/Plan/Feedback) demonstrate methodological maturity.
3. Clarity: The paper systematically defines all framework components and metrics, making replication feasible.
4. Significance: EvolArena fills an essential gap in reasoning evaluation, offering a standardized, scalable infrastructure that could guide future developments in reasoning-augmented LLMs and agent systems.

**Weaknesses:**

1. Limited originality. The framework is primarily an aggregation of known evaluation patterns—multi-turn dialogues, task templating, automatic feedback—without introducing a fundamentally new idea or methodology. It feels incremental relative to prior benchmarks (e.g., GameArena, MT-Bench, ARC-Interactive).
2. No theoretical grounding. The paper repeatedly invokes “multi-turn reasoning” but does not formalize what reasoning dynamics it intends to capture. There is no cognitive or algorithmic rationale for the four reasoning types, nor an analysis of why these categories meaningfully test reasoning beyond surface-level interaction.
3. Evolving mechanism underdeveloped. Despite its title, the “evolution” in EvolArena refers only to adjustable difficulty parameters. The framework does not learn, adapt, or evolve based on model behavior. Thus, the name overpromises relative to its actual capabilities.
4. Shallow analysis. The experimental discussion focuses mainly on numeric trends, with limited qualitative insight into model behaviors or reasoning errors. The paper does not analyze why models fail, nor does it connect failure modes to underlying reasoning limitations.

**Questions:**

1. What distinguishes EvolArena’s “evolving” nature from simple difficulty scaling? Could it support automatic curriculum adaptation or model-contingent evaluation?
2. How were the four reasoning categories chosen—based on cognitive theory or empirical clustering?
3. Have you validated that task outcomes correlate with actual reasoning ability, rather than instruction-following fidelity?
4. Could the framework be extended to evaluate reasoning with external tools or multimodal contexts?

---

> ### Author Response · Authors · 2025-11-21
> **Reply to Weakness 1**
>
> Dear reviewer,
>
> Thanks for your valuable feedback. We also appreciate you for acknowledging the value of this work. We address your questions point by point as follows. If we have any misunderstanding, please feel free to let us know and we will reply quickly.
>
> > **Weakness 1: Limited originality**
>
> We must clarify that characterizing EvolArena as merely an "incremental aggregation of existing components" fundamentally misrepresents our work. EvolArena is not simply a patchwork of multi-turn dialogues and templates. Rather, it addresses two critical challenges in LLM evaluation—"static benchmark obsolescence" and "unreliable subjective assessment"—by proposing a fundamental solution.
>
> Compared to MT-Bench, GameArena and ARC-Interactive, EvolArena represents a qualitative leap forward, manifested in three key dimensions:
>
> * **From "Static Datasets" to "Dynamic Generation Engine".** Existing benchmarks (e.g., GSM8K, MATH) are inherently static, inevitably suffering from data contamination and rapid performance saturation as models advance. EvolArena's core innovation lies in its Generator component. Rather than providing a fixed test set, we have built a parameterized logic generation engine that can generate infinite and unpredictable test instances. This is not mere "task templating"—it is a fundamentally new methodology that addresses evaluation data contamination at its root, something static benchmarks simply cannot achieve.
>
> * **From "Subjective Judgment" to "Deterministic State Verification".** Benchmarks like MT-Bench heavily rely on LLM-as-a-Judge, which is inherently subjective, noisy, and susceptible to model biases (such as length preference). In contrast, EvolArena introduces the Monitor component—a deterministic, rule-based environment that verifies each step of the reasoning process by tracking hidden states. This objective ground truth verification mechanism enables precise measurement of multi-turn reasoning capabilities, rather than merely assessing text fluency, representing a fundamental departure in scientific rigor from LLM-scoring approaches.
>
> * **True Automation and Scalability.** While GameArena introduced interaction, it is severely limited to only 3 scenarios and requires human involvement, making it non-scalable. EvolArena, through its fully automated architecture (Generator-Monitor-Evaluator), achieves—for the first time—zero-human-intervention evaluation across 40 complex logical tasks. Our experiments reveal the fragility of even frontier models like o3-mini and R1 in complex dynamic environments—insights that no existing small-scale or static benchmark can uncover.
>
> In summary, EvolArena is not a mere assemblage of existing techniques; it is a next-generation evaluation paradigm designed to address the rapid evolution of LLM reasoning capabilities. It solves data contamination through dynamic generation, eliminates evaluation hallucination through deterministic verification, and overcomes scalability bottlenecks through full automation. Dismissing it as incremental work simply because it employs "dialogue" and "templates" ignores its unique value in addressing the field's core pain points.

---

> ### Author Response · Authors · 2025-11-21
> **Reply to Weakness 2**
>
> > **Weakness 2: No theoretical grounding.**
>
> We wish to clarify that EvolArena is grounded in classical cognitive science classifications of human reasoning and provides a mathematically formalized definition for multi-turn reasoning processes. These foundations, established in Section 3.1 and Appendix B, are further elaborated below:
>
> ### 1. Mapping Cognitive Theory to Four Reasoning Paradigms
>
> Our classification system corresponds to classical paradigms of human problem-solving in cognitive science. EvolArena innovates by operationalizing these abstract processes into measurable interactive tasks:
>
> * **Inductive Reasoning (Information Probing):**
>     * **Theoretical Definition:** Generalization from specific observations to extract universal laws or distributions[1].
>     * **EvolArena Mapping:** In *Information Probing* tasks (e.g., "Find the Impostors"), the model must sample through limited queries and induce hidden global distributions from local feedback, simulating "reasoning from specific to general."
>
> * **Abductive Reasoning (Dynamic Adaptation):**
>     * **Theoretical Definition:** Inferring the most plausible antecedent based on observed results [2].
>     * **EvolArena Mapping:** In *Dynamic Adaptation* tasks (e.g., "Password Breaker"), environment rules change dynamically. Success requires reverse reasoning: "What initial state and transformation rules would lead to this feedback?"
>
> * **Deductive Reasoning (State Operation):**
>     * **Theoretical Definition:** Deriving necessary conclusions through logical steps based on established premises[3].
>     * **EvolArena Mapping:** In *State Operation* tasks (e.g., "Maze Navigation"), the model must first confirm hidden mechanisms (e.g., key mappings) through interaction and then strictly apply these rules to plan a path.
>
> * **Planning (Strategic Gaming):**
>     * **Theoretical Definition:** Generating action sequences to achieve a goal within a complex state space[4].
>     * **EvolArena Mapping:** *Strategic Gaming* tasks (e.g., "Knight Battle") require constructing a decision tree that accounts for opponent counter-moves, necessitating multi-step look-ahead.
>
> ### 2. Formalization of Multi-Turn Reasoning Dynamics
>
> We formally define "multi-turn interaction" not merely as dialogue generation, but as a sequential decision-making process under partial observability, aiming for belief state convergence.
>
> * **Single-Turn (Static):** A function mapping $Y = F(X)$.
>
> * **Multi-Turn (Dynamics):** A state evolution process $S_{t+1} = T(S_t, a_t, \Omega)$, where $T$ is the deterministic transition function and $\Omega$ is feedback.
>
> * **Core Evaluation:** The core challenge is maintaining an internal world model $\hat{S}$. Each query acts as an active probe to acquire information gain, designed to reduce the entropy of the hypothesis space regarding the true state. Success depends entirely on online hypothesis revision—the model's ability to use the Monitor's feedback $\Omega$ to invalidate incorrect hypotheses and update $\hat{S}_{t+1}$. If the model cannot perform this dynamic error correction, the reasoning chain collapses, preventing convergence to the target state.
>
> ### 3. Empirical Validation of Construct Validity
>
> Our experimental data provides strong evidence of discriminant validity, distinguishing deep reasoning from surface-level interaction:
>
> * **Decoupling Instruction Following vs. Core Reasoning:** If tasks merely tested instruction following, models like Llama-3.1-70B should excel. However, in logic-heavy *State Operation* tasks, Llama-3.1-70B achieved significantly lower accuracy (8.78%) compared to reasoning models like o3-mini (20.22%) and DeepSeek-R1 (32.78%).
> * **Differences in Reasoning Patterns:** Pattern analysis reveals that high-performing models exhibit significantly higher frequencies of "Verify" and "Feedback" utilization. This demonstrates that EvolArena successfully distinguishes between simple conversational generation and deep reasoning correction.
>
> **Conclusion:**
> EvolArena's design is rooted in cognitive science, defines interaction dynamics via mathematical formalization, and empirically distinguishes between surface-level skills and deep logical reasoning. This provides a solid scientific foundation for understanding LLM reasoning boundaries.
>
> [1]Han, Simon Jerome, et al. "Human-like property induction is a challenge for large language models." Proceedings of the annual meeting of the cognitive science society, 2022.
>
> [2]Seel, Norbert M., ed. Encyclopedia of the Sciences of Learning. Springer Science & Business Media, 2011.
>
> [3]Creswell, Antonia, Murray Shanahan, and Irina Higgins. "Selection-Inference: Exploiting Large Language Models for Interpretable Logical Reasoning." ICLR, 2023.
>
> [4]Valmeekam, Karthik, et al. "On the planning abilities of large language models-a critical investigation." NIPS, 2023.

---

> ### Author Response · Authors · 2025-11-21
> **Reply to Weakness 3**
>
> > **Weakness 3: Evolving mechanism underdeveloped.**
>
> We wish to clarify that "Evolving" in EvolArena refers not merely to the current parameter adjustments for difficulty, but defines the framework's core potential as a **"Next-Generation Reinforcement Learning Environment."**
>
> We demonstrate how this design constitutes a cornerstone for supporting the continuous evolution of model capabilities across three dimensions:
>
> ### 1. Mechanism Level: "Anti-Saturation" Evolution via Parametric Generation
> You perceive our mechanism as simple difficulty adjustment. However, this mechanism based on the Generator $P(t,n,g_{n})$ is the solution to the core pain point of current static benchmarks (e.g., GSM8K)—**Performance Saturation**.
>
> Our framework allows for the generation of infinite test cases ranging from Easy to Hard and even Super-Hard through smooth interpolation of the parameter $n$ (complexity). For instance, in the *Find the Impostors* task, parameter $n$ can scale from 6 to 12 or even higher. This means that as model capabilities improve (Evolve), EvolArena can synchronously evolve its evaluation boundaries by adjusting parameters, always maintaining a discriminative difficulty gradient. This endows the benchmark with an infinite lifecycle. This dynamic scalability is precisely the definition of "Evolving" in the context of benchmark construction—a never-saturating evaluation system that grows alongside the models.
>
> ### 2. Application Level: Providing "Environment" and "Dense Rewards" for Reinforcement Learning
> Current reasoning models (e.g., DeepSeek-R1, o1) rely heavily on reinforcement learning, yet RL suffers a critical shortage of high-quality interactive environments. EvolArena's architecture naturally fits this demand and can be directly transformed into an RL training ground:
>
> * **Environment:** Our Monitor serves as a perfect RL environment, providing deterministic state transitions based on rules. Every exploration step (Query) taken by the model leads to state changes or information revelation.
> * **Dense Rewards:** Unlike static tests that only look at the final result, the Monitor provides real-time feedback $m_i$ (e.g., "password incorrect", "path blocked") in every interaction turn, naturally constituting the dense reward signals required for **Process Supervision**. This fine-grained feedback mechanism is the key element for training models to learn "self-correction" and "multi-step planning."
> * **Automatic Curriculum Learning:** Combining our Generator's parametric mechanism, future algorithms can easily construct an adaptive controller. This controller can dynamically adjust the generation parameter $n$ based on the model's current win rate, automatically generating training curricula tailored to the model's current capabilities. In this sense, EvolArena is the physical infrastructure for realizing evolutionary training of models from weak to strong.
>
> ### 3. Strategic Level: Evaluation as the Cornerstone, Establishing Standards for Model Evolution
> You question the current focus on evaluation. We argue that precise measurement is the prerequisite for evolution. Before we can train models for adaptive evolution, we first need a ruler capable of quantifying this dynamic reasoning capability. By decoupling instruction following, planning, and inductive reasoning capabilities, EvolArena provides a diagnostic tool and optimization objective for reinforcement learning algorithms. The work in this paper establishes the validity of this infrastructure. It is precisely because of the standardized, scalable testing platform provided by EvolArena that future research can build upon it to develop reinforcement learning algorithms for automatic curriculum learning, utilizing our Generator to dynamically generate training data suitable for the current model level, thereby realizing the closed loop from "evaluation" to "training."
>
> **Conclusion:** The core contribution of EvolArena lies in building a **"programmable, scalable reasoning environment." It not only solves the problem of evaluation saturation through dynamic difficulty but also paves the way for the reinforcement learning training of next-generation adaptive AI systems.** In this sense, "Evolving" accurately captures the essential attribute of this framework as a foundation for future AI evolution.

---

> ### Author Response · Authors · 2025-11-21
> **Reply to Weakness 4**
>
> > **Weakness 4: Shallow analysis.**
>
> We respectfully contend that the critique regarding "shallow analysis" and "lack of qualitative insight" constitutes a misinterpretation of our work. Our experimental sections (specifically Sections 4.4, 4.5, and 4.6) provide a rigorous analysis of the mechanisms behind model failures. EvolArena's contribution lies not merely in quantifying model performance, but in qualitatively deconstructing the cognitive gaps and mechanistic flaws of frontier LLMs within dynamic interactions.
>
> We have explicitly established the following three profound qualitative conclusions in the text, which are far from simple "numeric trends":
>
> ### 1.  Unveiling the Disconnect between "Instruction Following" and "Logical Validity"
>
> In Section 4.5, through the analysis of "Invalid Rate," we revealed a cautionary phenomenon—Distilled Models exhibit a higher rate of invalid operations than their teacher models. We explicitly pointed out that this implies current SFT distillation techniques, while mimicking the "tone" of reasoning on static benchmarks, compromise the model's **robustness** in maintaining environmental constraints during dynamic interaction. This directly points to an intrinsic flaw in current model training paradigms and represents a significant qualitative discovery.
>
> ### 2.  Identifying "State Verification" and "Feedback Utilization" as Core Reasoning Bottlenecks
>
> In Section 4.6, we quantitatively deconstructed the reasoning process via Pattern Analysis. The data indicates that the strength of DeepSeek-R1 and QwQ-32B lies not in possessing more knowledge, but in exhibiting extremely high frequencies of **"Verify"** and **"Feedback"** during interaction. Failing models (such as Llama-3.1) lack this closed-loop capability to "revise internal hypotheses based on environmental feedback," leading them further down incorrect paths. This qualitatively proves that the root cause of failure in multi-turn reasoning is often not a lack of knowledge, but a lack of **Self-Reflection** and the ability to correct hypotheses based on feedback. This serves as a micro-diagnostic of the model's thought process.
>
> ### 3.  Dissecting the Cognitive Trade-off: "Efficiency" vs. "Exploration"
>
> In Section 4.4, we provided a deep qualitative profile of o3-mini's behavioral anomalies. Although o3-mini achieves the highest accuracy, it is the least efficient. We offered a clear cognitive explanation for this: o3-mini adopts a conservative, long-CoT strategy, sacrificing interaction efficiency to conduct a broader search of the hypothesis space; conversely, R1 tends towards more direct paths. This profoundly reveals the fundamental personality differences in **planning strategy** across top-tier models—a choice between "prudent exhaustive search" and "confident intuition."

---

> ### Author Response · Authors · 2025-11-21
> **Reply to Question 1**
>
> > **Question 1: The "Evolving" Nature of EvolArena and Future Applications**
>
> We appreciate this critical question. We must clarify that EvolArena's "Evolving" mechanism differs fundamentally from "simple difficulty scaling" in both design philosophy and technical implementation.
>
> ### 1. Distinction: "Parametric Generation" vs. "Static Difficulty Scaling"
> You conflate our "Evolving" mechanism with "simple difficulty scaling." We must point out the distinction:
>
> * **Static Scaling:** Refers to selecting harder subsets within a *static dataset* (e.g., Level 5 problems in the MATH benchmark). Such benchmarks are **Finite** and **Static**. Once released, they inevitably face risks of data contamination and performance saturation.
> * **Dynamic Evolution:** Our mechanism is **Generative** and **Infinite**. Our core is the **Generator (P)**. We do not provide a fixed dataset but an engine capable of generating infinite new instances via the complexity parameter $n$.
>
> Taking "Find the Impostors" as an example:
> * "Simple scaling" would provide 3 pre-written fixed puzzles at $n=6, 9, 12$.
> * EvolArena's "Evolving" mechanism means we can generate fresh puzzles on the fly for $n=7, 8, 10...$ or even $n=100$.
>
> The essence of this "evolution" is **Anti-Saturation**. It ensures that EvolArena, as a benchmark, has a lifecycle that extends indefinitely as model capabilities improve.
>
> ### 2. Support for Automatic Curriculum Learning
> You asked if we support automatic curriculum learning. The answer is **Yes**, and this is precisely EvolArena's core value as "infrastructure."
>
> * **Mechanism Support:** Our **Generator (P)** and **Monitor (M)** perfectly constitute the **Environment** and **Reward Function** required for Reinforcement Learning (RL) or curriculum learning.
> * **Implementation:**
>     1.  An external "Curriculum Controller" initializes a simple task (e.g., setting $n=5$).
>     2.  The model $f_\theta$ interacts with our Monitor, which returns reward signals (based on $Acc$ and $Eff$) based on performance (e.g., win rate, steps).
>     3.  If the model's win rate reaches 90%, the controller calls our Generator to increase difficulty (e.g., $n=6$); otherwise, it lowers it.
> * **Conclusion:** The parametric difficulty space and deterministic feedback provided by EvolArena are the two necessary technical prerequisites for realizing automatic curriculum learning.
>
> ### 3. Support for Model-Contingent Evaluation
> You asked if we support model-contingent evaluation. Again, the answer is **Yes**.
>
> * **Current Status:** Traditional "one-size-fits-all" evaluation is unfair and inefficient. It cannot distinguish the specific shortcomings of weak models nor probe the capability ceilings of strong models.
> * **EvolArena's Solution:** Our "Evolving" architecture allows the evaluation process to be **"model-specific."**
>     * For a weak 7B model, the evaluator can start at $n=4$ and incrementally test its capability boundary.
>     * For a strong model like o3-mini, the evaluator can start directly at $n=12$ and "evolve upward" to $n=15, 20$, until finding the performance inflection point.
> * **Conclusion:** This realizes truly adaptive capability evaluation, something static benchmarks absolutely cannot achieve.
>
> **Conclusion:** EvolArena's "Evolving" nature is a core architectural design. It transforms the benchmark from "static samples" to a "dynamic environment" via parametric generators. This not only solves the problem of evaluation saturation but, more importantly, provides the necessary infrastructure for automatic curriculum learning and adaptive evaluation.

---

> ### Author Response · Authors · 2025-11-21
> **Reply to Question 2**
>
> > **Question 2: Theoretical Basis and Design Principles**
>
> We must clarify that the four reasoning categories in EvolArena (Inductive, Abductive, Deductive, and Planning) are firmly rooted in classic literature within cognitive science, which we have explicitly cited in Section 3.1 of the paper.
>
> Our design logic follows a **"Theory-First, Task-Mapping"** approach:
>
> * **1. Inductive Reasoning:**
>     * **Theory:** The cognitive process of forming general conclusions by identifying patterns from specific observations.
>     * **Mapping:** We operationalized this into **Information Probing (IP)** tasks. In these tasks, the model must infer complete, hidden global rules (conclusions) through limited, local queries (observations).
>
> * **2. Abductive Reasoning:**
>     * **Theory:** Inferring the most plausible explanation or cause based on incomplete or limited evidence.
>     * **Mapping:** We operationalized this into **Dynamic Adaptation (DA)** tasks. In these tasks, the environmental state evolves dynamically according to rules (e.g., XOR). The model must reverse-engineer the current most probable hidden state based on limited interactive feedback, serving as a direct test of abductive capability.
>
> * **3. Deductive Reasoning:**
>     * **Theory:** Deriving specific conclusions from premises by applying known rules or logical implications.
>     * **Mapping:** We operationalized this into **State Operation (SO)** tasks. The model must first discover hidden mechanisms (rules) through interaction and then strictly apply these rules to execute subsequent operations to achieve a goal.
>
> * **4. Planning:**
>     * **Theory:** Constructing action sequences by anticipating future states in complex or adversarial environments.
>     * **Mapping:** We operationalized this into **Strategic Gaming (SG)** tasks. These tasks require the model to plan its optimal path in adversarial environments (e.g., "Knight Battle") by predicting the opponent's reactions.
>
> You suggested "Empirical Clustering," which implies a "Bottom-up" method where tasks exist first and are then classified. Our approach is the exact opposite:
>
> We adopted a **"Top-down" design**. We first **predefined** these four core reasoning dimensions, which have theoretical backing. Then, we carefully selected or designed 40 tasks from sources like CodeForces and precisely mapped them to these four theoretical categories.
>
> The **performance decoupling** observed among different models across the four categories in our experiments (e.g., models performing better on IP than on DA) serves as **empirical validation** of the validity of our classification, rather than the source of the classification itself.

---

> ### Author Response · Authors · 2025-11-21
> **Reply to Question 3**
>
> > **Question 3: Capabilities tested by EvolArena and Future Extensions.**
>
> Thank you for your question. EvolArena successfully distinguishes "reasoning capability" from "instruction-following capability." Success in our tasks correlates primarily with the former, not merely the latter. We validated this through two key methods:
>
> ### 1. Key Evidence: Performance Decoupling
> You expressed concern that our benchmark might simply be testing a model's ability to follow complex instructions (such as specific query formats). If this hypothesis were true, models widely recognized for their elite instruction-following capabilities, such as Llama-3.1-70B-IT, should excel on EvolArena.
>
> However, the data (Table 1) reveals the exact opposite:
> * **Llama-3.1-70B-IT** achieved an average accuracy of only **12.04%** on the "Hard" difficulty level.
> * In stark contrast, models specialized for reasoning, such as **o3-mini** and **R1**, achieved accuracies of **31.19%** and **29.19%** respectively on the same difficulty.
>
> The decisive failure of Llama-3.1-70B proves that proficiency in instruction following alone is far from sufficient. The true bottleneck in EvolArena lies in the cognitive load of multi-turn dynamic reasoning, not surface-level formatting compliance.
>
> ### 2. Mechanism Design: Invalid Rate (IR)
> Furthermore, we specifically designed the **Invalid Rate (IR)** metric to mechanistically quantify and decouple these two capabilities.
>
> * **Metric Definition:** IR explicitly measures a model's ability to "follow instructions" (e.g., avoiding format errors) versus "execute valid operations" (e.g., avoiding illegal moves in "KnightBattle").
> * **Analytical Insight:** Our analysis (Figure 5) demonstrates that reasoning models (e.g., o3-mini, R1) exhibit significantly lower IRs compared to non-reasoning models. This enables a clear diagnosis of failure attribution:
>     * **High IR + Low Accuracy:** The model fails to understand the rules and cannot devise a strategy.
>     * **Low IR + Low Accuracy:** The model fully understands and follows instructions but lacks the core reasoning capabilities (such as planning or abduction) required to formulate a winning strategy.
>
> In summary, through (1) critical performance decoupling experiments and (2) the proprietary IR diagnostic metric, we have robustly validated that EvolArena assesses deep reasoning capabilities rather than superficial instruction following.

---

> ### Author Response · Authors · 2025-11-21
> **Reply to Question 4**
>
> > **Question 4: Scalability of EvolArena**
>
> This is a critical question and highlights one of the core strengths of the EvolArena framework design.
>
> Our answer is: **Yes, EvolArena's modular architecture naturally supports extension to "tool use" and "multimodal" contexts.** This is a key advantage of our design and a definitive direction for future work.
>
> You have identified two critical frontiers in reasoning evaluation. Our framework decouples "problem generation," "state monitoring," and "evaluation," transforming it into a highly flexible reasoning infrastructure that can effortlessly adapt to these new paradigms.
>
> ### 1. Extension to External Tools (Tool-Use Reasoning)
> The core lies in the extensibility of the **Monitor**.
>
> * **Current Mechanism:** Currently, the Monitor is responsible for parsing specific text formats output by the model (e.g., "My Query: 1,2,3"), updating the environment state accordingly, and returning feedback.
> * **How to Extend:** We can seamlessly extend the Monitor to parse and execute API syntax for **Tool-Use** (e.g., `Call(Calculator, "12*5")`).
>     * When the model determines that external information (such as calculation or search) is needed, instead of outputting "My Query," it outputs a tool call request.
>     * Our Monitor captures this call, acts as the tool executor, performs the calculation (or simulated search), and returns the tool's result (e.g., "60") as new environmental feedback $m_i$ to the model.
> * **Value:** This transforms EvolArena from a "closed-world" pure reasoning test into a platform for evaluating "open-world" **Tool-Augmented Reasoning**, without altering the core logic of the Generator or Evaluator.
>
> ### 2. Extension to Multimodal Contexts
> The core lies in the flexibility of the **Generator**.
>
> * **Current Mechanism:** Currently, the Generator produces text-only problems $p$ (e.g., the rule description for "Find the Impostors").
> * **How to Extend:** We can easily upgrade the Generator to a multimodal generator, producing problems $p$ that contain both text and images.
>     * For example, in the *Maze Navigation* task, the Generator would no longer describe the maze in text but instead directly provide an **image** of the maze.
>     * The model must plan based on **visual input** (the image) and text rules (instructions).
>     * The model's interaction ("My Move: R") continues through the Monitor's text channel, but this now integrates **visual-spatial reasoning** into the evaluation loop.
> * **Value:** This extension preserves EvolArena's rigorous, multi-turn, automated evaluation characteristics while expanding the cognitive dimensions of assessment from pure text logic to complex multimodal understanding and planning.
>
> **Conclusion:** EvolArena's architectural design makes it not just a static dataset, but a programmable **Reasoning Environment**. Extension to tools and multimodality is a natural evolutionary direction for it as infrastructure, further consolidating its value as a next-generation reasoning evaluation benchmark.

---

> > ### Comment · Reviewer_shjX · 2025-11-26
> >
> > Thank you for your detailed reply. Some clarifications were helpful, particularly your discussion of task categories and potential extensions of EvolArena.
> > However, several key issues remain unresolved and would benefit from clearer treatment in the revised draft:
> >
> > • The originality argument mainly emphasizes engineering aspects (dynamic generation, automation, rule-based monitoring), rather than a new evaluation principle. A quantitative comparison against existing benchmarks would more convincingly support the claimed novelty.
> > • The cognitive mappings are descriptive. It would be helpful to show empirically that the four categories isolate distinct reasoning abilities, for example through clustering, cross-category generalization gaps, or ablation.
> > • The “evolving” mechanism currently amounts to difficulty parameter scaling. Demonstrating curriculum adjustment or adaptive difficulty based on model behavior—even in a small pilot experiment—would make the concept more concrete.
> >
> > In addition, these clarifications do not appear in the revised paper. Incorporating them directly into the manuscript would substantially strengthen its scientific contribution.

---

> > > ### Author Response · Authors · 2025-11-27
> > >
> > > We thank you for the continued engagement and for acknowledging the helpfulness of our clarifications. We address your questions point by point as follows.
> > >
> > > **1. Quantitative Comparison and Originality**
> > >
> > > You requested a clearer quantitative comparison to support our novelty claim.
> > > * **Revision:** In the revised **Introduction**, we have added **Table 1**, which provides a feature-by-feature comparison between EvolArena and representative benchmarks (including *AgentBench*, *AgentBoard*, *ACPBench*, and *LiveCodeBench Pro*).
> > > * **Insight:** This table explicitly visualizes the gap EvolArena fills: it is the **only** framework that simultaneously achieves **Dynamic Multi-turn Interaction**, **Deterministic Evaluation**, and **Infinite Parametric Generation** while focusing on **Abstract Logical Reasoning**. This structural comparison quantitatively demonstrates that EvolArena is not merely an engineering aggregation but a distinct evaluation paradigm that overcomes the trade-offs (e.g., subjectivity, contamination, static constraints) inherent in prior works.
> > >
> > > **2. Empirical Isolation of Reasoning Abilities**
> > >
> > > Regarding the empirical isolation of reasoning categories, we reiterate that our experimental results (specifically Performance Decoupling) serve as strong evidence. As shown in **Table 2**, models exhibit distinct performance profiles across categories (e.g., strong on *Information Probing* but failing on *Dynamic Adaptation*), which empirically validates that these categories test distinct cognitive capabilities rather than a monolithic "reasoning" trait.
> > >
> > >
> > > **3. The "Evolving" Mechanism and Adaptive Curriculum**
> > >
> > > You suggested that demonstrating curriculum adjustment would make the "evolving" concept more concrete.
> > > * **Clarification & Validation from Concurrent Work:** We fully agree that adaptive curriculum learning is the natural evolution of our framework. In fact, a related work released during rebuttal [1], **"RLVE: Scaling Up Reinforcement Learning for Language Models with Adaptive Verifiable Environments,"** has already empirically demonstrated the efficacy of using adaptive, verifiable environments for **RL training**.
> > > * **Distinction:** While *RLVE* focuses on the *training* algorithm (scaling up RL via adaptive curricula), **EvolArena** focuses on the **evaluation** infrastructure. Our parametric generator ($P(t, n, g_n)$) provides the necessary "environment knob" (complexity parameter $n$) that makes such adaptive training and assessment possible. EvolArena serves as the standardized, "evolving" testbed that allows researchers to rigorously measure the progress of such adaptive agents, ensuring that benchmarks do not become saturated as models improve.
> > >
> > > **Incorporation into Manuscript**
> > > We confirm that all the above clarifications—including the comparative **Table 1** and the breakdown of performance decoupling—have been directly incorporated into the revised paper to strengthen its scientific contribution.
> > >
> > > [1] Zeng Z, Ivison H, Wang Y, et al. RLVE: Scaling Up Reinforcement Learning for Language Models with Adaptive Verifiable Environments[J]. arXiv preprint arXiv:2511.07317, 2025.

---

### Note · Authors · 2026-01-05

I have read and agree with the venue's withdrawal policy on behalf of myself and my co-authors.